# The Checklist of Sicilian Macrofungi: Second Edition

**DOI:** 10.3390/jof8060566

**Published:** 2022-05-25

**Authors:** Valeria Ferraro, Giuseppe Venturella, Fortunato Cirlincione, Giulia Mirabile, Maria Letizia Gargano, Pasqualina Colasuonno

**Affiliations:** 1Department of Agricultural, Food and Forest Sciences, University of Palermo, Viale delle Scienze, Bldg. 5, 90128 Palermo, Italy; valeria.ferraro@unipa.it (V.F.); giuseppe.venturella@unipa.it (G.V.); giulia.mirabile@unipa.it (G.M.); 2Department of Agricultural and Environmental Science, University of Bari Aldo Moro, Via Amendola 165/A, 70126 Bari, Italy; pasqualina.colasuonno@uniba.it

**Keywords:** biodiversity, basidiomycota, ascomycota, ecology, distribution, number of fungi, Sicily, open science, the network for the study of mycological diversity

## Abstract

Approximately 30 years after the publication of the first Sicilian checklist of macrofungi, a new updated version is presented here. The census of macromycetes was carried out through periodic observations in different agricultural and forest ecosystems, in urban areas, in public and private gardens, and in botanical gardens. The 1919 infraspecific taxa included in 508 genera belonging to 152 families were collected in the Sicilian territory. Ectomycorrhizal fungi are the most represented ecological category, followed by saprotrophs on wood, saprotrophs on litter, and terricolous saprotrophs. The interest in this rich group of organisms is evidenced by the nutritional and therapeutic value of a high percentage of species. The actions linked to the National Recovery and Resilience Plan and The Network for the Study of Mycological Diversity will further increase the number of macrofungi for Sicily in the future.

## 1. Introduction

The checklists are modern tools for evaluation of fungal diversity and valuable papers to highlight ecological data, and tips for management and exploitation of protected areas, agro- and forest ecosystems. Relevant studies were published around the world—in Europe and in Italy [1,2,3,4,5,6,7,8].

The assessment of fungal diversity in the Italian regions is still incomplete and needs further investigation. In most Italian regions, it is difficult to document the exact number of mushrooms due to lack of mycologists. Even greater difficulty is faced in the evaluation of the presence of hypogeous macrofungi as few research groups have dogs trained to harvest.

The territories included in the Med-Checklist consider the Italian peninsula as a separate region from a floristic point of view, and its two major islands, Sicily and Sardinia, are separately coded [9]. The environmental peculiarities of Sicily in close relation to different habitats and ecosystems [10] determine high levels of biodiversity in all groups of organisms (plants, lichens, fungi, bryophytes, algae, and insects).

On the basis of literature data reported from 1814 to 1991, a preliminary survey on fungal diversity in Sicily was carried out by Venturella [11], with an estimate of approximately 750 macromycetes. Subsequently, Venturella et al. [12] published a report on the state of fungal diversity in Italy and analyzed the number of macrofungi for each region. This study showed that Sicily boasts a high number of macrofungi compared to other Italian regions.

From 1991 to present, the study of macromycete diversity has continued unabated through an intensive exploration of forest ecosystems, natural parks, nature reserves, public and private gardens, botanical gardens, cultivated and uncultivated lands.

Based on data collected during 30 years of long-term observation, an up-to-date assessment of fungal diversity of Sicily is reported in this paper.

## 2. An Outline of Vegetation Types of Sicily

The physiographic uniqueness of Sicily (southern Italy) coupled with the paleogeographic vicissitudes, over time have determined the progressive evolution of a rich vascular flora (ca. 3000 species) and the presence of a number of rare species or in any case of considerable taxonomic significance [13]. In agreement with Fenaroli and Giacomini [14] and Di Martino and Raimondo [15], Sicily is considered as a floristic area in itself, well characterized by a considerable endemic contingent (approximately l% of the entire flora), which can be defined as a Sicilian domain according to phyto-corological criteria.

The forest area of Sicily corresponds to 512,121 hectares and includes 58 types of forest vegetation [16]. The coastal and hilly belt is characterized by evergreen oak forests with *Quercus ilex* L. subsp. ilex, *Q. coccifera* L., and *Q. suber* L. as well as semi-evergreen oaks *Quercus pubescens* Willd. s.l. and *Q. virgiliana* (Ten.) Ten. In the warmer areas near the sea, forests can be replaced by aspects of Mediterranean maquis such as those characterized by *Juniperus turbinata* Guss., *J. oxycedrus* L. subsp. *oxycedrus*, *J. macrocarpa* Sm., *Pistacia lentiscus* L. and other sclerophyllous shrubs such as *Cistus creticus* L., *C. salvifolius* L. and *C. monspeliensis* L.

In the areas most subject to disturbance, the vegetation is currently represented by *Ampelodesmos mauritanicus* (Poir.) T. Durand & Schinz and other types of grassland and shrub communities. Even rarer are the natural forests of Mediterranean conifers such as *Pinus halepensis* Mill., *P. pinea* L. and *P. pinaster* Aiton, which are considered pioneer plants. In the hills, the natural forest vegetation, in the hottest and driest conditions, consists of mixed forests of deciduous oaks with dominance of *Q. pubescens*, while in cooler climatic conditions, with a marked oceanic climate, are more diffused *Q. cerris* L., *Q. petraea* (Matt.) Liebl and the endemic *Q. gussonei* (Borzí) Brullo. These species have in some cases been replaced for agricultural purposes or for the use of wood by *Corylus avellana* L., *Fraxinus ornus* L., and *Castanea sativa* Mill. The degradation aspects are represented by mesophilic shrubs dominated by different taxa belonging to the family of Rosaceae such as *Pyrus pyraster*. (L.) Burgsd. and *P. amygdaliformis* Vill. The forest vegetation of the mountain area is characterized by forests of *Fagus sylvatica* L. sometimes mixed with *Acer pseudoplatanus* L., *Quercus petraea* (Mattuschka) Liebl., *Taxus baccata* L. and *llex aquifolium* L. An exclusive characteristic of Sicily is the presence of forests of *Q. ilex* that reach an altitude of 1500 m well above the maximum elevation found in other parts of Italy. Of considerable interest is also the presence of different endemic trees and shrubs and evidence of relict forest vegetation such as *Abies nebrodensis* (Lojac.) Mattei, *Betula aetnensis* Raf., *Celtis aetnensis* (Tornab.) Strobl, *Populus tremula* L., *Pinus laricio* Poir., *Quercus gussonei* (Borzì) Brullo, *Zelkova sicula* Di Pasquale, Garfi & Quézel, and *Genista aetnensis* (Raf. ex Biv.) DC. Upland shrubs such as *Sorbus graeca* (Spach) Schauer and *Berberis aetnensis* C. Presl. are found in some areas lacking tree forest vegetation. Woody species that vegetate above the edge of the forest are found only on Mount Etna, but also in the Madonie. This vegetation is dominated by the shrubs of *Astragalus nebrodensis* on the Madonie and by *A. siculum* on Mount Etna.

The presence of *Fagus sylvatica* L. in Sicily can therefore be considered a remnant of the glaciations, when conditions had to be favorable even at lower altitudes.

The vegetation of the watercourses is characterized by typical riparian species such as *Salix pedicellata* Desf., *S. alba* L. and *S. purpurea* L., associated with *Populus nigra* L., *P. alba* L., *Laurus nobilis* L., *Sambucus ebulus* L., *Cornus sanguinea* L., *Prunus mahaleb* L. and *Alnus cordata* (Loisel.) Desf. [17].

In the sandy dunes, some plant associations such as *Salsolo kali*–*Cakiletum maritimae* Costa & Mansanet 1981, corr. Rivas-Martínez et al. 1992 and *Salsolo kali*–*Euphorbietum peplis* Géhu et al. 1984 are distributed.

## 3. Materials and Methods

The census of macromycetes was carried out through periodic observations, weekly or fortnightly (monthly during summer), in different agricultural and forest ecosystems, in urban areas, in public and private gardens, and in botanical gardens. For each species, the fresh basidiomata and ascomata were collected and subsequently identified on the basis of macroscopic (pileus, flesh, lamellae, stipe, and type of occurrence: solitary, grouped, clustered, cespitose, color of spore prints, etc.) and microscopic characters (spores, basidia, asci, cystidia, pileipellis, element of the stipe surface, etc.), also with the help of distilled water, immersion oil, chemical reagents (Melzer’s reagent, KOH, Ammoniated Congo Red, Cotton blue-lactic acid), and analytical keys related to the different genera. Most specimens were identified solely on the basis of morphological characteristics; for some critical groups (e.g., *Daldinia* and *Pleurotus*), reference was made to previous studies using molecular analysis [18,19,20]. The dried specimens, prepared in a universal dryer 475 Watt stainless steel structure with 5 baskets, are kept in the Herbarium SAF of the Department of Agricultural, Food and Forest Sciences (SAAF) of the University of Palermo. The nomenclature of fungi follows Index Fungorum while that of plants refers to Euro + Med PlantBase.

## 4. Results

The 1919 infraspecific taxa included in 508 genera belonging to 152 families were collected in the Sicilian territory (Table 1).

Families with the highest number of species are Russulaceae (164 taxa) and Agaricaceae (125 taxa).

According to Index Fungorum, 131 taxa of uncertain taxonomic classification are reported in Table 1 as *Incertae sedis.* Other families, with number of infraspecific taxa greater than 50 (Figure 1), are Boletaceae (78), Cortinariaceae (72), Hygrophoraceae (63), Inocybaceae (58), Hymenogastraceae (56), Tricholomataceae (54), Amanitaceae (51), Pyronemataceae (51), and Psathyrellaceae (51).

The highest number of infraspecific taxa (Figure 2) is found in the province of Palermo (1610) followed by in Messina (1546) and Catania (1517). The province with the lowest number of infraspecific taxa is Caltanissetta (330).

The taxa listed in Table 1 belong to 15 ecological categories. For ten taxa, the ecological category to be assigned is unclear and is therefore listed as Unknown (UNK).

Ectomycorrhizal fungi (637 taxa) are the most represented ecological category (Figure 3), followed by saprotrophs on wood (484 taxa), saprotrophs on litter (386 taxa), terricolous saprotrophs (279 taxa), saprotrophs on mosses (39 taxa) saprotrophs on dung (39 taxa), saprotroph on burnt ground (19 taxa), necrotroph parasites (8 taxa), saprotrophs on leavers (5 taxa), and saprotrophs on humus (2 taxa). Three saprotrophic species characterized by particular substrates such as exuviae of different insects [*Cordyceps militaris* (L.) Fr.], cupules of *Castanea sativa* Miller [*Lanzia echinophila* (Bull.) Korf], and cladodes of Opuntia ficus-indica [*Pleurotus opuntiae* (Durieu & Lév.) Sacc.] complete the list.

## 5. Discussion

Fungi are recognized worldwide as fundamental components in every type of ecosystem [21]. After a long period in which fungi have received insufficient consideration in biodiversity studies [22], in the last twenty-five years, there has been a growing interest in this group of living organisms, which are of great importance as a food source for future generations [23], potential remedies for human and animal diseases [24], and for the protection of the environment [25].

The climatic features of Sicily are favorable to the development of basidiomata and ascomata of different fungal species. The observations carried out during the present study showed a constant presence throughout the year of macrofungi in ecosystems. Most species are collected in fall, although the presence of macrofungi appears to be strongly affected in some years by prolonged periods of drought. In the woods of the hinterland, it is possible to have optimal conditions of temperature and humidity able to favor the appearance of macrofungi even in the summer. In Sicily, spring is not a season of abundance for fungal fruiting and ascomycetes predominate among species. In winter and summer, pathogenic basidiomycetes and wood saprotrophs can be predominantly observed.

Due to the presence of the three major regional parks (Madonie Park, Nebrodi Park, and Etna Park) in these territories, the largest number of infraspecific taxa is found in the provinces of Palermo, Messina and Catania. In the other provinces, reafforestations are prevalent and thus the composition of fungal coenoses is reduced in quality and quantity.

The greatest number of species is found within beech forests. The low night temperatures, the summer storms and the presence of occult precipitations determine, starting from the second decade of August, the early appearance of macrofungi that are generally found, at lower altitudes, in autumn within the forest ecosystems. A rich number of lignicolous species, mostly pathogenic and to a lesser extent saprotrophic, has also been surveyed within beech forests, growing on trees, branches of different sizes that have fallen to the ground and stumps. A similar condition of prevalence of mycorrhizal species is found in oak and holly forests and in chestnut groves. In the latter type of forest, an important role in the decomposition of the abundant litter layer is provided by *Marasmius bulliardii* while it is very common to observe *Fistulina hepatica* at the base of chestnut trees. In *Quercus ilex* forests, fungi responsible for wood decay (*Armillaria mellea*, *Fuscoporia torulosa*, *Cyclocybe cylindracea*, *Laetiporus sulphureus*, and *Daldinia raimundi*) are observed on stumps and trunks. Compared with other oak forests, fewer macrofungi are reported *Quercus suber* forests and in the maquis. This is due to the greater state of degradation of the forest coenosis due to frequent fires that cause a significant reduction in the number of mycorrhizal species and saprotrophs, the latter negatively affected by the drastic reduction in plant residues. In pastures, grasslands, and garrigues, the number of mycorrhizal species is limited to those of shrubs of the genus *Cistus* (*Lactarius cistophilus* and *Leccinellum corsicum*). Saprotrophs on litter and lignicolous species are widespread in riparian vegetation.

The high percentage of mycorrhizal species within the natural forests and reforestations of Sicily allows to exclude at the moment a decline in forest ecosystems. In some forested areas, frequent fires cause a reduction in mycorrhizal species and saprotrophs on litter due to the drastic reduction in plant residues. Conversely, silvicultural practices and the resulting failure to remove woody debris of different sizes promote the appearance of crusty, shelf-like, or gelatinous lignicolous macrofungi.

The presence of prized and edible truffles, especially within the holm oak and pine woods, is of relevant interest especially for the areas of the Sicilian hinterland characterized by a depressed economy [26]. These prized truffles and other mushrooms of excellent edibility and high organoleptic qualities (e.g., *Pleurotus* ssp.) represent a valuable example for the economic enhancement of fungal diversity in Sicily. Of great interest is the presence on the Sicilian territory of rare species classified according to IUCN criteria as Vulnerable (*Alessioporus ichnusanus*) [27], Least Concern (*Poronia punctata*) [28], and Endangered (*Pleurotus nebrodensis*) [29]. Mushrooms are considered healthy food components to be preserved for future generations in terms of nutritional, bioactive and therapeutic value. Among the species surveyed in Sicily, there are medicinal mushrooms, such as *Grifola frondosa*, *Ganoderma lucidum*, and *Ericium erinaceus*, with proven pharmacological activities including antibacterial, antifungal, antiviral, cytotoxic, immunomodulating, anti-inflammatory, antioxidative, antiallergic, antidepressive, antihyperlipidemic, antidiabetic, digestive, hepatoprotective, neuroprotective, nephroprotective, osteoprotective, and hypotensive activities.

## 6. Conclusions

Although the number of fungal species is still in constant flux, Sicily is among the Italian regions with the highest diversity of fungal species, many of which are of potential applicative interest and economic interest. Therefore, it is necessary to raise the level of knowledge and attention on the part of local communities towards this group of organisms. The data reported in this survey strongly increase the knowledge on distribution and ecology of macromycetes in Sicily and may help better valorize such non-timber products. The outcome of the checklist carried out in Sicily could provide the Regional Administration with useful information for the conservation and exploitation of fungi in natural habitats, with particular reference to forest ecosystems. Legislation should limit the level of danger existing for the conservation of fungal diversity and support such strategies with more rigorous limits against wild edible mushroom overharvesting. The cultivation of some prized edible mushrooms, i.e., *Pleurotus* and *Tuber* species, can provide an important income in agroecosystems, especially in rural or marginal areas.

The National Recovery and Resilience Plan (NRRP) and The Network for the Study of Mycological Diversity (NSMD), recently launched, will contribute to implement the monitoring of mycological biodiversity within national terrestrial habitats.

These actions will be carried out with the contribution of institutions such as the Higher Institute for Environmental Protection and Research (ISPRA) of the Italian Ministry of Ecological Transition and the Working Group for Mycology of the Italian Botanical Society through an Open Science initiative.

## Figures and Tables

**Figure 1 jof-08-00566-f001:**
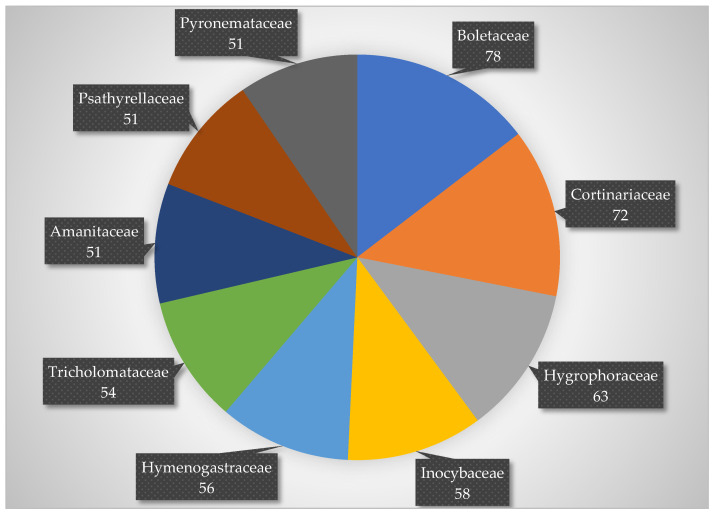
Most representative families with number of taxa greater than 50.

**Figure 2 jof-08-00566-f002:**
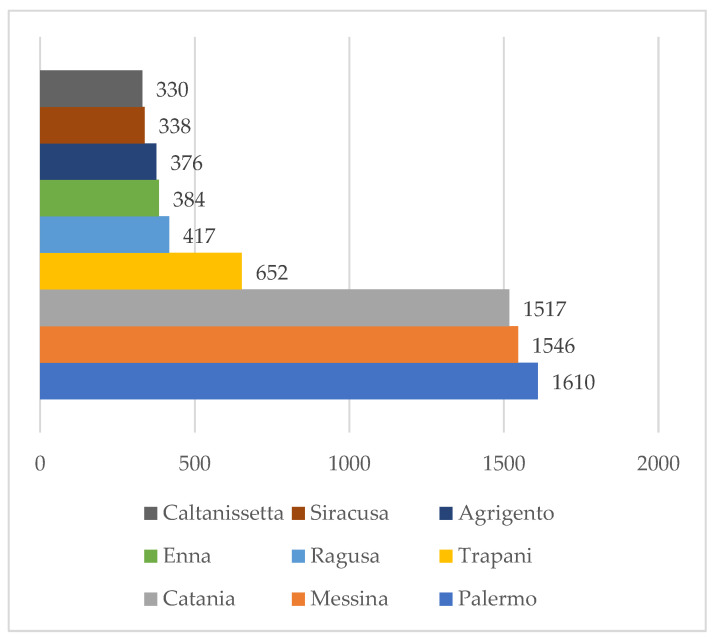
Number of taxa per provinces.

**Figure 3 jof-08-00566-f003:**
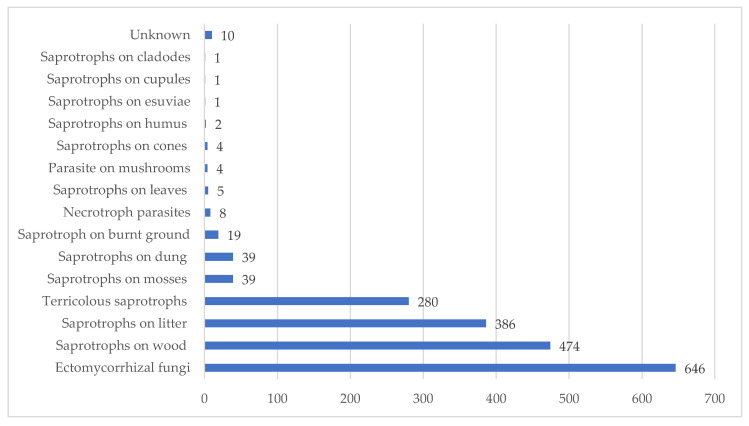
Number of taxa per ecological categories.

**Table 1 jof-08-00566-t001:** List of recorded taxa with indications on family, province and ecological category. *Provinces:* AG = Agrigento, CL = Caltanissetta, CT = Catania, EN = Enna, ME = Messina, PA = Palermo, RG = Ragusa, SR = Siracusa, and TP = Trapani. *Ecological categories*: Em = Ectomycorrhizal, Pm = Parasite on mushrooms, Pn = Necrotroph parasites, Sbg = Saprotrophs on burnt ground, Sd = Saprotrophs on dung, Sc = Saprotrophs on cones, Scl = Saprotrophs on cladodes, Scu = Saprotrophs on cupules, Se = Saprotrophs on esuviae, Sh = Saprotrophs on humus, Sl = Saprotrophs on litter, Sle = Saprotrophs on leaves, Sm = Saprotrophs on mosses, St = Terricolous saprotrophs, Sw = Saprotrophs on wood, and UNK = Unknown.

Taxon	Family	Provinces	Ecological Categories
*Abortiporus biennis* (Bull.) Singer	Podoscyphaceae	AG, CL, CT, EN, ME, PA, RG, SR, TP	Sw
*Agaricus altipes* (F.H. Møller) F.H. Møller	Agaricaceae	AG, CL, CT, EN, ME, PA, RG, SR, TP	St
*Agaricus aridicola* Geml, Geiser & Royse ex Mateos, J. Morales et al.	Agaricaceae	CT, ME, PA, RG, SR, TP	St
*Agaricus arvensis* Schaeff.	Agaricaceae	AG, CL, CT, EN, ME, PA, RG, SR, TP	St
*Agaricus augustus* Fr. var. *augustus*	Agaricaceae	CT, ME, PA, TP	St
*Agaricus benesii* (Pilát) Pilát	Agaricaceae	CT, ME, PA	St
*Agaricus bernardii* Quél.	Agaricaceae	CT, ME, PA	St
*Agaricus bisporatus* Contu	Agaricaceae	CL, CT, ME,	St
*Agaricus bisporus* (J. E. Lange) Imbach	Agaricaceae	CT, ME, PA	St
*Agaricus bitorquis* (Quél.) Sacc.	Agaricaceae	CT, ME, PA	St
*Agaricus bohusii* Bon	Agaricaceae	CT, ME, PA	St
*Agaricus bresadolanus* Bohus	Agaricaceae	CT, ME, PA	St
*Agaricus brunneolus* (J.E. Lange) Pilát	Agaricaceae	CT, ME	St
*Agaricus campestris* L.	Agaricaceae	AG, CL, CT, EN, ME, PA, RG, SR, TP	St
*Agaricus cappellianus* Hlaváček	Agaricaceae	CT, ME, PA	St
*Agaricus chionodermus* Pilát	Agaricaceae	CT, ME, PA	St
*Agaricus comtulus* Fr.	Agaricaceae	AG, CT, ME, PA	St
*Agaricus cupreobrunneus* (Jul. Schäff. & Steer) Pilát	Agaricaceae	CT, ME, PA, TP	St
*Agaricus cupressicola* Bon & Grilli	Agaricaceae	CL, CT, ME, PA	St
*Agaricus devoniensis* P.D. Orton	Agaricaceae	CT, ME, PA, RG, SR	St
*Agaricus fuscofibrillosus* (F.H. Møller) Pilát	Agaricaceae	CT, ME	St
*Agaricus gennadii* (Chatin & Boud.) P.D. Orton	Agaricaceae	CL, CT, ME	St
*Agaricus impudicus* (Rea) Pilát	Agaricaceae	CT, ME	St
*Agaricus iodosmus* Heinem.	Agaricaceae	CT, ME, PA	St
*Agaricus langei* (F.H. Møller) F.H. Møller	Agaricaceae	CL, CT, ME, PA	St
*Agaricus lanipes* (F.H. Møller & Jul. Schäff.) Hlaváček	Agaricaceae	CT, ME, PA	St
*Agaricus lepiotoides* Berk. & Broome	Agaricaceae	CT, ME, PA	St
*Agaricus litoralis* (Wakef. & A. Pearson) Pilát	Agaricaceae	CT, ME, PA, TP	St
*Agaricus luteomaculatus* F.H. Møller	Agaricaceae	PA, TP	St
*Agaricus lutosus* F.H. Møller	Agaricaceae	CT, ME	St
*Agaricus menieri* Bon	Agaricaceae	CT, PA, RG	St
*Agaricus moelleri* Wasser	Agaricaceae	CT, ME, PA	St
*Agaricus moellerianus* Bon	Agaricaceae	CT, ME	St
*Agaricus osecanus* Pilát	Agaricaceae	CT, ME, PA	St
*Agaricus phaeolepidotus* F.H. Møller	Agaricaceae	CT, ME, PA	St
*Agaricus pilatianus* (Bohus) Bohus	Agaricaceae	AG, CL, CT, ME, PA	St
*Agaricus placomyces* Peck	Agaricaceae	CT, ME, PA	St
*Agaricus porphyrizon* P.D. Orton	Agaricaceae	PA, TP	St
*Agaricus praeclaresquamosus* A.E. Freeman var. *praeclaresquamosus*	Agaricaceae	PA, TP	St
*Agaricus pseudopratensis* (Bohus) Wasser	Agaricaceae	CL, ME, PA, SR	St
*Agaricus semotus* Fr.	Agaricaceae	CT, ME	St
*Agaricus subperonatus* (J.E. Lange) Singer	Agaricaceae	CT, ME, PA	St
*Agaricus sylvaticus* Schaeff.	Agaricaceae	CT, ME, PA, TP	St
*Agaricus sylvicola* (Vittad.) Peck	Agaricaceae	AG, CL, CT, ME, PA	St
*Agaricus pilatianus* (Bohus) Bohus	Agaricaceae	CT, ME, EN, PA, TP	St
*Agaricus urinascens* (Jul. Schäff. & F.H. Møller) Singer	Agaricaceae	CL, CT, ME, PA	St
*Agaricus xanthodermus* Genev. subsp. *xanthodermus*	Agaricaceae	AG, CL, CT, EN, ME, PA, RG, SR, TP	St
*Agaricus xanthodermus* var. *griseus* (A. Pearson) Bon & Cappelli	Agaricaceae	AG, CL, CT, EN, ME, PA, RG, SR, TP	St
*Agrocybe molesta* (Lasch) Singer	Strophariaceae	CT, ME, PA	St
*Agrocybe pediades* (Fr.) Fayod	Strophariaceae	CT, ME, PA	St
*Agrocybe praecox* (Pers.) Fayod	Strophariaceae	CL, PA, TP	St
*Agrocybe putaminum* (Maire) Singer	Strophariaceae	PA, TP	St
*Agrocybe vervacti* (Fr.) Singer	Strophariaceae	CT, ME, PA	St
*Albatrellopsis confluens* (Alb. & Schwein.) Teixeira	Albatrellaceae	AG, CL, CT, EN, ME, PA, RG, SR, TP	Em
*Albatrellus ovinus* (Schaeff.) Kotl. & Pouzar	Albatrellaceae	CT, ME, PA	Em
*Aleuria aurantia* (Pers.) Fuckel	Pyronemataceae	AG, CL, CT, EN, ME, PA, RG, SR, TP	St
*Aleurodiscus disciformis* (DC.) Pat.	Stereaceae	CT, ME, PA	Sh
*Aleurodiscus dextrinoideocerussatus* Manjón, M.N. Blanco & G. Moreno	Stereaceae	PA, TP	Sh
*Alessioporus ichnusanus* (Alessio, Galli & Littini) Gelardi, Vizzini & Simonini	Boletaceae	CT, ME, PA, TP	Em
*Alutaceodontia alutacea* (Fr.) Hjortstam & Ryvarden	Schizoporaceae	CT, ME, PA	Sw
*Amanita albogrisescens* Contu	Amanitaceae	AG, CL, CT, EN, ME, PA, RG, SR, TP	Em
*Amanita baccata* (Fr.) Gillet	Amanitaceae	CT, ME, PA	Em
*Amanita badia* (Schaeff.) Bon & Contu	Amanitaceae	CT, ME, PA	Em
*Amanita battarrae* (Boud.) Bon	Amanitaceae	CT, ME, PA	Em
*Amanita beckeri* Huijsman	Amanitaceae	CT, ME, PA	Em
*Amanita boudieri* Barla	Amanitaceae	CT, ME, PA	Em
*Amanita caesarea* (Scop.) Pers.	Amanitaceae	AG, CL, CT, EN, ME, PA, RG, SR, TP	Em
*Amanita ceciliae* (Berk. & Broome) Bas	Amanitaceae	CT, ME, PA, TP	Em
*Amanita cistetorum* Contu & Pacioni	Amanitaceae	CT, ME, PA, TP	Em
*Amanita citrina* Pers.	Amanitaceae	AG, CL, CT, EN, ME, PA, RG, SR, TP	Em
*Amanita codinae* (Maire) Bertault	Amanitaceae	CT, ME, PA	Em
*Amanita crocea* (Quél.) Singer	Amanitaceae	AG, CL, CT, EN, ME, PA, RG, SR, TP	Em
*Amanita decipiens* (Trimbach) Jacquet.	Amanitaceae	CT, ME, PA	Em
*Amanita dryophila* Consiglio & Contu	Amanitaceae	CT, ME, PA	Em
*Amanita echinocephala* (Vittad.) Quél.	Amanitaceae	CT, ME, PA	Em
*Amanita eliae* Quél.	Amanitaceae	CT, ME, PA	Em
*Amanita excelsa* (Fr.) Bertill.	Amanitaceae	CT, ME, PA	Em
*Amanita franchetii* (Boud.) Fayod	Amanitaceae	CT, ME, PA	Em
*Amanita fulva* Fr.	Amanitaceae	AG, CL, CT, EN, ME, PA, RG, SR, TP	Em
*Amanita gemmata* (Fr.) Bertill.	Amanitaceae	AG, CL, CT, EN, ME, PA, RG, SR, TP	Em
*Amanita gracilior* Bas & Honrubia	Amanitaceae	CT, ME, PA	Em
*Amanita lactea* Malençon, Romagn. & D.A. Reid	Amanitaceae	CT, ME	Em
*Amanita lepiotoides* Barla	Amanitaceae	CT, ME, PA	Em
*Amanita lividopallescens* (Gillet) Bigeard & H. Guill.	Amanitaceae	AG, CL, CT, EN, ME, PA, RG, SR, TP	Em
*Amanita mairei* Foley	Amanitaceae	AG, CL, CT, EN, ME, PA, RG, SR, TP	Em
*Amanita magnivolvata* Aalto	Amanitaceae	CT, ME, PA	Em
*Amanita muscaria* (L.) Lam.	Amanitaceae	AG, CL, CT, EN, ME, PA, RG, SR, TP	Em
*Amanita ovoidea* (Bull.) Link	Amanitaceae	AG, CL, CT, EN, ME, PA, RG, SR, TP	Em
*Amanita pachyvolvata* (Bon) Krieglst.	Amanitaceae	CT, ME, PA	Em
*Amanita pantherina* (DC.) Krombh.	Amanitaceae	AG, CL, CT, EN, ME, PA, RG, SR, TP	Em
*Amanita phalloides* (Vaill. ex Fr.) Link	Amanitaceae	AG, CL, CT, EN, ME, PA, RG, SR, TP	Em
*Amanita ponderosa* Malençon & R. Heim	Amanitaceae	CT, ME	Em
*Amanita porphyria* Alb. & Schwein.	Amanitaceae	CT, ME, PA	Em
*Amanita proxima* Dumée	Amanitaceae	CT, ME, PA, TP	Em
*Amanita rubescens* Pers.	Amanitaceae	AG, CL, CT, EN, ME, PA, RG, SR, TP	Em
*Amanita singeri* Bas	Amanitaceae	CT, ME	Em
*Amanita strobiliformis* (Paulet ex Vittad.) Bertill.	Amanitaceae	CT, ME, PA	Em
*Amanita submembranacea* (Bon) Gröger	Amanitaceae	CT, ME	Em
*Amanita subnudipes* (Romagn.) Tulloss	Amanitaceae	AG, CL, CT, EN, ME, PA, RG, SR, TP	Em
*Amanita tarda* (Trimbach) Contu	Amanitaceae	CT, ME, PA	Em
*Amanita vaginata* (Bull.) Lam.	Amanitaceae	AG, CL, CT, EN, ME, PA, RG, SR, TP	Em
*Amanita verna* Bull. ex Lam.	Amanitaceae	AG, CT, ME, PA, TP	Em
*Amanita vittadinii* var. *vittadinii* (Moretti) Vittad.	Amanitaceae	CT, ME, PA	Em
*Amaropostia stiptica* (Pers.) B.K. Cui, L.L. Shen & Y.C. Dai	Incertae sedis	AG, CL, CT, EN, ME, PA, RG, SR, TP	Sw
*Amaurodon viridis* (Alb. & Schwein.) J. Schröt.	Thelephoraceae	CT, ME, PA, TP	Sw
*Amphinema byssoides* (Pers.) J. Erikss.	Atheliaceae	CT, ME, PA, TP	Em
*Ampulloclitocybe clavipes* (Pers.) Redhead, Lutzoni, Moncalvo & Vilgalys	Hygrophoraceae	CT, ME, PA	Sl
*Amylocorticiellum subillaqueatum* (Litsch.) Spirin & Zmitr.	Amylocorticiaceae	ME, PA	Sw
*Amylocorticium cebennense* (Bourdot) Pouzar	Amylocorticiaceae	CT, ME, PA, TP	Sw
*Amyloporia xantha* (Fr.) Bondartsev & Singer	Fomitopsidaceae	AG, CL, CT, EN, ME, PA, RG, SR, TP	Sw
*Amylostereum chailletii* (Pers.) Boidin	Echinodontiaceae	CT, ME	Sw
*Amylostereum laevigatum* (Fr.) Boidin	Echinodontiaceae	CT, ME, PA	Sw
*Annulohypoxylon cohaerens* (Pers.) Y.M. Ju, J.D. Rogers & H.M. Hsieh	Hypoxylaceae	CT, ME, PA	Sw
*Anthostoma gastrinum* (Fr.) Sacc.	Diatrypaceae	AG, CL, CT, EN, ME, PA, RG, SR, TP	Sw
*Anthracobia macrocystis* (Cooke) Boud.	Pyronemataceae	AG, CL, CT, EN, ME, PA, RG, SR, TP	Sbg
*Anthracobia melaloma* (Alb. & Schwein.) Arnould	Pyronemataceae	AG, CL, CT, EN, ME, PA, RG, SR, TP	Sbg
*Antrodia albida* (Fr.) Donk	Fomitopsidaceae	CT, ME, PA	Sw
*Antrodia gossypium* (Speg.) Ryvarden	Fomitopsidaceae	CT, ME, PA	Sw
*Antrodia mappa* (Overh. & J. Lowe) Miettinen & Vlasák	Fomitopsidaceae	CT, ME, PA	Sw
*Apioperdon pyriforme* (Schaeff.) Vizzini	Lycoperdaceae	AG, CL, CT, EN, ME, PA, RG, SR, TP	Sw
*Armillaria cepistipes* Velen.	Physalacriaceae	CT, ME	Pn
*Armillaria gallica* Marxm. & Romagn.	Physalacriaceae	CT, ME, PA	Pn
*Armillaria lutea* Gillet	Physalacriaceae	CT, ME, PA	Pn
*Armillaria mellea* (Vahl) P. Kumm.	Physalacriaceae	AG, CL, CT, EN, ME, PA, RG, SR, TP	Pn
*Arrhenia baeospora* (Singer) Redhead, Lutzoni, Moncalvo & Vilgalys	Hygrophoraceae	AG, CL, CT, EN, ME, PA, RG, SR, TP	Sm
*Arrhenia elegans* (Pers.) Redhead, Lutzoni, Moncalvo & Vilgalys	Hygrophoraceae	AG, CL, CT, EN, ME, PA, RG, SR, TP	Sm
*Arrhenia lilacinicolor* (Bon) P.-A. Moreau & Courtec.	Hygrophoraceae	CT, ME, PA	Sm
*Arrhenia rickenii* (Hora) Watling	Hygrophoraceae	CT, ME, PA, SR, TP	Sm
*Arrhenia spathulata* (Fr.) Redhead	Hygrophoraceae	CT, PA, RG, SR	Sm
*Artomyces pyxidatus* (Pers.) Jülich	Auriscalpiaceae	CT, ME, PA	Sw
*Ascobolus furfuraceus* Pers.	Ascobolaceae	AG, CL, CT, EN, ME, PA, RG, SR, TP	Sd
*Ascotremella faginea* (Peck) Seaver	Gelatinodiscaceae	CT, ME, PA	Sw
*Aspidella solitaria* (Bull.) E.-J. Gilbert	Amanitaceae	CL, CT, ME, PA	Em
*Aspropaxillus candidus* (Bres.) M.M. Moser	Tricholomataceae	CT, ME, PA	Sl
*Aspropaxillus giganteus* (Sowerby) Kühner & Maire	Tricholomataceae	CT, ME, PA	Sl
*Aspropaxillus lepistoides* (Maire) Kühner & Maire	Tricholomataceae	CT, ME	Sl
*Asterophora lycoperdoides* (Bull.) Ditmar	Lyophyllaceae	CT, ME	Pm
*Asterophora parasitica* (Bull.) Singer	Lyophyllaceae	CT, ME, PA	Pm
*Asterostroma cervicolor* (Berk. & M.A. Curtis) Massee	Peniophoraceae	ME, PA, TP	Sw
*Asterostroma gaillardii* Pat.,	Peniophoraceae	ME, PA, TP	Sw
*Astraeus hygrometricus* (Pers.) Morgan	Diplocystidiaceae	PA, RG, SR, TP	St
*Athelia acrospora* Jülich	Atheliaceae	ME, PA, TP	Sw
*Atheliachaete galactites* (Bourdot & Galzin) Ţura, Zmitr., Wasser & Spirin	Phanerochaetaceae	PA, TP	Sw
*Atheniella flavoalba* (Fr.) Redhead, Moncalvo, Vilgalys, Desjardin & B.A. Perry	Mycenaceae	PA, ME, TP	Sl
*Atractosporocybe inornata* (Sowerby) P. Alvarado, G. Moreno & Vizzini	Incertae Sedis	CT, ME, PA, TP	Sl
*Aureoboletus gentilis* (Quél.) Pouzar	Boletaceae	CT, ME, PA, TP	Em
*Aureoboletus moravicus* (Vaček) Klofac	Boletaceae	CT, ME, PA	Em
*Auricularia auricula-judae* (Bull.) Quél.	Auriculariaceae	AG, CL, CT, EN, ME, PA, RG, SR, TP	Sw
*Auricularia mesenterica* (Dicks.) Pers.	Auriculariaceae	AG, CL, CT, EN, ME, PA, RG, SR, TP	Sw
*Auriscalpium vulgare* Gray	Auriscalpiaceae	AG, CL, CT, EN, ME, PA, RG, SR, TP	Sc
*Australohydnum dregeanum* (Berk.) Hjortstam & Ryvarden	Irpicaceae	PA, TP	Sw
*Baeospora myosura* (Fr.) Singer	Incertae Sedis	AG, CL, CT, EN, ME, PA, RG, SR, TP	Sc
*Balsamia platyspora* Berk.	Helvellaceae	CT, ME	Em
*Balsamia vulgaris* Vittad.	Helvellaceae	CT, PA, SR	Em
*Battarrea phalloides* (Dicks.) Pers.	Agaricaceae	CL, CT, PA	St
*Bertia moriformis* (Tode) De Not.	Bertiaceae	AG, CL, CT, EN, ME, PA, RG, SR, TP	Sw
*Biscogniauxia mediterranea* (De Not.) Kuntze	Graphostromataceae	CT, ME, PA	Pn
*Biscogniauxia nummularia* (Bull.) Kuntze	Graphostromataceae	CT, ME, PA	Pn
*Bjerkandera adusta* (Willd.) P. Karst.	Phanerochaetaceae	AG, CL, CT, EN, ME, PA, RG, SR, TP	Pn
*Bolbitius titubans* (Bull.) Fr.	Bolbitiaceae	AG, CT, ME, PA	Sd
*Boletopsis grisea* (Peck) Bondartsev & Singer	Bankeraceae	CT, ME, PA	Em
*Boletopsis leucomelaena* (Pers.) Fayod	Bankeraceae	CT, ME, PA	Em
*Boletus aereus* Bull.	Boletaceae	AG, CT, CL, EN, ME, PA, RG, SR, TP	Em
*Boletus edulis* Bull.	Boletaceae	AG, CT, CL, EN, ME, PA, RG, SR, TP	Em
*Boletus pinophilus* Pilát & Dermek	Boletaceae	CT, ME, PA	Em
*Boletus reticulatus* Schaeff.	Boletaceae	AG, CT, CL, EN, ME, PA, RG, SR, TP	Em
*Boletus rhodopurpureus* Smotl.	Boletaceae	AG, CT, CL, EN, ME, PA, RG, SR, TP	Em
*Boletus speciosus* Frost	Boletaceae	CT, ME, PA	Em
*Bonomyces sinopicus* (Fr.) Vizzini	Pseudoclitocybaceae	AG, CT, ME, PA	Sl
*Botryobasidium isabellinum* (Fr.) D.P. Rogers	Botryobasidiaceae	ME, PA, TP	Sw
*Botryobasidium pruinatum* (Bres.) J. Erikss.	Botryobasidiaceae	ME, PA, TP	Sw
*Bovista aestivalis* (Bonord.) Demoulin	Lycoperdaceae	AG, PA, TP	Sl
*Bovista nigrescens* Pers.	Lycoperdaceae	CT, ME, PA, TP	Sl
*Bovista plumbea* Pers.	Lycoperdaceae	AG, ME, PA, TP	Sl
*Bovista tomentosa* (Vittad.) De Toni	Lycoperdaceae	CT, ME	Sl
*Bovistella utriformis* (Bull.) Demoulin & Rebriev	Lycoperdaceae	AG, CT, CL, EN, ME, PA, RG, SR, TP	Sl
*Britzelmayria multipedata* (Peck) D. Wächt. & A. Melzer	Psathyrellaceae	AG, ME, PA, TP	Sw
*Brunneoporus malicola* (Berk. & M.A. Curtis) Audet	Fomitopsidaceae	CT, ME, PA	Sw
*Buchwaldoboletus hemichrysus* (Berk. & M.A. Curtis) Pilát	Boletaceae	CT, ME, PA, TP	Sw
*Buchwaldoboletus lignicola* (Kallenb.) Pilát	Boletaceae	CT, PA, TP	Sw
*Bulgaria inquinans* (Pers.) Fr.	Phacidiaceae	AG, CT, CL, EN, ME, PA, RG, SR, TP	Sw
*Butyriboletus appendiculatus* (Schaeff.) D. Arora & J.L. Frank	Boletaceae	AG, CT, CL, EN, ME, PA, RG, SR, TP	Em
*Butyriboletus fechtneri* (Velen.) D. Arora & J.L. Frank	Boletaceae	AG, CT, CL, EN, ME, PA, RG, SR, TP	Em
*Butyriboletus pseudoregius* (Heinr. Huber) D. Arora & J.L. Frank	Boletaceae	AG, CT, CL, EN, ME, PA, RG, SR, TP	Em
*Butyriboletus regius* (Krombh.) D. Arora & J.L. Frank	Boletaceae	AG, CL, CT, EN, ME, PA, RG, SR, TP	Em
*Byssocorticium atrovirens* (Fr.) Bondartsev & Singer	Atheliaceae	ME, PA, TP	Sw
*Byssomerulius corium* (Pers.) Parmasto	Irpicaceae	AG, CL, CT, EN, ME, PA, RG, SR, TP	Sw
*Byssomerulius hirtellus* (Burt) Parmasto	Irpicaceae	ME, PA, TP	Sw
*Byssonectria deformis* (P. Karst.) U. Lindem. & M. Vega	Pyronemataceae	ME, PA, TP	Sbg
*Callistosporium luteo-olivaceum* (Berk. & M.A. Curtis) Singer	Callistosporiaceae	CT, ME, PA	Sw
*Caloboletus calopus* (Pers.) Vizzini	Boletaceae	AG, CL, CT, EN, ME, PA, RG, SR, TP	Em
*Caloboletus radicans* (Pers.) Vizzini	Boletaceae	AG, CL, CT, EN, ME, PA, RG, SR, TP	Em
*Calocera cornea* (Batsch) Fr.	Dacrymycetaceae	AG, CL, CT, EN, ME, PA, RG, SR, TP	Sw
*Calocera viscosa* (Pers.) Fr.	Dacrymycetaceae	AG, CL, CT, EN, ME, PA, RG, SR, TP	Sw
*Calocybe chrysenteron* (Bull.) Singer	Lyophyllaceae	AG, CL, CT, EN, ME, PA, RG, SR, TP	St
*Calocybe favrei* (R. Haller Aar. & R. Haller Suhr) Bon	Lyophyllaceae	CT, ME	St
*Calocybe gambosa* (Fr.) Donk	Lyophyllaceae	AG, CL, CT, EN, ME, PA, RG, SR, TP	St
*Calocybe gangraenosa* (Fr.) V. Hofst., Moncalvo, Redhead & Vilgalys	Lyophyllaceae	AG, ME, PA	St
*Calocybe hypoxantha* (Joss. & Riousset) Bon	Lyophyllaceae	CT, ME	St
*Calocybe ionides* (Bull.) Donk	Lyophyllaceae	CT, ME, PA	St
*Calocybe littoralis* Ballero & Contu	Lyophyllaceae	CT, ME, PA	St
*Calocybe onychina* (Fr.) Donk	Lyophyllaceae	CT, ME, PA	St
*Calonarius callochrous* (Pers.) Niskanen & Liimat.	Cortinariaceae	CT, ME	Em
*Calonarius flavovirens* (Rob. Henry) Niskanen & Liimat.	Cortinariaceae	CT, ME	Em
*Calonarius frondosophilus* (Bidaud) Niskanen & Liimat.	Cortinariaceae	CT, ME	Em
*Calonarius olearioides* (Rob. Henry) Niskanen & Liimat.	Cortinariaceae	CT, ME	Em
*Calongea prieguensis* (Mor.-Arr., J. Gómez & Calonge) Healy, Bonito & Trappe	Pezizaceae	ME	UNK
*Calvatia candida* (Rostk.) Hollós	Lycoperdaceae	CT, ME	St
*Calvatia cyathiformis* (Bosc) Morgan	Lycoperdaceae	CT, ME, PA, TP	St
*Calvatia gigantea* (Batsch) Lloyd	Lycoperdaceae	CT, ME, PA	St
*Calycina citrina* (Hedw.) Gray	Pezizellaceae	AG, CL, CT, EN, ME, PA, RG, SR, TP	Sw
*Calycina herbarum* (Pers.) Gray	Pezizellaceae	CT, ME, PA	Sw
*Candolleomyces candolleanus* (Fr.) D. Wächt. & A. Melzer	Psathyrellaceae	CT, ME	Sw
*Candolleomyces leucotephrus* (Berk. & Broome) D. Wächt. & A. Melzer	Psathyrellaceae	CT, ME	Sw
*Cantharellula umbonata* (J.F. Gmel.) Singer	Hygrophoraceae	CT, ME	Em
*Cantharellus alborufescens* (Malençon) Papetti & S. Alberti	Hydnaceae	CT, ME, PA	Em
*Cantharellus amethysteus* (Quél.) Sacc.	Hydnaceae	CT, ME, PA	Em
*Cantharellus cibarius* Fr. var. *cibarius*	Hydnaceae	AG, CL, CT, EN, ME, PA, RG, SR, TP	Em
*Cantharellus cinereus* (Pers.) Fr.	Hydnaceae	AG, CL, CT, EN, ME, PA, RG, SR, TP	Em
*Cantharellus ferruginascens* P.D. Orton	Hydnaceae	CT, ME, TP	Em
*Cantharellus friesii* Quél.	Hydnaceae	CT, ME, PA	Em
*Cantharellus lilacinopruinatus* Hermitte, Eyssart. & Poumarat	Hydnaceae	CT, ME	Em
*Cantharellus melanoxeros* Desm.	Hydnaceae	CT, ME, PA, TP	Em
*Cantharellus pallens* Pilát	Hydnaceae	CT, ME, PA	Em
*Cellulariella warnieri* (Durieu & Mont.) Zmitr. & Malyshev	Polyporaceae	AG, CL, CT, EN, ME, PA, RG, SR, TP	Sw
*Ceraceomyces tessulatus* (Cooke) Jülich	Amylocorticiaceae	CT, ME, PA, TP	Sw
*Ceriporia alachuana* (Murrill) Hallenb.	Irpicaceae	CT, ME, PA, TP	Sw
*Ceriporia aurantiocarnescens* (Henn.) M. Pieri & B. Rivoire	Irpicaceae	CT, ME, PA, TP	Sw
*Ceriporia excelsa* S. Lundell ex Parmasto	Irpicaceae	CT, ME, PA	Sw
*Ceriporia griseoviolascens* M. Pieri & B. Rivoire	Irpicaceae	CT, ME, PA, TP	Sw
*Ceriporia purpurea* (Fr.) Donk	Irpicaceae	CT, ME, PA, TP	Sw
*Ceriporiopsis mucida* (Pers.) Gilb. & Ryvarden	Meruliaceae	CT, ME, PA, TP	Sw
*Cerioporus leptocephalus* (Jacq.) Zmitr.	Polyporaceae	CT, ME	Sw
*Cerioporus meridionalis* (A. David) Zmitr. & Kovalenko	Polyporaceae	PA, TP	Sw
*Cerioporus squamosus* (Huds.) Quél.	Polyporaceae	CT, ME, PA	Sw
*Cerioporus varius* (Pers.) Zmitr. & Kovalenko	Polyporaceae	CT, ME, PA	Sw
*Cerrena unicolor* (Bull.) Murrill	Cerrenaceae	AG, CL, CT, EN, ME, PA, RG, SR, TP	Sw
*Chalciporus piperatus* (Bull.) Bataille	Boletaceae	AG, CL, CT, EN, ME, PA, RG, SR, TP	Em
*Chalciporus rubinus* (W.G. Sm.) Singer	Boletaceae	CT, ME	Em
*Chamaemyces fracidus* (Fr.) Donk	Agaricaceae	CT, ME	Sl
*Cheilymenia aurantiacorubra* K.S. Thind & S.C. Kaushal	Pyronemataceae	AG, CL, CT, EN, ME, PA, RG, SR, TP	Sd
*Cheilymenia fimicola* (Bagl.) Dennis	Pyronemataceae	AG, CL, CT, EN, ME, PA, RG, SR, TP	Sd
*Cheilymenia granulata* (Bull.) J. Moravec	Pyronemataceae	AG, CL, CT, EN, ME, PA, RG, SR, TP	Sd
*Cheilymenia pulcherrima* (P. Crouan & H. Crouan) Boud.	Pyronemataceae	AG, CL, CT, EN, ME, PA, RG, SR, TP	Sd
*Cheilymenia raripila* (W. Phillips) Dennis	Pyronemataceae	AG, CL, CT, EN, ME, PA, RG, SR, TP	Sd
*Cheilymenia stercorea* (Pers.) Boud.	Pyronemataceae	AG, CL, CT, EN, ME, PA, RG, SR, TP	Sd
*Chlorociboria aeruginascens* (Nyl.) Kanouse ex C.S. Ramamurthi, Korf & L.R. Batra	Chlorociboriaceae	AG, CL, CT, EN, ME, PA, RG, SR, TP	St
*Chlorophyllum brunneum* (Farl. & Burt) Vellinga	Agaricaceae	CT, ME, PA	St
*Chlorophyllum molybdites* (G. Mey.) Massee	Agaricaceae	CT, ME, PA, TP	St
*Chlorophyllum rhacodes* (Vittad.) Vellinga	Agaricaceae	AG, CL, CT, EN, ME, PA, RG, SR, TP	St
*Choiromyces meandriformis* Vittad.	Tuberaceae	CT, ME, PA	UNK
*Chondrostereum purpureum* (Pers.) Pouzar	Cyphellaceae	AG, CL, CT, EN, ME, PA, RG, SR, TP	Sw
*Chroogomphus fulmineus* (R. Heim) Courtec.	Gomphidiaceae	AG, CT, ME, PA, TP	Em
*Chroogomphus helveticus* (Singer) M.M. Moser	Gomphidiaceae	AG, CT, ME, PA, TP	Em
*Chroogomphus rutilus* (Schaeff.) O.K. Mill.	Gomphidiaceae	AG, CL, CT, EN, ME, PA, RG, SR, TP	Em
*Ciboria amentacea* (Balb.) Fuckel	Sclerotiniaceae	AG, CL, CT, EN, ME, PA, RG, SR, TP	Sw
*Ciboria batschiana* (Zopf) N.F. Buchw.	Sclerotiniaceae	AG, CL, CT, EN, ME, PA, RG, SR, TP	Sw
*Ciboria polygoni-vivipari* Eckblad	Sclerotiniaceae	CT, ME, PA, RG, TP	Sw
*Cinereomyces lindbladii* (Berk.) Jülich	Gelatoporiaceae	CT, ME, PA, TP	Sw
*Clathrus archeri* (Berk.) Dring	Phallaceae	CT, ME	Sl
*Clathrus ruber* P. Micheli ex Pers.	Phallaceae	AG, CL, CT, EN, ME, PA, RG, SR, TP	Sl
*Clavaria acuta* Sowerby	Clavariaceae	AG, CL, CT, EN, ME, PA, RG, SR, TP	Sl
*Clavaria atrofusca* Velen.	Clavariaceae	AG, CT, ME, PA	Sl
*Clavaria fragilis* Holmsk.	Clavariaceae	AG, CT, ME, PA	Sl
*Clavaria fumosa* Pers.	Clavariaceae	AG, CL, CT, EN, ME, PA, RG, SR, TP	Sl
*Clavariadelphus flavoimmaturus* R.H. Petersen	Clavariadelphaceae	AG, CL, CT, EN, ME, PA, RG, SR, TP	Sl
*Clavariadelphus pistillaris* (L.) Donk	Clavariadelphaceae	AG, CL, CT, EN, ME, PA, RG, SR, TP	Sl
*Clavariadelphus truncatus* Donk	Clavariadelphaceae	AG, CL, CT, EN, ME, PA, RG, SR, TP	Sl
*Clavulina amethystina* (Bull.) Donk	Hydnaceae	AG, CL, CT, EN, ME, PA, RG, SR, TP	Em
*Clavulina cinerea* (Bull.) J. Schröt.	Hydnaceae	AG, CL, CT, EN, ME, PA, RG, SR, TP	Em
*Clavulina coralloides* (L.) J. Schröt.	Hydnaceae	CT, ME, PA	Em
*Clavulina cristata* (Holmsk.) J. Schröt. var. *cristata*	Hydnaceae	AG, CL, CT, EN, ME, PA, RG, SR, TP	Em
*Clavulina rugosa* (Bull.) J. Schröt.	Hydnaceae	AG, CL, CT, EN, ME, PA, RG, SR, TP	Em
*Clavulinopsis corniculata* (Schaeff.) Corner	Clavariaceae	AG, CL, CT, EN, ME, PA, RG, SR, TP	Em
*Clavulinopsis fusiformis* (Sowerby) Corner	Clavariaceae	AG, CL, CT, EN, ME, PA, RG, SR, TP	Em
*Clavulinopsis laeticolor* (Berk. & M.A. Curtis) R.H. Petersen	Clavariaceae	CT, ME, PA, TP	Em
*Clitocella ammophila* (Malençon) Consiglio	Entolomataceae	CL, CT, PA, RG	Em
*Clitocella popinalis* (Fr.) Kluting, T.J. Baroni & Bergemann	Entolomataceae	CL, CT, ME, PA, RG, TP	Em
*Clitocybe agrestis* Harmaja	Incertae Sedis	CT, ME, PA	Sl
*Clitocybe amarescens* Harmaja	Incertae Sedis	CL, CT, ME, PA, RG, TP	Sl
*Clitocybe concava* (Scop.) Gillet	Incertae Sedis	AG, CL, CT, EN, ME, PA, RG, SR, TP	Sl
*Clitocybe cistophila* Bon & Contu	Incertae Sedis	CT, ME	Sl
*Clitocybe costata* Kühner & Romagn.	Incertae Sedis	AG, PA, TP	Sl
*Clitocybe dealbata* (Sowerby) P. Kumm.	Incertae Sedis	AG, CL, CT, EN, ME, PA, RG, SR, TP	Sl
*Clitocybe diatreta* (Fr.) P. Kumm.	Incertae Sedis	CT, ME	Sl
*Clitocybe fasciculata* H.E. Bigelow & A.H. Sm.	Incertae Sedis	CT, ME, PA	Sl
*Clitocybe fragrans* (With.) P. Kumm.	Incertae Sedis	AG, CL, CT, EN, ME, PA, RG, SR, TP	Sl
*Clitocybe hydrogramma* (Bull.) P. Kumm.	Incertae Sedis	CT, ME, PA	Sl
*Clitocybe infundibuliformis* (Schaeff.) Quél.	Incertae Sedis	AG, CL, CT, EN, ME, PA, RG, SR, TP	Sl
*Clitocybe leucodiatreta* Bon	Incertae Sedis	CL, PA, RG, SR	Sl
*Clitocybe metachroa* (Fr.) P. Kumm.	Incertae Sedis	CT, ME	Sl
*Clitocybe nebularis* (Batsch) P. Kumm.	Incertae Sedis	CT, ME, PA	Sl
*Clitocybe obsoleta* (Batsch) Quél.	Incertae Sedis	AG, CL, PA, TP	Sl
*Clitocybe odora* (Bull.) P. Kumm.	Incertae Sedis	AG, CL, CT, EN, ME, PA, RG, SR, TP	Sl
*Clitocybe phaeophthalma* (Pers.) Kuyper	Incertae Sedis	AG, CL, CT, EN, ME, PA, RG, SR, TP	Sl
*Clitocybe phyllophila* (Pers.) P. Kumm.	Incertae Sedis	CL, CT, ME, PA, TP	Sl
*Clitocybe rivulosa* (Pers.) P. Kumm.	Incertae Sedis	CL, CT, PA, RG, SR	Sl
*Clitocybe subspadicea* (J.E. Lange) Bon & Chevassut	Incertae Sedis	CT, ME	Sl
*Clitocybe truncicola* (Peck) Sacc.	Incertae Sedis	CT, ME, PA	Sl
*Clitocybe umbilicata* P. Kumm.	Incertae Sedis	AG, PA, TP	Sl
*Clitocybe vibecina* (Fr.) Quél.	Incertae Sedis	AG, CL, CT, EN, ME, PA, RG, SR, TP	Sl
*Clitopaxillus alexandri* (Gillet) G. Moreno, Vizzini, Consiglio & P. Alvarado	Pseudoclitocybaceae	AG, CL, CT, EN, ME, PA, RG, SR, TP	Sl
*Clitopilus cystidiatus* Hauskn. & Noordel.	Entolomataceae	CT, ME	Em
*Clitopilus geminus* (Paulet) Noordel. & Co-David	Entolomataceae	AG, CT, ME, PA, SR, TP	Em
*Clitopilus prunulus* (Scop.) P. Kumm.	Entolomataceae	AG, CL, CT, EN, ME, PA, RG, SR, TP	Em
*Chlorophyllum agaricoides* (Czern.) Vellinga	Incertae Sedis	CT, ME	Sl
*Collybia butyracea* (Bull.) P. Kumm.	Omphalotaceae	AG, CL, CT, EN, ME, PA, RG, SR, TP	Sl
*Collybia cirrhata* (Schumach.) Quél.	Incertae Sedis	CT, ME, PA	Sl
*Collybia cookei* (Bres.) J.D. Arnold	Incertae Sedis	AG, CL, CT, EN, ME, PA, RG, SR, TP	Sl
*Collybia nivea* (Mont.) Dennis	Incertae Sedis	CT, ME	Sl
*Collybia ozes* (Fr.) P. Karst.	Incertae Sedis	CT, ME	Sl
*Coltricia perennis* (L.) Murrill	Hymenochaetaceae	AG, CL, CT, EN, ME, PA, RG, SR, TP	Sw
*Colus hirudinosus* Cavalier & Séchier	Phallaceae	PA, TP	Sl
*Coniophora arida* (Fr.) P. Karst.	Coniophoraceae	PA, TP	Sw
*Coniophora olivacea* (Fr.) P. Karst.	Coniophoraceae	AG, CL, CT, EN, ME, PA, RG, SR, TP	Sw
*Coniophora puteana* (Schumach.) P. Karst.	Coniophoraceae	AG, PA	Sw
*Conocybe aporos* Kits van Wav.	Bolbitiaceae	CT, ME	Sl
*Conocybe aurea* (Jul. Schäff.) Hongo	Bolbitiaceae	CT, ME	Sl
*Conocybe blattaria* (Fr.) Kühner	Bolbitiaceae	CT, EN, ME, PA, TP	Sl
*Conocybe dunensis* T.J. Wallace	Bolbitiaceae	CL, PA	Sl
*Conocybe filaris* (Fr.) Kühner	Bolbitiaceae	CT, ME, PA, SR	Sl
*Conocybe intrusa* (Peck) Singer	Bolbitiaceae	CT, ME	Sl
*Conocybe pilosella* (Pers.) Kühner	Bolbitiaceae	CT, ME	Sl
*Conocybe pubescens* (Gillet) Kühner	Bolbitiaceae	AG, CL, CT, EN, ME, PA, RG, SR, TP	Sl
*Conocybe rugosa* (Peck) Watling	Bolbitiaceae	CT, EN, ME, PA, TP	Sl
*Conocybe tenera* (Schaeff.) Fayod	Bolbitiaceae	PA, TP	Sl
*Contumyces vesuvianus* (V. Brig.) Redhead, Moncalvo, Vilgalys & Lutzoni	Rickenellaceae	CT, EN, ME, PA, TP	Sm
*Coprinellus angulatus* (Peck) Redhead, Vilgalys & Moncalvo	Psathyrellaceae	AG, CL, CT, EN, ME, PA, RG, SR, TP	Sl
*Coprinellus disseminatus* (Pers.) J.E. Lange	Psathyrellaceae	AG, CL, CT, EN, ME, PA, RG, SR, TP	Sl
*Coprinellus domesticus* (Bolton) Vilgalys, Hopple & Jacq. Johnson	Psathyrellaceae	AG, CL, CT, EN, ME, PA, RG, SR, TP	Sw
*Coprinellus ephemerus* (Bull.) Redhead, Vilgalys & Moncalvo	Psathyrellaceae	AG, CL, CT, EN, ME, PA, RG, SR, TP	Sw
*Coprinellus impatiens* (Fr.) J.E. Lange	Psathyrellaceae	CT, ME, PA	Sl
*Coprinellus micaceus* (Bull.) Vilgalys, Hopple & Jacq. Johnson	Psathyrellaceae	AG, CL, CT, EN, ME, PA, RG, SR, TP	Sw
*Coprinellus radians* (Desm.) Vilgalys, Hopple & Jacq. Johnson	Psathyrellaceae	CT, ME, PA, TP	Sl
*Coprinellus silvaticus* (Peck) Gminder	Psathyrellaceae	AG, CL, CT, EN, ME, PA, RG, SR, TP	Sl
*Coprinellus truncorum* (Scop.) Redhead, Vilgalys & Moncalvo	Psathyrellaceae	AG, CL, CT, EN, ME, PA, RG, SR, TP	Sw
*Coprinellus xanthothrix* (Romagn.) Vilgalys, Hopple & Jacq. Johnson	Psathyrellaceae	CT, ME	Sl
*Coprinopsis atramentaria* (Bull.) Redhead, Vilgalys & Moncalvo	Psathyrellaceae	AG, CL, CT, EN, ME, PA, RG, SR, TP	Sl
*Coprinopsis kimurae* (Hongo & Aoki) Redhead, Vilgalys & Moncalvo	Psathyrellaceae	CT, ME	Sl
*Coprinopsis lagopus* (Fr.) Redhead, Vilgalys & Moncalvo	Psathyrellaceae	AG, CL, CT, EN, ME, PA, RG, SR, TP	Sl
*Coprinopsis marcescibilis* (Britzelm.) Örstadius & E. Larss.	Psathyrellaceae	CT, ME, PA	Sl
*Coprinopsis melanthina* (Fr.) Örstadius & E. Larss.	Psathyrellaceae	CL, PA	Sl
*Coprinopsis nivea* (Pers.) Redhead, Vilgalys & Moncalvo	Psathyrellaceae	AG, CL, CT, EN, ME, PA, RG, SR, TP	Sd
*Coprinopsis patouillardii* (Quél.) Gminder	Psathyrellaceae	AG, CL, CT, EN, ME, PA, RG, SR, TP	Sl
*Coprinopsis picacea* (Bull.) Redhead, Vilgalys & Moncalvo	Psathyrellaceae	AG, CL, CT, EN, ME, PA, RG, SR, TP	Sl
*Coprinopsis radiata* (Bolton) Redhead, Vilgalys & Moncalvo	Psathyrellaceae	CT, ME, PA	Sl
*Coprinopsis stercorea* (Fr.) Redhead, Vilgalys & Moncalvo	Psathyrellaceae	AG, CL, CT, EN, ME, PA, RG, SR, TP	Sd
*Coprinopsis strossmayeri* (Schulzer) Redhead, Vilgalys & Moncalvo	Psathyrellaceae	PA	Sw
*Coprinus alopecia* Lasch	Agaricaceae	CT, ME, PA	Sl
*Coprinus astroideus* (Fr.) Fr.	Agaricaceae	AG, CL, CT, EN, ME, PA, RG, SR, TP	Sl
*Coprinus comatus* (O.F. Müll.) Pers.	Agaricaceae	AG, CL, PA, TP	Sl
*Coprinus sterquilinus* (Fr.) Fr.	Agaricaceae	AG, CL, CT, EN, ME, PA, RG, SR, TP	Sd
*Coprinus vosoustii* Pilát	Agaricaceae	AG, CL, CT, EN, ME, PA, RG, SR, TP	Sl
*Coprotus lacteus* (Cooke & W. Phillips) Kimbr., Luck-Allen & Cain	Incertae sedis	CT, ME, PA	Sd
*Cordyceps militaris* (L.) Fr.	Cordycipitaceae	TP	Se
*Corinectria fuckeliana* (C. Booth) C. González & P. Chaverri	Nectriaceae	PA, TP	Sw
*Coriolopsis gallica* (Fr.) Ryvarden	Polyporaceae	AG, CL, CT, EN, ME, PA, RG, SR, TP	Sw
*Coronicium gemmiferum* (Bourdot & Galzin) J. Erikss. & Ryvarden	Pterulaceae	PA, TP	Sw
*Corticium roseum* Pers.	Corticiaceae	AG, CL, CT, EN, ME, PA, RG, SR, TP	Sw
*Cortinarius alboviolaceus* (Pers.) Fr.	Cortinariaceae	CT, ME, PA	Em
*Cortinarius anomalus* (Fr.) Fr.	Cortinariaceae	CT, ME, PA	Em
*Cortinarius anthracinus* Fr.	Cortinariaceae	CT, ME, PA	Em
*Cortinarius arcuatorum* Rob. Henry	Cortinariaceae	CT, ME, PA	Em
*Cortinarius atrovirens* Kalchbr.	Cortinariaceae	CT, ME, PA	Em
*Cortinarius balteatocumatilis* Rob. Henry	Cortinariaceae	CT, ME, PA	Em
*Cortinarius bisporiger* Contu	Cortinariaceae	CT, ME, PA	Em
*Cortinarius bulliardii* (Pers.) Fr.	Cortinariaceae	CT, ME, PA	Em
*Cortinarius caerulescens* (Schaeff.) Fr.	Cortinariaceae	CT, ME, PA	Em
*Cortinarius caesiocanescens* M.M. Moser	Cortinariaceae	CT, ME, PA	Em
*Cortinarius caligatus* Malençon	Cortinariaceae	CT, ME, PA	Em
*Cortinarius callochrous* (Pers.) Gray	Cortinariaceae	CT, ME, PA	Em
*Cortinarius camphoratus* (Fr.) Fr.	Cortinariaceae	CT, ME	Em
*Cortinarius cedretorum* Maire	Cortinariaceae	CT, ME, PA, TP	Em
*Cortinarius cinnabarinus* Fr.	Cortinariaceae	CT, ME, PA	Em
*Cortinarius collinitus* (Sowerby) Gray	Cortinariaceae	CT, ME	Em
*Cortinarius cotoneus* Fr.	Cortinariaceae	AG, PA, TP	Em
*Cortinarius croceus* (Schaeff.) Gray	Cortinariaceae	CT, ME	Em
*Cortinarius crystallinus* Fr.	Cortinariaceae	CT, ME	Em
*Cortinarius cyaneus* (Bres.) M.M. Moser	Cortinariaceae	CT, ME, PA	Em
*Cortinarius decipiens* (Pers.) Fr.	Cortinariaceae	CT, ME, PA	Em
*Cortinarius diabolicus* (Fr.) Fr.	Cortinariaceae	CT, ME	Em
*Cortinarius dibaphus* Fr.	Cortinariaceae	CT, ME, PA	Em
*Cortinarius diosmus* Kühner	Cortinariaceae	CT, ME	Em
*Cortinarius duracinus* Fr.	Cortinariaceae	CT, ME	Em
*Cortinarius elatior* Fr.	Cortinariaceae	CT, ME, PA, TP	Em
*Cortinarius elegantior* (Fr.) Fr.	Cortinariaceae	CT, ME, PA	Em
*Cortinarius elegantissimus* Rob. Henry	Cortinariaceae	CT, ME, PA	Em
*Cortinarius fulvo-ochrascens* Rob. Henry	Cortinariaceae	CT, ME, PA	Em
*Cortinarius glaucopus* (Schaeff.) Gray	Cortinariaceae	CT, ME, PA	Em
*Cortinarius herculeus* Malençon	Cortinariaceae	CT, ME, PA	Em
*Cortinarius hinnuleus* Fr.	Cortinariaceae	CT, ME, PA	Em
*Cortinarius hinnuloides* Rob. Henry	Cortinariaceae	CT, ME	Em
*Cortinarius iliopodius* (Bull.) Fr.	Cortinariaceae	CT, ME	Em
*Cortinarius infractus* (Pers.) Fr.	Cortinariaceae	CT, ME, PA	Em
*Cortinarius luridus* Rob. Henry	Cortinariaceae	CT, ME	Em
*Cortinarius meridionalis* Bidaud, Moënne-Locc. & Reumaux	Cortinariaceae	CT, ME, PA	Em
*Cortinarius olearioides* Rob. Henry	Cortinariaceae	CT, ME, PA	Em
*Cortinarius olivascentium* Rob. Henry	Cortinariaceae	CT, ME, PA	Em
*Cortinarius orellanus* Fr.	Cortinariaceae	CT, ME, PA	Em
*Cortinarius phaeophyllus* P. Karst.	Cortinariaceae	CT, ME, PA	Em
*Cortinarius praestans* (Cordier) Gillet	Cortinariaceae	CT, ME, PA	Em
*Cortinarius pulchripes* J. Favre	Cortinariaceae	CT, ME	Em
*Cortinarius purpurascens* Fr. var. *purpurascens*	Cortinariaceae	CT, ME	Em
*Cortinarius rapaceus* Fr.	Cortinariaceae	CT, ME	Em
*Cortinarius rigens* (Pers.) Fr.	Cortinariaceae	CT, ME, PA	Em
*Cortinarius rubellus* Cooke	Cortinariaceae	CT, ME, PA	Em
*Cortinarius rufo-olivaceus* (Pers.) Fr.	Cortinariaceae	CT, ME, PA	Em
*Cortinarius salor* Fr.	Cortinariaceae	CT, ME, PA	Em
*Cortinarius saporatus* Britzelm.	Cortinariaceae	AG, CT, ME, PA	Em
*Cortinarius saturninus* (Fr.) Fr.	Cortinariaceae	CT, ME, PA	Em
*Cortinarius scutulatus* (Fr.) Fr.	Cortinariaceae	CT, ME, PA	Em
*Cortinarius semisanguineus* (Fr.) Gillet	Cortinariaceae	CT, ME, PA	Em
*Cortinarius sodagnitus* Rob. Henry	Cortinariaceae	CT, ME, PA	Em
*Cortinarius splendens* Rob. Henry	Cortinariaceae	CT, ME, PA	Em
*Cortinarius trivialis* J.E. Lange	Cortinariaceae	CT, ME, PA, TP	Em
*Cortinarius variecolor* (Pers.) Fr.	Cortinariaceae	CT, ME, PA	Em
*Cortinarius venetus* (Fr.) Fr.	Cortinariaceae	CT, ME, PA	Em
*Cortinarius violaceus* (L.) Gray	Cortinariaceae	CT, ME, PA	Em
*Cortinarius volvatus* A.H. Sm.	Cortinariaceae	CT, ME	Em
*Cortinarius zinziberatus* (Scop.) Fr.	Cortinariaceae	CT, ME, PA	Em
*Cortinarius xanthophyllus* (Cooke) Rob. Henry	Cortinariaceae	CT, ME, PA	Em
*Craterellus cornucopioides* (L.) Pers.	Hydnaceae	CT, ME, PA, TP	Em
*Craterellus lutescens* (Fr.) Fr.	Hydnaceae	CT, ME, PA	Em
*Craterellus tubaeformis* (Fr.) Quél.	Hydnaceae	CT, ME, PA	Em
*Craterellus undulatus* (Pers.) E. Campo & Papetti	Hydnaceae	CT, ME	Em
*Crepidotus applanatus* (Pers.) P. Kumm.	Crepidotaceae	AG, CL, CT, EN, ME, PA, RG, SR, TP	Sw
*Crepidotus autochthonus* J.E. Lange	Crepidotaceae	CT, ME, PA	Sw
*Crepidotus bresadolae* Pilát	Crepidotaceae	CT, ME	Sw
*Crepidotus calolepis* (Fr.) P. Karst.	Crepidotaceae	CT, ME, PA, TP	Sw
*Crepidotus caspari* Velen.	Crepidotaceae	CT, ME, PA	Sw
*Crepidotus cesatii* (Rabenh.) Sacc.	Crepidotaceae	CT, ME, PA	Sw
*Crepidotus epibryus* (Fr.) Quél.	Crepidotaceae	CT, ME, PA	Sw
*Crepidotus mollis* (Schaeff.) Staude var. *mollis*	Crepidotaceae	CT, ME, PA, TP	Sw
*Crepidotus subverrucisporus* Pilát	Crepidotaceae	CT, ME	Sw
*Crepidotus variabilis* (Pers.) P. Kumm.	Crepidotaceae	AG, PA, SR, TP	Sw
*Crinipellis scabella* (Alb. & Schwein.) Kuyper	Marasmiaceae	AG, CL, CT, ME, PA, TP	Sl
*Cristinia helvetica* (Pers.) Parmasto	Stephanosporaceae	CT, ME, PA, TP	Sw
*Cristinia rhenana* Grosse-Brauckm.	Stephanosporaceae	CT, ME, PA, TP	Sw
*Cryptomarasmius corbariensis* (Roum.) T.S. Jenkinson & Desjardin	Physalacriaceae	AG, CL, CT, EN, ME, PA, RG, SR, TP	Sle
*Crucibulum crucibuliforme* (Scop.) V.S. White	Incertae Sedis	AG, CL, CT, EN, ME, PA, RG, SR, TP	Sw
*Crucibulum laeve* (Huds.) Kambly	Incertae Sedis	AG, CT, ME, PA	Sw
*Cuphophyllus fornicatus* (Fr.) Lodge, Padamsee & Vizzini	Hygrophoraceae	CT, ME, PA, TP	Sl
*Cuphophyllus pratensis* (Pers.) Bon	Hygrophoraceae	CT, ME, PA	Sl
*Cuphophyllus russocoriaceus* (Berk. & T.K. Mill.) Bon,	Hygrophoraceae	AG, CL, CT, EN, ME, PA, RG, SR, TP	Sl
*Cuphophyllus virgineus* (Wulfen) Kovalenko	Hygrophoraceae	AG, CT, ME, PA, TP	Sl
*Cupreoboletus poikilochromus* (Pöder, Cetto & Zuccher.) Simonini, Gelardi & Vizzini	Boletaceae	CT, ME, PA	Em
*Cyanoboletus flavosanguineus* (Lavorato & Simonini) Pierotti	Boletaceae	CT, ME, PA	Em
*Cyanoboletus pulverulentus* (Opat.) Gelardi, Vizzini & Simonini	Boletaceae	CT, ME, PA, TP	Em
*Cyanosporus caesius* (Schrad.) McGinty	Polyporaceae	CT, ME, PA	Sw
*Cyanosporus subcaesius* (A. David) B.K. Cui, L.L. Shen & Y.C. Dai	Polyporaceae	CT, ME, PA, TP	Sw
*Cyathicula cyathoidea* (Bull.) Thüm.	Helotiaceae	CT, ME, PA	Sw
*Cyathus olla* (Batsch) Pers.	Incertae Sedis	AG, CL, CT, EN, ME, PA, RG, SR, TP	Sl
*Cyathus stercoreus* (Schwein.) De Toni	Incertae Sedis	AG, CL, CT, EN, ME, PA, RG, SR, TP	Sd
*Cyathus striatus* (Huds.) Willd.	Incertae Sedis	AG, CL, CT, EN, ME, PA, RG, SR, TP	Sw
*Cyclocybe cylindracea* (DC.) Vizzini & Angelini	Tubariaceae	AG, CL, CT, EN, ME, PA, RG, SR, TP	Sw
*Cylindrobasidium laeve* (Pers.) Chamuris	Physalacriaceae	AG, CL, CT, EN, ME, PA, RG, SR, TP	Sw
*Cylindrobasidium evolvens* (Fr.) Jülich	Physalacriaceae	CT, ME, PA, TP	Sw
*Cystidiopostia hibernica* (Berk. & Broome) B.K. Cui, L.L. Shen & Y.C. Dai	Incertae sedis	CT, ME, PA	Sw
*Cystoderma amianthinum* (Scop.) Fayod	Incertae Sedis	CT, ME, PA, TP	Sm
*Cystoderma carcharias* (Pers.) Fayod	Incertae Sedis	CT, ME, PA	Sm
*Cystodermella adnatifolia* (Peck) Harmaja	Incertae Sedis	CT, ME	Sm
*Cystodermella ambrosii* (Bres.) Smith & Singer	Incertae Sedis	CT, ME, PA	Sm
*Cystodermella cinnabarina* (Alb. & Schwein.) Harmaja	Incertae Sedis	CT, ME, PA	Sm
*Cystodermella granulosa* (Batsch) Harmaja	Incertae Sedis	CT, ME, PA	Sm
*Cystodermella terryi* (Berk. & Broome) Bellù	Incertae Sedis	CT, ME, PA	Sm
*Cystolepiota cystophora* (Malencon) M. Bon	Agaricaceae	CT, ME, PA	Sm
*Cystostereum pini-canadense* (Schwein.) Parmasto	Cystostereaceae	CT, ME, PA, TP	Sw
*Dacrymyces capitatus* Schwein.	Dacrymycetaceae	AG, CL, CT, EN, ME, PA, RG, SR, TP	Sw
*Dacrymyces minor* Peck	Dacrymycetaceae	AG, CL, CT, EN, ME, PA, RG, SR, TP	Sw
*Dacrymyces stillatus* Nees	Dacrymycetaceae	AG, CL, CT, EN, ME, PA, RG, SR, TP	Sw
*Dacrymyces variisporus* McNabb	Dacrymycetaceae	AG, CL, CT, EN, ME, PA, RG, SR, TP	Sw
*Daedalea quercina* (L.) Pers.	Fomitopsidaceae	AG, CL, CT, EN, ME, PA, RG, SR, TP	Sw
*Daedaleopsis confragosa* (Bolton) J. Schröt.	Polyporaceae	AG, CL, CT, EN, ME, PA, RG, SR, TP	Sw
*Daedaleopsis nitida* (Durieu & Mont.) Zmitr. & Malysheva	Polyporaceae	CT, ME, PA	Sw
*Daldinia concentrica* (Bolton) Ces. & De Not.	Hypoxylaceae	AG, CL, CT, PA, RG, TP	Sw
*Daldinia martinii* M. Stadler, Venturella & Wollw.	Hypoxylaceae	PA	Sw
*Daldinia raimundi* M. Stadler, Venturella & Wollw.	Hypoxylaceae	PA	Sw
*Daldinia vernicosa* Ces. & De Not.	Hypoxylaceae	CT, ME	Sw
*Daleomyces phillipsii* (Massee) Seaver	Pezizaceae	CT, ME	Sbg
*Dasyscyphella nivea* (R. Hedw.) Raitv.	Lachnaceae	AG, CL, CT, EN, ME, PA, RG, SR, TP	Sw
*Deconica coprophila* (Bull.) P. Karst.	Strophariaceae	AG, CL, CT, EN, ME, PA, RG, SR, TP	Sd
*Deconica merdaria* (Fr.) Noordel.	Strophariaceae	AG, CL, CT, EN, ME, PA, RG, SR, TP	Sd
*Deconica montana* (Pers.) P.D. Orton	Strophariaceae	CT, ME	Sd
*Delicatula integrella* (Pers.) Fayod	Incertae Sedis	CT, ME	Sm
*Dematophora necatrix* R. Hartig	Xylariaceae	PA, TP	Sw
*Dendrothele acerina* (Pers.) P.A. Lemke	Incertae Sedis	PA, TP	Sw
*Desarmillaria tabescens* (Scop.) R.A. Koch & Aime	Physalacriaceae	AG, CL, CT, EN, ME, PA, RG, SR, TP	Pn
*Descolea alba* (Klotzsch) Kuhar, Nouhra & M.E. Sm.	Bolbitiaceae	PA, RG	UNK
*Dialonectria episphaeria* (Tode) Cooke	Nectriaceae	PA, TP	Sw
*Diatrype disciformis* (Hoffm.) Fr	Diatrypaceae	AG, CL, CT, EN, ME, PA, RG, SR, TP	Sw
*Diatrype stigma* (Hoffm.) Fr.	Diatrypaceae	AG, CL, CT, EN, ME, PA, RG, SR, TP	Sw
*Diatrypella quercina* (Pers.) Cooke	Diatrypaceae	AG, CL, CT, EN, ME, PA, RG, SR, TP	Sw
*Discina ancilis* (Pers.) Sacc.	Discinaceae	CT, ME, PA	Sl
*Discina fastigiata* (Krombh.) Svrček & J. Moravec	Discinaceae	CT, ME	Sl
*Discina melaleuca* Bres.	Discinaceae	CT, ME, PA	Sl
*Disciotis venosa* (Pers.) Arnould,	Morchellaceae	CT, ME, PA	Sl
*Disciseda bovista* (Klotzsch) Henn.	Agaricaceae	AG, CL, CT, EN, ME, PA, RG, SR, TP	Sl
*Dissingia confusa* (Harmaja) K. Hansen & X.H. Wang	Helvellaceae	AG, CL, CT, EN, ME, PA, RG, SR, TP	Sl
*Dissingia leucomelaena* (Pers.) K. Hansen & X.H. Wang	Helvellaceae	CL, PA, RG, TP	Sl
*Dumontinia tuberosa* (Bull.) L.M. Kohn	Sclerotiniaceae	AG, CL, CT, EN, ME, PA, RG, SR, TP	Sl
*Echinoderma asperum* (Pers.) Bon	Agaricaceae	PA, TP	Sl
*Echinoderma echinaceum* (J.E. Lange) Bon	Agaricaceae	CT, ME, PA	Sl
*Efibula tuberculata* (P. Karst.) Zmitr. & Spirin	Irpicaceae	AG, CL, CT, EN, ME, PA, RG, SR, TP	Sw
*Eichleriella deglubens* (Berk. & Broome) Lloyd	Auriculariaceae	ME, PA	Sw
*Elaphomyces anthracinus* Vittad.	Elaphomycetaceae	CT, ME, PA	Em
*Elaphomyces citrinus* Vittad.	Elaphomycetaceae	CT, ME, PA	Em
*Elaphomyces granulatus* Fr.	Elaphomycetaceae	CT, ME, PA	Em
*Elaphomyces maculatus* Vittad.	Elaphomycetaceae	CT, ME, PA	Em
*Elaiopezia polaripapulata* (J. Moravec) Van Vooren	Pezizaceae	PA, TP	Sl
*Entocybe nitida* (Quél.) T.J. Baroni, Largent & V. Hofst.	Entolomataceae	ME, CT	Sl
*Entocybe turbida* (Fr.) T.J. Baroni, V. Hofst. & Largent	Entolomataceae	AG, CL, CT, EN, ME, PA, RG, SR, TP	Sl
*Entoloma aethiops* (Scop.) G. Stev.	Entolomataceae	CT, ME	Sl
*Entoloma aprile* (Britzelm.) Sacc.	Entolomataceae	AG, CL, CT, EN, ME, PA, RG, SR, TP	Sl
*Entoloma asprellum* (Fr.) Fayod	Entolomataceae	AG, CL, CT, EN, ME, PA, RG, SR, TP	Sl
*Entoloma atrocoeruleum* Noordel.	Entolomataceae	AG, CL, CT, EN, ME, PA, RG, SR, TP	Sl
*Entoloma bloxamii* (Berk. & Broome) Sacc.	Entolomataceae	CT, ME, PA	Sl
*Entoloma caeruleum* (P.D. Orton) Noordel.	Entolomataceae	CT, ME, PA	Sl
*Entoloma caesiocinctum* (Kühner) Noordel.	Entolomataceae	CT, ME	Sl
*Entoloma chalybeum* (Pers.) Noordel.	Entolomataceae	CT, ME	Sl
*Entoloma clypeatum* (L.) P. Kumm.	Entolomataceae	AG, CL, CT, EN, ME, PA, RG, SR, TP	Sl
*Entoloma conferendum* (Britzelm.) Noordel.	Entolomataceae	CT, ME, PA	Sl
*Entoloma corvinum* (Kühner) Noordel.	Entolomataceae	CT, ME, PA	Sl
*Entoloma griseoluridum* (Kühner) M.M. Moser	Entolomataceae	CT, PA, TP	Sl
*Entoloma hirtipes* (Schumach.) M.M. Moser	Entolomataceae	CT, ME, PA	Sl
*Entoloma incanum* (Fr.) Hesler	Entolomataceae	CT, ME, PA	Sl
*Entoloma indutoides* (P.D. Orton) Noordel.	Entolomataceae	CT, ME, PA	Sl
*Entoloma juncinum* (Kühner & Romagn.) Noordel.	Entolomataceae	CT, ME	Sl
*Entoloma kuehnerianum* Noordel.	Entolomataceae	CT, ME	Sl
*Entoloma lividoalbum* (Kühner & Romagn.) Kubička	Entolomataceae	PA, TP	Sl
*Entoloma mougeotii* (Fr.) Hesler	Entolomataceae	CT, ME, PA	Sl
*Entoloma papillatum* (Bres.) Dennis	Entolomataceae	CT, ME	Sl
*Entoloma plebeioides* (Schulzer) Noordel.	Entolomataceae	CT, ME, PA	Sl
*Entoloma plebejum* (Kalchbr.) Noordel.	Entolomataceae	CT, ME, PA, RG, TP	Sl
*Entoloma rhodopolium* (Fr.) P. Kumm.	Entolomataceae	AG, CL, CT, EN, ME, PA, RG, SR, TP	Sl
*Entoloma sericeum* Quél.	Entolomataceae	CT, ME, PA	Sl
*Entoloma serrulatum* (Fr.) Hesler	Entolomataceae	CT, ME, PA, TP	Sl
*Entoloma sinuatum* (Bull.) P. Kumm.	Entolomataceae	AG, CL, CT, EN, ME, PA, RG, SR, TP	Sl
*Entoloma undatum* (Gillet) M.M. Moser	Entolomataceae	CT, ME, PA, TP	Sl
*Erastia salmonicolor* (Berk. & M.A. Curtis) Niemelä & Kinnunen	Incertae sedis	ME, PA, TP	Sw
*Eutypella prunastri* (Pers.) Sacc.	Diatrypaceae	ME, PA, TP	Sw
*Exidia glandulosa* (Bull.) Fr.	Auriculariaceae	AG, CL, CT, EN, ME, PA, RG, SR, TP	Sw
*Exidia thuretiana* (Lév.) Fr.	Auriculariaceae	AG, CL, CT, EN, ME, PA, RG, SR, TP	Sw
*Exidiopsis calcea* (Pers.) K. Wells	Auriculariaceae	CT, ME, PA	Sw
*Exidiopsis effusa* Bref.	Auriculariaceae	CT, ME, PA	Sw
*Exidiopsis galzinii* (Bres.) Killerm.	Auriculariaceae	CT, ME, PA	Sw
*Exidiopsis leucophaea* (Bres.) K. Wells	Auriculariaceae	CT, ME, PA	Sw
*Faerberia carbonaria* (Alb. & Schwein.) Pouzar	Incertae Sedis	PA, TP	Sbg
*Fayodia anthracobia* (J. Favre) Knudsen	Incertae Sedis	CT, ME	Sbg
*Fibroporia vaillantii* (DC.) Parmasto	Fibroporiaceae	CT, ME	Sw
*Fibulomyces mutabilis* (Bres.) Jülich	Atheliaceae	PA	Sw
*Fischerula macrospora* Mattir.	Morchellaceae	PA	UNK
*Fistulina hepatica* (Schaeff.) With.	Incertae Sedis	CT, ME, PA	Sw
*Flammulaster carpophilus* (Fr.) Earle ex Vellinga	Tubariaceae	CT, ME, PA	Sw
*Flammulina velutipes* (Curtis) Singer	Physalacriaceae	AG, CL, CT, EN, ME, PA, RG, SR, TP	Sw
*Fomes fomentarius* (L.) Fr.	Polyporaceae	CT, EN, ME, PA	Sw
*Fomitiporia junipericola* Rivoire & Pirlot	Hymenochaetaceae	CT, ME	Sw
*Fomitiporia punctata* (P. Karst.) Murrill	Hymenochaetaceae	CT, ME, PA	Sw
*Fomitiporia pseudopunctata* (A. David, Dequatre & Fiasson) Fiasson	Hymenochaetaceae	ME	Sw
*Fuscoporia contigua* (Pers.) G. Cunn.	Hymenochaetaceae	CT, ME, PA	Sw
*Fuscoporia torulosa* (Pers.) T. Wagner & M. Fisch.	Hymenochaetaceae	AG, CL, CT, EN, ME, PA, RG, SR, TP	Sw
*Fuscopostia leucomallella* (Murrill) B.K. Cui, L.L. Shen & Y.C. Dai	Incertae sedis	PA	Sw
*Galerina graminea* (Velen.) Kühner	Hymenogastraceae	CL, CT, PA	Sl
*Galerina hypnorum* (Schrank) Kühner	Hymenogastraceae	TP	Sl
*Galerina graminea* (Velen.) Kühner	Hymenogastraceae	CT, ME	Sl
*Galerina marginata* (Batsch) Kühner	Hymenogastraceae	CT, ME, PA	Sl
*Galerina stylifera* (G.F. Atk.) A.H. Sm. & Singer	Hymenogastraceae	CT, ME	Sl
*Galerina triscopa* (Fr.) Kühner	Hymenogastraceae	CT, ME	Sl
*Gamundia leucophylla* (Gillet) H.E. Bigelow	Incertae Sedis	CT, ME	Sl
*Ganoderma adspersum* (Schulzer) Donk	Polyporaceae	AG, TP	Sw
*Ganoderma applanatum* (Pers.) Pat.	Polyporaceae	AG, CL, CT, EN, ME, PA, RG, SR, TP	Sw
*Ganoderma australe* (Fr.) Pat.	Polyporaceae	PA	Sw
*Ganoderma lucidum* (Curtis) P. Karst.	Polyporaceae	AG, CL, CT, EN, ME, PA, RG, SR, TP	Sw
*Ganoderma resinaceum* Boud.	Polyporaceae	AG, CL, CT, EN, ME, PA, RG, SR, TP	Sw
*Ganoderma tsugae* Murrill	Polyporaceae	AG, CL, CT, EN, ME, PA, RG, SR, TP	Sw
*Ganoderma valesiacum* Boud.	Polyporaceae	CT, ME, PA	Sw
*Geastrum ambiguum* Mont.	Geastraceae	CT, ME	Sl
*Geastrum campestre* Morgan	Geastraceae	CT, ME, PA	Sl
*Geastrum coronatum* Pers.	Geastraceae	CT, ME, PA	Sl
*Geastrum elegans* Vittad.	Geastraceae	CT, ME, PA, TP	Sl
*Geastrum fimbriatum* Fr.	Geastraceae	CT, ME, PA, RG	Sl
*Geastrum floriforme* Vittad.	Geastraceae	TP	Sl
*Geastrum fornicatum* (Huds.) Hook.	Geastraceae	CT, ME, PA	Sl
*Geastrum kotlabae* V.J. Staněk	Geastraceae	CT, ME	Sl
*Geastrum lageniforme* Vittad.	Geastraceae	CT, ME, PA	Sl
*Geastrum nanum* Pers.	Geastraceae	CT, ME, PA	Sl
*Geastrum pectinatum* Pers.	Geastraceae	CT, ME, PA, TP	Sl
*Geastrum rufescens* Pers.	Geastraceae	CT, ME, PA, TP	Sl
*Geastrum saccatum* Fr.	Geastraceae	CT, ME, PA, TP	Sl
*Geastrum striatum* DC.	Geastraceae	CT, ME, PA, TP	Sl
*Geastrum triplex* Jungh.	Geastraceae	AG, PA	Sl
*Genea fragrans* (Wallr.) Sacc.	Pyronemataceae	EN, ME, PA, SR	UNK
*Genea hispidula* Berk. ex Tul. & C. Tul.	Pyronemataceae	PA	Em
*Genea lespiaultii* Corda	Pyronemataceae	EN, ME, PA, SR	Em
*Genea sphaerica* Tul. & C. Tul.	Pyronemataceae	CT, PA	Em
*Genea verrucosa* Vittad.	Pyronemataceae	AG, ME, PA, TP, SR	Em
*Geoglossum barlae* Boud.	Geoglossaceae	CT, ME, PA, TP	Sl
*Geoglossum cookeanum* Nannf. ex Minter & P.F. Cannon	Geoglossaceae	CT, ME, PA, TP	Sl
*Geoglossum peckianum* Cooke	Geoglossaceae	CT, ME	Sl
*Geoglossum simile* Peck	Geoglossaceae	CT, ME	Sl
*Geoglossum umbratile* Sacc.	Geoglossaceae	PA, TP	Sl
*Geopora arenicola* (Lév.) Kers	Pyronemataceae	PA, RG, SR	Sl
*Geopora arenosa* (Fuckel) S. Ahmad	Pyronemataceae	CL, SR	Sl
*Geopora clausa* (Tul. & C. Tul.) Burds.	Pyronemataceae	CT, ME, PA, TP	Sl
*Geopora cooperi* Harkn.	Pyronemataceae	CT, ME, PA, TP	Sl
*Geopora foliacea* (Schaeff.) S. Ahmad	Pyronemataceae	CT, ME, PA, TP	Sl
*Geopora sumneriana* (Cooke ex W. Phillips) M. Torre	Pyronemataceae	AG, PA, TP	Sl
*Geopyxis carbonaria* (Alb. & Schwein.) Sacc.	Pyronemataceae	AG, PA, TP	Sbg
*Geoscypha ampelina* (Gillet) Van Vooren & Dougoud	Pezizaceae	CT, ME	Sl
*Gliophorus irrigatus* (Pers.) A.M. Ainsw. & P.M. Kirk	Hygrophoraceae	TP	Sl
*Gliophorus psittacinus* (Schaeff.) Herink	Hygrophoraceae	PA, TP	Sl
*Gloeocystidiellum luridum* (Bres.) Boidin	Stereaceae	CT, ME, PA	Sw
*Gloeocystidiellum porosum* (Berk. & M.A. Curtis) Donk	Stereaceae	PA	Sw
*Gloeodontia columbiensis* Burt ex Burds. & Lombard	Incertae Sedis	PA, TP	Sw
*Gloeophyllum abietinum* (Bull.) P. Karst.	Gloeophyllaceae	PA	Sw
*Gloeophyllum sepiarium* (Wulfen) P. Karst.	Gloeophyllaceae	AG, CL, PA, TP	Sw
*Gloiothele citrina* (Pers.) Ginns & G.W. Freeman	Peniophoraceae	PA, TP	Sw
*Gloiothele lactescens* (Berk.) Hjortstam	Peniophoraceae	PA, TP	Sw
*Gomphidius glutinosus* (Schaeff.) Fr.	Gomphidiaceae	AG, CL, CT, EN, ME, PA, RG, SR, TP	Em
*Gomphidius tyrrhenicus* D. Antonini & M. Antonini	Gomphidiaceae	CT, ME	Em
*Grifola frondosa* (Dicks.) Gray	Grifolaceae	CT, ME, PA	Sw
*Gymnopilus hybridus* (Gillet) Maire	Hymenogastraceae	AG	Sw
*Gymnopilus junonius* (Fr.) P.D. Orton	Hymenogastraceae	AG, CT, ME, PA, TP	Sw
*Gymnopilus luteofolius* (Peck) Singer	Hymenogastraceae	CT, ME	Sw
*Gymnopilus penetrans* (Fr.) Murrill	Hymenogastraceae	CL, CT, ME, TP	Sw
*Gymnopilus sapineus* (Fr.) Murrill	Hymenogastraceae	PA, TP	Sw
*Gymnopilus stabilis* (Weinm.) Kühner & Romagn. ex Bon	Hymenogastraceae	CT, ME	Sw
*Gymnopilus suberis* (Maire) Singer	Hymenogastraceae	PA, TP, RG, SR	Sw
*Gymnopus alkalivirens* (Singer) Halling	Omphalotaceae	PA	Sw
*Gymnopus androsaceus* (L.) Della Magg. & Trassin.	Omphalotaceae	CT, ME, PA	Sw
*Gymnopus aquosus* (Bull.) Antonín & Noordel.	Omphalotaceae	PA, TP	Sw
*Gymnopus brassicolens* (Romagn.) Antonín & Noordel.	Omphalotaceae	PA, TP	Sl
*Gymnopus dryophilus* (Bull.) Murrill	Omphalotaceae	AG, CL, CT, EN, ME, PA, RG, SR, TP	Sl
*Gymnopus erythropus* (Pers.) Antonín, Halling & Noordel.	Omphalotaceae	CT, ME, PA	Sl
*Gymnopus foetidus* (Sowerby) P.M. Kirk	Omphalotaceae	CT, ME, PA	Sl
*Gymnopus fusipes* (Bull.) Gray	Omphalotaceae	AG, CL, CT, EN, ME, PA, RG, SR, TP	Sl
*Gymnopus hariolorum* (Bull.) Antonín, Halling & Noordel.	Omphalotaceae	CT, ME, PA, TP	Sl
*Gymnopus impudicus* (Fr.) Antonín, Halling & Noordel.	Omphalotaceae	CT, ME, PA, TP	Sl
*Gymnopus ocior* (Pers.) Antonín & Noordel.	Omphalotaceae	PA, TP	Sl
*Gyrodon lividus* (Bull.) Sacc.	Paxillaceae	AG, CL, CT, EN, ME, PA, RG, SR, TP	Em
*Gyromitra esculenta* (Pers.) Fr.	Discinaceae	AG, CL, CT, EN, ME, PA, RG, SR, TP	Sl
*Gyromitra gigas* (Krombh.) Cooke	Discinaceae	CT, ME	Sl
*Gyromitra infula* (Schaeff.) Quél.	Discinaceae	CT, ME, PA	Sl
*Gyromitra leucoxantha* (Bres.) Harmaja	Discinaceae	CT, ME	Sl
*Gyroporus castaneus* (Bull.) Quél.	Gyroporaceae	AG, CL, CT, EN, ME, PA, RG, SR, TP	Em
*Gyroporus cyanescens* (Bull.) Quél.	Gyroporaceae	AG, CL, CT, EN, ME, PA, RG, SR, TP	Em
*Hapalopilus rutilans* (Pers.) Murrill	Phanerochaetaceae	CT, ME, PA	Sw
*Hebeloma album* Peck	Hymenogastraceae	CT, ME	Sl
*Hebeloma ammophilum* Bohus	Hymenogastraceae	CL, CT, PA	Sl
*Hebeloma anthracophilum* Maire	Hymenogastraceae	AG, CL, CT, EN, ME, PA, RG, SR, TP	Sl
*Hebeloma birrus* (Fr.) Gillet	Hymenogastraceae	CT, ME, PA	Sl
*Hebeloma bulbiferum* Maire	Hymenogastraceae	CT, ME	Sl
*Hebeloma cistophilum* Maire	Hymenogastraceae	PA, TP	Sl
*Hebeloma crustuliniforme* (Bull.) Quél.	Hymenogastraceae	AG, CL, CT, EN, ME, PA, RG, SR, TP	Sl
*Hebeloma fastibile* (Pers.) P. Kumm.	Hymenogastraceae	CT, ME, TP	Sl
*Hebeloma favrei* Romagn. & Quadr.	Hymenogastraceae	CT, ME	Sl
*Hebeloma hiemale* Bres.	Hymenogastraceae	AG, CL, CT, EN, ME, PA, RG, SR, TP	Sl
*Hebeloma laterinum* (Batsch) Vesterh.	Hymenogastraceae	CT, ME, PA, TP	Sl
*Hebeloma leucosarx* P.D. Orton	Hymenogastraceae	CT, ME, PA	Sl
*Hebeloma malenconii* Bellù & Lanzoni	Hymenogastraceae	CT, ME, PA	Sl
*Hebeloma mesophaeum* (Pers.) Quél.	Hymenogastraceae	CT, ME, PA, TP	Sl
*Hebeloma populinum* Romagn.	Hymenogastraceae	CT, ME	Sl
*Hebeloma pusillum* J.E. Lange	Hymenogastraceae	CT, ME	Sl
*Hebeloma radicosum* (Bull.) Ricken	Hymenogastraceae	AG, CL, CT, EN, ME, PA, RG, SR, TP	Sl
*Hebeloma sacchariolens* Quél.	Hymenogastraceae	AG, CL, CT, EN, ME, PA, RG, SR, TP	Sl
*Hebeloma sarcophyllum* (Peck) Sacc.	Hymenogastraceae	PA, TP	Sl
*Hebeloma sinapizans* (Paulet) Gillet	Hymenogastraceae	AG, CL, CT, EN, ME, PA, RG, SR, TP	Sl
*Hebeloma subcaespitosum* Bon	Hymenogastraceae	SR	Sl
*Hebeloma subsaponaceum* P. Karst.	Hymenogastraceae	CT, ME	Sl
*Hebeloma subtortum* P. Karst.	Hymenogastraceae	CT, ME	Sl
*Hebeloma testaceum* Quél.	Hymenogastraceae	CT, ME, PA	Sl
*Hebeloma theobrominum* Quadr.	Hymenogastraceae	CT, ME	Sl
*Hebeloma versipelle* (Fr.) Gillet	Hymenogastraceae	AG, CL, CT, EN, ME, PA, RG, SR, TP	Sl
*Helvella acetabulum* (L.) Quél.	Helvellaceae	AG, CL, PA	Sl
*Helvella atra* J. König	Helvellaceae	PA, TP	Sl
*Helvella costifera* Nannf.	Helvellaceae	PA	Sl
*Helvella crispa* (Scop.) Fr.	Helvellaceae	AG, CL, CT, EN, ME, PA, RG, SR, TP	Sl
*Helvella elastica* Bull.	Helvellaceae	CT, ME, PA, TP	Sl
*Helvella ephippium* Lév.	Helvellaceae	CT, ME, PA	Sl
*Helvella fibrosa* (Wallr.) Korf	Helvellaceae	CT, ME, PA	Sl
*Helvella fusca* Gillet	Helvellaceae	CT, ME, PA	Sl
*Helvella juniperi* M. Filippa & Baiano	Helvellaceae	CT, ME	Sl
*Helvella lacunosa* Afzel.	Helvellaceae	AG, ME, PA, TP	Sl
*Helvella leucopus* Pers.	Helvellaceae	CT, ME	Sl
*Helvella panormitana* Inzenga	Helvellaceae	CT, ME, PA	Sl
*Helvella solitaria* P. Karst.	Helvellaceae	CT, ME, PA, TP	Sl
*Helvella spadicea* Schaeff.	Helvellaceae	CT, ME, PA	Sl
*Hemileccinum depilatum* (Redeuilh) Šutara	Boletaceae	CT, ME, PA	Em
*Hemileccinum impolitum* (Fr.) Šutara	Boletaceae	AG, CT, ME, PA, TP	Em
*Hemimycena cucullata* (Pers.) Singer	Mycenaceae	CT, ME, PA	Sm
*Hemimycena lactea* (Pers.) Singer	Mycenaceae	AG, PA, TP	Sm
*Henningsomyces candidus* (Pers.) Kuntze	Incertae Sedis	PA TP	Sw
*Hericium cirrhatum* (Pers.) Nikol.	Hericiaceae	PA	Sw
*Hericium coralloides* (Scop.) Pers.	Hericiaceae	CT, ME, PA	Sw
*Hericium erinaceus* (Bull.) Pers.	Hericiaceae	ME, PA	Sw
*Heterobasidion annosum* (Fr.) Bref.	Bondarzewiaceae	AG, CL, CT, EN, ME, PA, RG, SR, TP	Sw
*Hohenbuehelia atrocoerulea* (Fr.) Singer	Pleurotaceae	ME, PA	St
*Hohenbuehelia mastrucata* (Fr.) Singer	Pleurotaceae	CT, ME, PA	St
*Hohenbuehelia petaloides* (Bull.) Schulzer	Pleurotaceae	CT, ME, PA, TP	St
*Hohenbuehelia tremula* (Schaeff.) Thorn & G.L. Barron	Pleurotaceae	CT, ME	St
*Hortiboletus engelii* (Hlaváček) Biketova & Wasser	Boletaceae	CT, ME, PA	Em
*Hortiboletus rubellus* (Krombh.) Simonini, Vizzini & Gelardi	Boletaceae	AG, CL, CT, EN, ME, PA, RG, SR, TP	Em
*Humaria hemisphaerica* (F.H. Wigg.) Fuckel	Pyronemataceae	PA, TP	Em
*Hydnangium carneum* Wallr.	Hydnangiaceae	EN, PA, RG, TP	UNK
*Hydnellum aurantiacum* (Batsch) P. Karst.	Bankeraceae	CT, ME	St
*Hydnellum concrescens* (Pers.) Banker	Bankeraceae	PA, TP	St
*Hydnellum ferrugineum* (Fr.) P. Karst.	Bankeraceae	PA, TP	St
*Hydnellum glaucopus* (Maas Geest. & Nannf.) E. Larss., K.H. Larss. & Kõljalg	Bankeraceae	CT, ME, PA	St
*Hydnellum lepidum* (Maas Geest.) E. Larss., K.H. Larss. & Kõljalg	Bankeraceae	CT, ME, PA	St
*Hydnocristella himantia* (Schwein.) R.H. Petersen	Lentariaceae	PA, TP	St
*Hydnocystis bombycina* (Vittad.) Healy & M.E. Sm.	Pyronemataceae	ME, PA	UNK
*Hydnocystis piligera* Tul.	Pyronemataceae	CT, ME, PA	UNK
*Hydnoporia corrugata* (Fr.) K.H. Larss. & Spirin	Hymenochaetaceae	PA, TP	Sw
*Hydnoporia tabacina* (Sowerby) Spirin, Miettinen & K.H. Larss.	Hymenochaetaceae	CT, ME, PA	Sw
*Hydnum albidum* Peck	Hydnaceae	AG, CL, CT, EN, ME, PA, RG, SR, TP	St
*Hydnum amarescens* Quél.	Hydnaceae	AG, CL, CT, EN, ME, PA, RG, SR, TP	St
*Hydnum magnorufescens* Vizzini, Picillo & Contu	Hydnaceae	CT, ME	St
*Hydnum repandum* L.	Hydnaceae	PA, TP	St
*Hydropus floccipes* (Fr.) Singer	Mycenaceae	CT, ME	Sw
*Hygrocybe acutoconica* (Clem.) Singer	Hygrophoraceae	CT, ME, PA	St
*Hygrocybe aurantiosplendens* R. Haller Aar.	Hygrophoraceae	CT, ME, PA	St
*Hygrocybe ceracea* (Sowerby) P. Kumm.	Hygrophoraceae	CT, ME, PA	St
*Hygrocybe cereopallida* (Clémençon) P. Roux & Eyssart.	Hygrophoraceae	CT, ME	St
*Hygrocybe chlorophana* (Fr.) Wünsche	Hygrophoraceae	CT, ME, PA, TP	St
*Hygrocybe citrina* (Rea) J.E. Lange	Hygrophoraceae	CT, ME, PA, TP	St
*Hygrocybe citrinovirens* (J.E. Lange) Jul. Schäff.	Hygrophoraceae	CT, ME, PA, TP	St
*Hygrocybe coccinea* (Schaeff.) P. Kumm.	Hygrophoraceae	AG, PA	St
*Hygrocybe conica* (Schaeff.) P. Kumm.	Hygrophoraceae	AG, CL, CT, EN, ME, PA, RG, SR, TP	St
*Hygrocybe conicoides* (P.D. Orton) P.D. Orton & Watling	Hygrophoraceae	PA, SR	St
*Hygrocybe flavescens* (Kauffman) Singer	Hygrophoraceae	AG, CL, CT, EN, ME, PA, RG, SR, TP	St
*Hygrocybe glutinipes* (J.E. Lange) R. Haller Aar.	Hygrophoraceae	CT, ME, PA	St
*Hygrocybe marchii* (Bres.) F.H. Møller	Hygrophoraceae	CT, ME, PA	St
*Hygrocybe miniata* (Fr.) P. Kumm.	Hygrophoraceae	CT, ME	St
*Hygrocybe mollis* (Berk. & Broome) M.M. Moser	Hygrophoraceae	CT, ME	St
*Hygrocybe mucronella* (Fr.) P. Karst.	Hygrophoraceae	CT, ME, PA, TP	St
*Hygrocybe nigrescens* (Quél.) Kühner	Hygrophoraceae	CT, ME, PA, TP	St
*Hygrocybe punicea* (Fr.) P. Kumm.	Hygrophoraceae	AG, CL, CT, EN, ME, PA, RG, SR, TP	St
*Hygrocybe quieta* (Kühner) Singer	Hygrophoraceae	CT, ME	St
*Hygrocybe spadicea* (Scop.) P. Karst.	Hygrophoraceae	AG, CL, CT, EN, ME, PA, RG, SR, TP	St
*Hygrocybe splendidissima* (P.D. Orton) M.M. Moser	Hygrophoraceae	CT, ME, PA	St
*Hygrocybe subpapillata* Kühner	Hygrophoraceae	CT, ME, PA	St
*Hygrophoropsis aurantiaca* (Wulfen) Maire	Hygrophoropsidaceae	AG, CL, CT, EN, ME, PA, RG, SR, TP	St
*Hygrophorus agathosmus* (Fr.) Fr.	Hygrophoraceae	CT, ME, PA	Em
*Hygrophorus arbustivus* Fr.	Hygrophoraceae	CT, ME, PA	Em
*Hygrophorus atramentosus* (Alb. & Schwein.) H. Haas & R. Haller Aar. ex Bon	Hygrophoraceae	CT, ME, PA	Em
*Hygrophorus calophyllus* P. Karst.	Hygrophoraceae	CT, ME, PA	Em
*Hygrophorus camarophyllus* (Alb. & Schwein.) Dumée, Grandjean & Maire	Hygrophoraceae	AG, CL, CT, EN, ME, PA, RG, SR, TP	Em
*Hygrophorus chrysodon* (Batsch) Fr.	Hygrophoraceae	CT, ME, PA	Em
*Hygrophorus cossus* (Sowerby) Fr.	Hygrophoraceae	CT, ME, PA, TP	Em
*Hygrophorus discoxanthus* (Fr.) Rea	Hygrophoraceae	CT, ME, PA, TP	Em
*Hygrophorus discoideus* (Pers.) Fr	Hygrophoraceae	CT, ME	Em
*Hygrophorus eburneus* (Bull.) Fr.	Hygrophoraceae	ME, PA	Em
*Hygrophorus erubescens* (Fr.) Fr.	Hygrophoraceae	CT, ME, PA	Em
*Hygrophorus fagi* G. Becker & Bon	Hygrophoraceae	CT, ME	Em
*Hygrophorus gliocyclus* Fr.	Hygrophoraceae	CT, ME	Em
*Hygrophorus hyacinthinus* Quél.	Hygrophoraceae	CT, ME	Em
*Hygrophorus hypothejus* (Fr.) Fr.	Hygrophoraceae	CT, ME, PA	Em
*Hygrophorus latitabundus* Britzelm.	Hygrophoraceae	AG, PA, TP	Em
*Hygrophorus leucophaeo-ilicis* Bon & Chevassut	Hygrophoraceae	CT, ME, PA	Em
*Hygrophorus lindtneri* M.M. Moser	Hygrophoraceae	CT, ME, PA	Em
*Hygrophorus melizeus* (Fr.) Fr.	Hygrophoraceae	CT, ME, PA	Em
*Hygrophorus nemoreus* (Pers.) Fr.	Hygrophoraceae	CT, ME, PA	Em
*Hygrophorus olivaceoalbus* (Fr.) Fr.	Hygrophoraceae	AG, CL, CT, EN, ME, PA, RG, SR, TP	Em
*Hygrophorus penarius* Fr.	Hygrophoraceae	PA, TP	Em
*Hygrophorus persoonii* Arnolds	Hygrophoraceae	PA, TP	Em
*Hygrophorus pseudodiscoideus* (Maire) Malençon & Bertault	Hygrophoraceae	CT, ME, PA	Em
*Hygrophorus roseodiscoideus* Bon & Chevassut	Hygrophoraceae	AG, PA	Em
*Hygrophorus russula* (Schaeff. ex Fr.) Kauffman	Hygrophoraceae	AG, CL, CT, EN, ME, PA, RG, SR, TP	Em
*Hygrophorus speciosus* Peck	Hygrophoraceae	CT, ME, PA	Em
*Hymenochaete rubiginosa* (Dicks.) Lév.	Hymenochaetaceae	AG, CL, CT, EN, ME, PA, RG, SR, TP	Sw
*Hymenogaster bulliardii* Vittad.	Hymenogastraceae	CT, ME, PA	Em
*Hymenogaster calosporus* Tul. & C. Tul.	Hymenogastraceae	TP	Em
*Hymenogaster citrinus* Vittad.	Hymenogastraceae	PA	Em
*Hymenogaster griseus* Vittad.	Hymenogastraceae	CT, ME, TP	Em
*Hymenogaster hessei* Soehner	Hymenogastraceae	PA, SR	Em
*Hymenogaster luteus* Vittad.	Hymenogastraceae	PA, SR	Em
*Hymenogaster lycoperdineus* Vittad.	Hymenogastraceae	AG, ME, PA, TP	Em
*Hymenogaster muticus* Berk. & Broome	Hymenogastraceae	AG, TP	Em
*Hymenogaster olivaceus* Vittad.	Hymenogastraceae	AG, CT, ME, PA	Em
*Hymenogaster populetorum* Tul. & C. Tul.	Hymenogastraceae	AG, PA, TP	Em
*Hymenogaster rehsteineri* Bucholtz	Hymenogastraceae	PA, SR	Em
*Hymenogaster thwaitesii* Berk. & Broome	Hymenogastraceae	PA	Em
*Hymenogaster vulgaris* Tul. & C. Tul.	Hymenogastraceae	PA	Em
*Hymenopellis radicata* (Relhan) R.H. Petersen	Physalacriaceae	PA	Sw
*Hymenopellis xeruloides* (Bon) R.H. Petersen	Physalacriaceae	SR	Sw
*Hymenoscyphus calyculus* (Fr.) W. Phillips	Helotiaceae	PA, SR	Sw
*Hymenoscyphus fructigenus* (Bull.) Gray	Helotiaceae	AG, CL, CT, EN, ME, PA, RG, SR, TP	Sw
*Hyphoderma medioburiense* (Burt) Donk	Hyphodermataceae	PA, TP	Sw
*Hyphoderma setigerum* (Fr.) Donk	Hyphodermataceae	PA	Sw
*Hyphodermella corrugata* (Fr.) J. Erikss. & Ryvarden	Phanerochaetaceae	PA, TP	Sw
*Hyphodontia arguta* (Fr.) J. Erikss.	Hyphodontiaceae	PA, TP	Sw
*Hyphodontia pallidula* (Bres.) J. Erikss.	Hyphodontiaceae	PA, TP	Sw
*Hyphodontia quercina* (Pers.) J. Erikss.	Hyphodontiaceae	PA, TP	Sw
*Hyphodontia spathulata* (Schrad.) Parmasto	Hyphodontiaceae	EN	Sw
*Hypholoma capnoides* (Fr.) P. Kumm.	Strophariaceae	PA	Sw
*Hypholoma dispersum* Quél.	Strophariaceae	PA	Sw
*Hypholoma fasciculare* (Huds.) P. Kumm.	Strophariaceae	AG, PA, TP	Sw
*Hypholoma lateritium* (Schaeff.) P. Kumm.	Strophariaceae	CT, ME	Sw
*Hypocrea gelatinosa* (Tode) Fr.	Hypocreaceae	CT, ME	Sw
*Hypoderma ferulae* Lantieri	Rhytismataceae	PA	Sw
*Hypoderma siculum* Lantieri, P.R. Johnst. & Medardi	Rhytismataceae	PA	Sw
*Hypomyces rosellus* (Alb. & Schwein.) Tul.	Hypocreaceae	PA	Sw
*Hypoxylon fragiforme* (Pers.) J. Kickx f.	Hypoxylaceae	PA	Sw
*Hypoxylon fuscum* (Pers.) Fr.	Hypoxylaceae	CT, ME	Sw
*Hypoxylon rubiginosum* (Pers.) Fr.	Hypoxylaceae	PA	Sw
*Hypsizygus tessulatus* (Bull.) Singer	Lyophyllaceae	CT, ME	Sw
*Hypsizygus ulmarius* (Bull.) Redhead	Lyophyllaceae	CT, ME, PA	Sw
*Hysterangium clathroides* Vittad.	Hysterangiaceae	CT, ME	Em
*Hysterangium coriaceum* R. Hesse	Hysterangiaceae	PA	Em
*Hysterangium inflatum* Rodway	Hysterangiaceae	AG, CL, PA, TP	Em
*Hysterangium stoloniferum* Tul. & C. Tul.	Hysterangiaceae	PA	Em
*Hysterium angustatum* Alb. & Schwein.	Hysteriaceae	PA	Sw
*Hysterium pulicare* Pers.	Hysteriaceae	PA	Sw
*Hysterographium fraxini* (Pers.) De Not.	Incertae sedis	PA, TP	Sw
*Ileodictyon gracile* Berk.	Phallaceae	TP	Sl
*Imleria badia* (Fr.) Vizzini	Boletaceae	CT, ME, PA	Em
*Imperator luteocupreus* (Bertéa & Estadès) Assyov, Bellanger et al.	Boletaceae	CT, ME, PA	Em
*Imperator rhodopurpureus* (Smotl.) Assyov, Bellanger, Bertéa et al.	Boletaceae	CT, ME, PA	Em
*Imperator torosus* (Fr.) Assyov, Bellanger, Bertéa et al.	Boletaceae	CT, ME, PA	Em
*Imperator xanthocyaneus* (Ramain) Klofac	Boletaceae	CT, ME	Em
*Infundibulicybe alkaliviolascens* (Bellù) Bellù	Incertae Sedis	CT, ME, PA	Sl
*Infundibulicybe geotropa* (Bull.) Harmaja	Incertae Sedis	AG, CL, CT, EN, ME, PA, RG, SR, TP	Sl
*Infundibulicybe gibba* (Pers.) Harmaja	Incertae Sedis	AG, CL, CT, EN, ME, PA, RG, SR, TP	Sl
*Infundibulicybe gigas* (Harmaja) Harmaja	Incertae Sedis	CT, ME, PA	Sl
*Infundibulicybe meridionalis* (Bon) Pérez-De-Greg.	Incertae Sedis	CT, ME	Sl
*Inocutis tamaricis* (Pat.) Fiasson & Niemelä	Hymenochaetaceae	AG, CL, CT, EN, ME, PA, RG, SR, TP	Sw
*Inocybe amethystina* Kuyper	Inocybaceae	AG, CL, CT, EN, ME, PA, RG, SR, TP	Em
*Inocybe asterospora* Quél.	Inocybaceae	CT, ME, PA	Em
*Inocybe aurantioumbonata* Franchi & M. Marchetti	Inocybaceae	CT, ME	Em
*Inocybe auricoma* (Batsch) Sacc.	Inocybaceae	CT, ME	Em
*Inocybe brunnea* Quél.	Inocybaceae	CT, ME	Em
*Inocybe caesariata* (Fr.) P. Karst.	Inocybaceae	CT, ME, TP	Em
*Inocybe castaneicolor* A. La Rosa, Bizio, Saitta & Tedersoo	Inocybaceae	CL, ME	Em
*Inocybe cincinnata* (Fr.) Quél.	Inocybaceae	CT, ME, PA	Em
*Inocybe curvipes* P. Karst.	Inocybaceae	CT, ME, PA	Em
*Inocybe dulcamara* (Pers.) P. Kumm.	Inocybaceae	PA, SR	Em
*Inocybe dunensis* P.D. Orton	Inocybaceae	SR	Em
*Inocybe eutheles* (Fr.) Quél.	Inocybaceae	TP	Em
*Inocybe flocculosa* Sacc.	Inocybaceae	CT, ME, PA	Em
*Inocybe fraudans* (Britzelm.) Sacc.	Inocybaceae	PA	Em
*Inocybe fuscidula* Velen.	Inocybaceae	PA	Em
*Inocybe geophylla* (Sowerby) P. Kumm.	Inocybaceae	AG, PA, TP	Em
*Inocybe godeyi* Gillet	Inocybaceae	CT, ME	Em
*Inocybe grammopodia* Malençon	Inocybaceae	CT, ME	Em
*Inocybe griseolilacina* J.E. Lange	Inocybaceae	PA	Em
*Inocybe halophila* R. Heim	Inocybaceae	SR	Em
*Inocybe hirtella* Bres.	Inocybaceae	PA	Em
*Inocybe lacera* (Fr.) P. Kumm.	Inocybaceae	PA	Em
*Inocybe mixtilis* (Britzelm.) Sacc.	Inocybaceae	PA	Em
*Inocybe muricellata* Bres.	Inocybaceae	CT, ME	Em
*Inocybe napipes* J.E. Lange	Inocybaceae	TP	Em
*Inocybe nitidiuscula* (Britzelm.) Lapl.	Inocybaceae	CT, ME	Em
*Inocybe obscurobadia* (J. Favre) Grund & D.E. Stuntz	Inocybaceae	AG, CL	Em
*Inocybe ochroalba* Bruyl.	Inocybaceae	PA	Em
*Inocybe olida* Maire	Inocybaceae	CT, ME	Em
*Inocybe phaeodisca* Kühner	Inocybaceae	PA	Em
*Inocybe phaeoleuca* Kühner	Inocybaceae	CT, ME	Em
*Inocybe praetervisa* Quél.	Inocybaceae	AG	Em
*Inocybe pyriodora* (Pers.) P. Kumm.	Inocybaceae	AG, PA, TP	Em
*Inocybe roseipes* Malençon	Inocybaceae	PA	Em
*Inocybe rufuloides* Bon	Inocybaceae	SR	Em
*Inocybe scabelliformis* Malençon	Inocybaceae	CT, ME	Em
*Inocybe sericeopoda* Furrer-Ziogas	Inocybaceae	PA	Em
*Inocybe serotina* Peck	Inocybaceae	CL	Em
*Inocybe sindonia* (Fr.) P. Karst.	Inocybaceae	PA	Em
*Inocybe splendens* R. Heim	Inocybaceae	PA	Em
*Inocybe tenebrosa* Quél.	Inocybaceae	PA	Em
*Inocybe whitei* (Berk. & Broome) Sacc.	Inocybaceae	CL, CT, ME, PA	Em
*Inonotus hispidus* (Bull.) P. Karst.	Hymenochaetaceae	ME, PA	Sw
*Inonotus rickii* (Pat.) D.A. Reid	Hymenochaetaceae	PA	Sw
*Inosperma adaequatum* (Britzelm.) Matheny & Esteve-Rav.	Inocybaceae	CT, ME	Em
*Inosperma bongardii* (Weinm.) Matheny & Esteve-Rav.	Inocybaceae	AG, CT, ME, PA, TP	Em
*Inosperma cervicolor* (Pers.) Matheny & Esteve-Rav.	Inocybaceae	CL, PA, TP	Em
*Inosperma cookei* (Bres.) Matheny & Esteve-Rav.	Inocybaceae	PA	Em
*Inosperma erubescens* (A. Blytt) Matheny & Esteve-Rav.	Inocybaceae	CT, ME, PA	Em
*Inosperma geraniodorum* (J. Favre) Matheny & Esteve-Rav.	Inocybaceae	CT, ME	Em
*Inosperma maculatum* (Boud.) Matheny & Esteve-Rav.	Inocybaceae	CT, ME	Em
*Inosperma pisciodorum* (Donadini & Riousset) Matheny & Esteve-Rav.	Inocybaceae	CT, ME	Em
*Irpiciporus pachyodon* (Pers.) Kotl. & Pouzar	Meruliaceae	CT, ME, PA	Sw
*Jackrogersella cohaerens* (Pers.) L. Wendt, Kuhnert & M. Stadler	Hypoxylaceae	CT, ME, PA	Sw
*Jimgerdemannia flammicorona* (Trappe & Gerd.) Trappe, Desirò, M.E. Sm. et al.	Endogonaceae	ME	Em
*Junghuhnia nitida* (Pers.) Ryvarden	Steccherinaceae	ME, PA	Sw
*Kavinia alboviridis* (Morgan) Gilb. & Budington	Lentariaceae	PA	Sw
*Kretzschmaria deusta* (Hoffm.) P.M.D. Martin	Xylariaceae	PA	Sw
*Kuehneromyces mutabilis* (Schaeff.) Singer & A.H. Sm.	Strophariaceae	AG, CL, CT, EN, ME, PA, RG, SR, TP	Sw
*Kurtia argillacea* (Bres.) Karasiński	Incertae sedis	PA, TP	Sw
*Laccaria amethystina* Cooke	Hydnangiaceae	AG, CL, CT, EN, ME, PA, RG, SR, TP	Em
*Laccaria bicolor* (Maire) P.D. Orton	Hydnangiaceae	AG, CL, CT, EN, ME, PA, RG, SR, TP	Em
*Laccaria bisporigera* Contu & Ballero	Hydnangiaceae	AG, CL, CT, EN, ME, PA, RG, SR, TP	Em
*Laccaria fraterna* (Sacc.) Pegler	Hydnangiaceae	AG, CL, CT, EN, ME, PA, RG, SR, TP	Em
*Laccaria laccata* (Scop.) Cooke	Hydnangiaceae	AG, CL, CT, EN, ME, PA, RG, SR, TP	Em
*Laccaria lateritia* Malençon	Hydnangiaceae	AG, CL, CT, EN, ME, PA, RG, SR, TP	Em
*Laccaria macrocystidiata* (Migl. & Lavorato) Pázmány	Hydnangiaceae	CT, ME, PA, TP	Em
*Laccaria proxima* (Boud.) Pat.	Hydnangiaceae	AG, CL, CT, EN, ME, PA, RG, SR, TP	Em
*Laccaria purpureobadia* D.A. Reid	Hydnangiaceae	CT, ME	Em
*Laccaria tetraspora* Singer	Hydnangiaceae	CT, PA, TP	Em
*Laccariopsis mediterranea* (Pacioni & Lalli) Vizzini	Physalacriaceae	AG, CT, ME, PA, RG, SR, TP	Sl
*Lachnellula subtilissima* (Cooke) Dennis	Lachnaceae	AG, CL, CT, EN, ME, PA, RG, SR, TP	Sw
*Lachnum bicolor* (Bull.) P. Karst.	Lachnaceae	AG, CL, CT, EN, ME, PA, RG, SR, TP	Sw
*Lachnum virgineum* (Batsch) P. Karst.	Lachnaceae	AG, CL, CT, EN, ME, PA, RG, SR, TP	Sw
*Lacrymaria lacrymabunda* (Bull.) Pat.	Psathyrellaceae	AG, CL, CT, EN, ME, PA, RG, SR, TP	Sl
*Lactarius acerrimus* Britzelm.	Russulaceae	AG, CL, CT, EN, ME, PA, RG, SR, TP	Em
*Lactarius acris* (Bolton) Gray	Russulaceae	CT, ME	Em
*Lactarius atlanticus* Bon	Russulaceae	AG, CT, ME, PA, TP	Em
*Lactarius aurantiacus* (Pers.) Gray	Russulaceae	CT, ME, PA	Em
*Lactarius azonites* (Bull.) Fr.	Russulaceae	CT, ME, PA, TP	Em
*Lactarius badiosanguineus* Kühner & Romagn.	Russulaceae	CT, ME, PA, TP	Em
*Lactarius blennius* (Fr.) Fr.	Russulaceae	CT, ME, PA	Em
*Lactarius camphoratus* (Bull.) Fr.	Russulaceae	CT, ME, PA, TP	Em
*Lactarius chrysorrheus* Fr.	Russulaceae	AG, CL, CT, EN, ME, PA, RG, SR, TP	Em
*Lactarius circellatus* Fr.	Russulaceae	CT, ME	Em
*Lactarius cistophilus* Bon & Trimbach	Russulaceae	CT, ME, PA, TP	Em
*Lactarius controversus* Pers.	Russulaceae	CT, ME, PA, TP	Em
*Lactarius cremor* Fr.	Russulaceae	CT, ME	Em
*Lactarius decipiens* Quél.	Russulaceae	CT, ME, PA, TP	Em
*Lactarius deliciosus* (L.) Gray	Russulaceae	AG, CL, CT, EN, ME, PA, RG, SR, TP	Em
*Lactarius fluens* Boud.	Russulaceae	CT, ME, PA, TP	Em
*Lactarius fuliginosus* (Fr.) Fr.	Russulaceae	PA, TP	Em
*Lactarius fulvissimus* Romagn.	Russulaceae	CT, ME, PA	Em
*Lactarius glyciosmus* (Fr.) Fr.	Russulaceae	CT, ME	Em
*Lactarius hepaticus* Plowr.	Russulaceae	CT, ME, PA, TP	Em
*Lactarius ilicis* Sarnari	Russulaceae	CT, ME, PA, TP	Em
*Lactarius lacunarum* Romagn. ex Hora	Russulaceae	CT, ME, PA	Em
*Lactarius mairei* Malençon	Russulaceae	AG, CT, ME, PA, TP	Em
*Lactarius mediterraneensis* Llistos. & Bellù	Russulaceae	ME, PA	Em
*Lactarius pallidus* Pers.	Russulaceae	CT, ME, PA, TP	Em
*Lactarius pterosporus* Romagn.	Russulaceae	CT, ME, PA	Em
*Lactarius pubescens* Fr.	Russulaceae	CT, ME	Em
*Lactarius pyrogalus* (Bull.) Fr.	Russulaceae	CT, ME, PA	Em
*Lactarius quietus* (Fr.) Fr.	Russulaceae	CT, ME, PA	Em
*Lactarius rubrocinctus* Fr.	Russulaceae	CT, ME, PA	Em
*Lactarius rufus* (Scop.) Fr.	Russulaceae	AG, CL, CT, EN, ME, PA, RG, SR, TP	Em
*Lactarius sanguifluus* (Paulet) Fr.	Russulaceae	AG, PA, TP	Em
*Lactarius semisanguifluus* R. Heim & Leclair	Russulaceae	AG, CT, ME, PA, TP	Em
*Lactarius serifluus* (DC.) Fr.	Russulaceae	AG, CT, ME, PA, TP	Em
*Lactarius subdulcis* (Pers.) Gray	Russulaceae	CT, ME, PA	Em
*Lactarius subumbonatus* Lindgr.	Russulaceae	PA, TP	Em
*Lactarius tesquorum* Malençon	Russulaceae	CL, ME, PA, TP	Em
*Lactarius uvidus* (Fr.) Fr.	Russulaceae	AG, CT, ME, PA, TP	Em
*Lactarius violascens* (J. Otto) Fr.	Russulaceae	AG, PA, TP	Em
*Lactarius zonarius* (Bull.) Fr.	Russulaceae	CT, ME, PA, TP	Em
*Lactifluus bertillonii* (Neuhoff ex Z. Schaef.) Verbeken	Russulaceae	CT, ME	Em
*Lactifluus glaucescens* (Crossl.) Verbeken	Russulaceae	CT, ME, PA	Em
*Lactifluus luteolus* (Peck) Verbeken	Russulaceae	CT, ME	Em
*Lactifluus pergamenus* (Sw.) Kuntze	Russulaceae	CT, ME, PA	Em
*Lactifluus piperatus* (L.) Roussel	Russulaceae	AG, CL, CT, EN, ME, PA, RG, SR, TP	Em
*Lactifluus rugatus* (Kühner & Romagn.) Verbeken	Russulaceae	AG, CL, CT, EN, ME, PA, RG, SR, TP	Em
*Lactifluus vellereus* (Fr.) Kuntze	Russulaceae	AG, CL, CT, EN, ME, PA, RG, SR, TP	Em
*Lactifluus volemus* (Fr.) Kuntze	Russulaceae	AG, CL, CT, EN, ME, PA, RG, SR, TP	Em
*Lactocollybia epia* (Berk. & Broome) Pegler	Incertae sedis	CT, ME	Sl
*Laeticutis cristata* (Schaeff.) Audet	Incertae sedis	CT, ME, PA	Sl
*Laetiporus sulphureus* (Bull.) Murrill	Laetiporaceae	AG, PA, RG, SR, TP	Sw
*Lanmaoa fragrans* (Vittad.) Vizzini, Gelardi & Simonini	Boletaceae	CT, ME, PA	Em
*Lamprospora ammophila* (Saut.) Boud.	Pyronemataceae	CT, ME	Sm
*Lamprospora dictydiola* Boud.	Pyronemataceae	RG	Sm
*Lamprospora miniata* De Not.	Pyronemataceae	RG	Sm
*Lanzia echinophila* (Bull.) Korf	Rutstroemiaceae	CT, ME, PA	Scu
*Lasiobolus cuniculi* Velen.	Ascodesmidaceae	PA	Sd
*Laxitextum bicolor* (Pers.) Lentz	Hericiaceae	PA	Sw
*Leccinellum corsicum* (Rolland) Bresinsky & Manfr. Binder	Boletaceae	AG, CL, CT, EN, ME, PA, RG, SR, TP	Em
*Leccinellum crocipodium* (Letell.) Della Magg. & Trassin.	Boletaceae	CT, ME, PA	Em
*Leccinellum griseum* (Quél.) Bresinsky & Manfr. Binder	Boletaceae	CT, ME, PA	Em
*Leccinellum lepidum* (H. Bouchet ex Essette) Bresinsky & Manfr. Binder	Boletaceae	CT, ME, PA, TP	Em
*Leccinellum pseudoscabrum* (Kallenb.) Mikšík	Boletaceae	CT, ME, PA	Em
*Leccinum albostipitatum* den Bakker & Noordel.	Boletaceae	CT, ME	Em
*Leccinum aurantiacum* (Bull.) Gray	Boletaceae	CT, ME, PA	Em
*Leccinum duriusculum* (Schulzer ex Kalchbr.) Singer	Boletaceae	CT, ME, PA	Em
*Leccinum melaneum* (Smotl.) Pilát & Dermek	Boletaceae	CT, ME, PA	Em
*Leccinum scabrum* (Bull.) Gray	Boletaceae	CT, ME, PA	Em
*Leccinum variicolor* Watling	Boletaceae	PA	Em
*Leccinum versipelle* (Fr. & Hök) Snell	Boletaceae	AG, CL, CT, EN, ME, PA, RG, SR, TP	Em
*Leccinum vulpinum* Watling	Boletaceae	PA	Em
*Legaliana badia* (Pers.) Van Vooren	Pezizaceae	PA	St
*Legaliana badiofuscoides* (Donadini) Van Vooren	Pezizaceae	PA	Sw
*Lentinellus cochleatus* (Pers.) P. Karst.	Auriscalpiaceae	CT, ME, PA	Sw
*Lentinellus flabelliformis* (Bolton) S. Ito	Auriscalpiaceae	CT, ME, PA	Sw
*Lentinellus micheneri* (Berk. & M.A. Curtis) Pegler	Auriscalpiaceae	CT, ME, PA, TP	Sw
*Lentinellus ursinus* (Fr.) Kühner	Auriscalpiaceae	CT, ME	Sw
*Lentinus arcularius* (Batsch) Zmitr.	Polyporaceae	ME, PA, TP	Sw
*Lentinus brumalis* (Pers.) Zmitr.	Polyporaceae	CT, ME, PA	Sw
*Lentinus scleropus* (Pers.) Fr.	Polyporaceae	CT, ME	Sw
*Lentinus substrictus* (Bolton) Zmitr. & Kovalenko	Polyporaceae	CT, ME, PA	Sw
*Lentinus tigrinus* (Bull.) Fr.	Polyporaceae	CT, ME, PA	Sw
*Lenzites betulinus* (L.) Fr.	Polyporaceae	CT, PA	Sw
*Leotia lubrica* (Scop.) Pers.	Leotiaceae	AG, CL, CT, EN, ME, PA, RG, SR, TP	St
*Lepiota brunneoincarnata* Chodat & C. Martín	Agaricaceae	AG, CL, CT, EN, ME, PA, RG, SR, TP	Sl
*Lepiota brunneolilacea* Bon & Boiffard	Agaricaceae	AG, CL, CT, EN, ME, PA, RG, SR, TP	Sl
*Lepiota castanea* Quél.	Agaricaceae	AG, CL, CT, EN, ME, PA, RG, SR, TP	Sl
*Lepiota clypeolaria* (Bull.) P. Kumm.	Agaricaceae	AG, CL, CT, EN, ME, PA, RG, SR, TP	Sl
*Lepiota cortinarius* J.E. Lange	Agaricaceae	AG, CL, CT, EN, ME, PA, RG, SR, TP	Sl
*Lepiota cristata* (Bolton) P. Kumm.	Agaricaceae	AG, CL, CT, EN, ME, PA, RG, SR, TP	Sl
*Lepiota echinella* Quél. & G.E. Bernard	Agaricaceae	CT, ME, PA	Sl
*Lepiota erminea* (Fr.) P. Kumm.	Agaricaceae	CT, ME, PA	Sl
*Lepiota felina* (Pers.) P. Karst.	Agaricaceae	CT, ME	Sl
*Lepiota forquignonii* Quél.	Agaricaceae	PA	Sl
*Lepiota griseovirens* Maire	Agaricaceae	CT, ME	Sl
*Lepiota helveola* Bres.	Agaricaceae	AG, PA	Sl
*Lepiota ignivolvata* Bousset & Joss. ex Joss.	Agaricaceae	PA	Sl
*Lepiota kuehneri* Huijsman ex Hora	Agaricaceae	CT, ME	Sl
*Lepiota lilacea* Bres.	Agaricaceae	CT, ME, PA	Sl
*Lepiota magnispora* Murrill	Agaricaceae	CL, CT, ME	Sl
*Lepiota oreadiformis* Velen.	Agaricaceae	PA	Sl
*Lepiota pseudohelveola* Kühner var. *pseudohelveola*	Agaricaceae	CT, ME, PA	Sl
*Lepiota pseudolilacea* Huijsman	Agaricaceae	CT, ME, PA	Sl
*Lepiota subincarnata* J.E. Lange	Agaricaceae	AG, CT, ME, PA, TP	Sl
*Lepiota wasseri* Bon	Agaricaceae	CT, ME	Sl
*Lepiota xanthophylla* P.D. Orton	Agaricaceae	TP	Sl
*Lepista amara* (Alb. & Schwein.) Maire	Incertae sedis	AG, PA, TP	Sl
*Lepista densifolia* (J. Favre) Singer & Clémençon	Incertae sedis	PA, TP	Sl
*Lepista glaucocana* (Bres.) Singer	Incertae sedis	CT, ME, PA	Sl
*Lepista irina* (Fr.) H.E. Bigelow	Incertae sedis	PA, TP	Sl
*Lepista nuda* (Bull.) Cooke	Incertae sedis	AG, PA, TP	Sl
*Lepista ovispora* (J.E. Lange) Gulden	Incertae sedis	CT, ME, PA	Sl
*Lepista panaeolus* (Fr.) P. Karst.	Incertae sedis	AG, PA	Sl
*Lepista personata* (Fr.) Cooke	Incertae sedis	CT, ME, PA	Sl
*Lepista rickenii* Singer	Incertae sedis	CT, ME, PA	Sl
*Lepista sordida* (Schumach.) Singer	Incertae sedis	AG, CL, CT, PA, RG, SR, TP	Sl
*Leptosporomyces fuscostratus* (Burt) Hjortstam	Atheliaceae	PA, TP	Sw
*Leptosporomyces septentrionalis* (J. Erikss.) Krieglst.	Atheliaceae	PA, TP	Sl
*Leratiomyces squamosus* (Pers.) Bridge & Spooner	Strophariaceae	CT, ME, PA	Sl
*Leucoagaricus americanus* (Peck) Vellinga	Agaricaceae	CT, ME, PA	Sl
*Leucoagaricus badhamii* (Berk. & Broome) Singer	Agaricaceae	CT, ME	Sl
*Leucoagaricus barssii* (Zeller) Vellinga	Agaricaceae	CT, ME, PA	Sl
*Leucoagaricus boudierianus* Bon	Agaricaceae	CL, CT, ME, PA	Sl
*Leucoagaricus bresadolae* (Schulzer) Bon & Boiffard var. *bresadolae*	Agaricaceae	CT, ME	Sl
*Leucoagaricus carneifolius* (Gillet) Wasser	Agaricaceae	CT, ME, PA	Sl
*Leucoagaricus cinerascens* (Quél.) Bon & Boiffard	Agaricaceae	CT, ME, PA	Sl
*Leucoagaricus leucothites* (Vittad.) Wasser	Agaricaceae	AG, CL, CT, EN, ME, PA, RG, SR, TP	Sl
*Leucoagaricus littoralis* (Menier) Bon & Boiffard	Agaricaceae	CL, CT, PA, RG	Sl
*Leucoagaricus macrorhizus* Locq. ex E. Horak	Agaricaceae	PA, TP	Sl
*Leucoagaricus menieri* (Sacc.) Singer	Agaricaceae	CT, ME, PA, RG, TP	Sl
*Leucoagaricus pilatianus* (Demoulin) Bon & Boiffard	Agaricaceae	CL, PA	Sl
*Leucoagaricus salmoneophyllus* Bon & Guinb.	Agaricaceae	CL	Sl
*Leucoagaricus serenus* (Fr.) Bon & Boiffard	Agaricaceae	CT, ME, PA	Sl
*Leucoagaricus singeri* (Bon ex Contu & Signor.) Consiglio & Contu	Agaricaceae	CT, ME, PA, SR	Sl
*Leucoagaricus subcretaceus* Bon	Agaricaceae	CT, ME	Sl
*Leucoagaricus subvolvatus* (Malençon & Bertault) Bon	Agaricaceae	CT, ME	Sl
*Leucoagaricus volvatus* Bon & A. Caball.	Agaricaceae	CT, ME, PA, SR	Sl
*Leucoagaricus wichanskyi* (Pilát) Bon & Boiffard	Agaricaceae	CT, ME, PA	Sl
*Leucocoprinus birnbaumii* (Corda) Singer	Agaricaceae	AG, CL, CT, EN, ME, PA, RG, SR, TP	Sl
*Leucocoprinus brebissonii* (Godey) Locq.	Agaricaceae	AG, CL, CT, EN, ME, PA, RG, SR, TP	Sl
*Leucocoprinus cepistipes* (Sowerby) Pat.	Agaricaceae	CT, ME, PA	Sl
*Leucocoprinus cretaceus* (Bull.) Locq.	Agaricaceae	CT, ME, PA	Sl
*Leucocoprinus flos-sulphuris* (Schnizl.) Cejp	Agaricaceae	CT, ME, PA	Sl
*Leucocortinarius bulbiger* (Alb. & Schwein.) Singer	Incertae sedis	CT, ME	Em
*Leucocybe candicans* (Pers.) Vizzini, P. Alvarado, G. Moreno & Consiglio	Incertae sedis	CT, ME, PA	Sl
*Leucocybe connata* (Schumach.) Vizzini, P. Alvarado, G. Moreno & Consiglio	Incertae sedis	CT, ME, PA	Sl
*Leucocybe houghtonii* (W. Phillips) Halama & Pencakowski	Incertae sedis	CT, ME, PA	Sl
*Leucogaster nudus* (Hazsl.) Hollós	Albatrellaceae	CT, ME, PA	UNK
*Leucoinocybe lenta* (Maire) Antonín, Borovička, Holec & Kolařík	Incertae sedis	CT, ME, PA	Em
*Leucopaxillus agrippinae* Buda, Consiglio, Setti & Vizzini	Tricholomataceae	CT, ME	Em
*Leucopaxillus gentianeus* (Quél.) Kotl.	Tricholomataceae	PA, TP	Em
*Leucopaxillus paradoxus* (Costantin & L.M. Dufour) Boursier	Tricholomataceae	AG, CL, CT, EN, ME, PA, RG, SR, TP	Em
*Lichenomphalia umbellifera* (L.) Redhead, Lutzoni, Moncalvo & Vilgalys	Hygrophoraceae	CT, ME	Sl
*Limacella furnacea* (Letell.) E.-J. Gilbert	Amanitaceae	CT, ME, PA	Sl
*Limacella subfurnacea* Contu	Amanitaceae	CL, CT, ME, PA, RG, SR, TP	Sl
*Limacellopsis guttata* (Pers.) Zhu L. Yang, Q. Cai & Y.Y. Cui	Amanitaceae	CT, ME, PA	Sl
*Lindtneria chordulata* (D.P. Rogers) Hjortstam	Stephanosporaceae	CT, ME, PA	Sw
*Lophiostoma compressum* (Pers.) Ces. & De Not.	Lophiostomataceae	CT, ME, PA	Sw
*Lophiostoma viridarium* Cooke	Lophiostomataceae	CT, ME	Sw
*Lophodermium arundinaceum* (Schrad.) Chevall.	Rhytismataceae	CT, ME, PA	Sw
*Lycoperdon atropurpureum* Vittad.	Lycoperdaceae	CT, ME, PA	Sl
*Lycoperdon caudatum* J. Schröt.	Lycoperdaceae	CT, ME	Sl
*Lycoperdon echinatum* Pers.	Lycoperdaceae	AG, CL, CT, EN, ME, PA, RG, SR, TP	Sl
*Lycoperdon ericaeum* Bonord.	Lycoperdaceae	CT, ME	Sl
*Lycoperdon excipuliforme* (Scop.) Pers.	Lycoperdaceae	AG, CT, ME, PA, TP	Sl
*Lycoperdon lividum* Pers.	Lycoperdaceae	CT, ME, PA	Sl
*Lycoperdon mammiforme* Pers.	Lycoperdaceae	PA, TP	Sl
*Lycoperdon marginatum* Vittad.	Lycoperdaceae	CT, ME	Sl
*Lycoperdon molle* Pers.	Lycoperdaceae	PA, TP	Sl
*Lycoperdon nigrescens* Pers.	Lycoperdaceae	CT, ME, PA, TP	Sl
*Lycoperdon perlatum* Pers.	Lycoperdaceae	AG, CT, ME, PA, TP	Sl
*Lycoperdon pratense* Pers.	Lycoperdaceae	CT, ME, PA	Sl
*Lycoperdon radicatum* Durieu & Mont.	Lycoperdaceae	CT, ME, PA	Sl
*Lycoperdon spadiceum* Schaeff.	Lycoperdaceae	CT, ME, PA	Sl
*Lycoperdon umbrinum* Pers.	Lycoperdaceae	CT, ME, PA	Sl
*Lyomyces crustosus* (Pers.) P. Karst.	Schizoporaceae	CT, ME, PA	Sw
*Lyomyces juniperi* (Bourdot & Galzin) Riebesehl & Langer	Schizoporaceae	CT, ME, PA	Sw
*Lyophyllum aemiliae* Consiglio	Lyophyllaceae	CT, ME, TP	St
*Lyophyllum amariusculum* Clémençon	Lyophyllaceae	CT, ME	St
*Lyophyllum buxeum* (Maire) Singer	Lyophyllaceae	AG, CT, PA, RG	St
*Lyophyllum caerulescens* Clémençon ex Kibby	Lyophyllaceae	CT, ME	St
*Lyophyllum crassifolium* (Sacc.) Singer	Lyophyllaceae	AG, PA	St
*Lyophyllum decastes* (Fr.) Singer	Lyophyllaceae	AG, CL, CT, EN, ME, PA, RG, SR, TP	St
*Lyophyllum deliberatum* (Britzelm.) Kreisel	Lyophyllaceae	CT, ME, PA, TP	St
*Lyophyllum immundum* (Berk.) Kühner	Lyophyllaceae	CT, ME, PA, TP	St
*Lyophyllum infumatum* (Bres.) Kühner	Lyophyllaceae	PA, TP	St
*Lyophyllum littorale* (Ballero & Contu) Contu	Lyophyllaceae	CT, ME, PA	St
*Lyophyllum loricatum* (Fr.) Kühner	Lyophyllaceae	CT, ME, PA	St
*Lyophyllum paelochroum* Clémençon	Lyophyllaceae	CT, ME, PA	St
*Lyophyllum tenebrosum* Clémençon	Lyophyllaceae	CT, ME	St
*Macrocystidia cucumis* (Pers.) Joss.	Macrocystidiaceae	CT, ME, PA	Sl
*Macrolepiota excoriata* (Schaeff.) Wasser	Agaricaceae	AG, CL, CT, EN, ME, PA, RG, SR, TP	Sl
*Macrolepiota fuligineosquarrosa* Malençon	Agaricaceae	AG, CL, CT, EN, ME, PA, RG, SR, TP	Sl
*Macrolepiota fuliginosa* (Barla) Bon	Agaricaceae	AG, CL, CT, EN, ME, PA, RG, SR, TP	Sl
*Macrolepiota mastoidea* (Fr.) Singer	Agaricaceae	AG, CL, CT, EN, ME, PA, RG, SR, TP	Sl
*Macrolepiota olivascens* Singer & M.M. Moser	Agaricaceae	AG, CL, CT, EN, ME, PA, RG, SR, TP	Sl
*Macrolepiota permixta* (Barla) Pacioni	Agaricaceae	AG, CL, CT, EN, ME, PA, RG, SR, TP	Sl
*Macrolepiota procera* (Scop.) Singer	Agaricaceae	AG, CL, CT, EN, ME, PA, RG, SR, TP	Sl
*Macrolepiota venenata* Bon	Agaricaceae	AG, CL, CT, EN, ME, PA, RG, SR, TP	Sl
*Mallocybe gymnocarpa* (Kühner) Matheny & Esteve-Rav.	Inocybaceae	CT, ME	Em
*Mallocybe heimii* (Bon) Matheny & Esteve-Rav.	Inocybaceae	PA, SR	Em
*Mallocybe leucoblema* (Kühner) Matheny & Esteve-Rav.	Inocybaceae	CT, ME	Em
*Mallocybe terrigena* (Fr.) Matheny, Vizzini & Esteve-Rav.	Inocybaceae	CT, ME	Em
*Marasmiellus candidus* (Fr.) Singer	Omphalotaceae	AG, CL, CT, EN, ME, PA, RG, SR, TP	Sl
*Marasmiellus confluens* (Pers.) J.S. Oliveira	Marasmiaceae	AG, CL, CT, EN, ME, PA, RG, SR, TP	Sl
*Marasmiellus peronatus* (Bolton) J.S. Oliveira	Marasmiaceae	AG, CL, CT, EN, ME, PA, RG, SR, TP	Sl
*Marasmiellus quercophilus* (Pouzar) J.S. Oliveira	Marasmiaceae	CT, ME, PA, TP	Sl
*Marasmiellus ramealis* (Bull.) Singer	Marasmiaceae	AG, CL, CT, EN, ME, PA, RG, SR, TP	Sw
*Marasmiellus trabutii* (Maire) Singer	Omphalotaceae	CT, ME, PA, SR	Sl
*Marasmius bulliardii* Quél.	Marasmiaceae	AG, CL, CT, EN, ME, PA, RG, SR, TP	Sl
*Marasmius cohaerens* (Pers.) Cooke & Quél.	Marasmiaceae	CT, ME, PA, SR	Sl
*Marasmius collinus* (Scop.) Singer	Marasmiaceae	CT, ME, PA, SR	Sl
*Marasmius epiphylloides* (Rea) Sacc. & Trotter	Marasmiaceae	CT, ME, PA, TP	Sle
*Marasmius epiphyllus* (Pers.) Fr.	Marasmiaceae	AG, CL, CT, EN, ME, PA, RG, SR, TP	Sle
*Marasmius epodius* Bres.	Marasmiaceae	CL, PA, SR	Sl
*Marasmius hudsonii* (Pers.) Fr.	Marasmiaceae	CT, ME, PA	Sle
*Marasmius oreades* (Bolton) Fr.	Marasmiaceae	AG, CL, CT, EN, ME, PA, RG, SR, TP	Sl
*Marasmius rotula* (Scop.) Fr.	Marasmiaceae	AG, CL, CT, EN, ME, PA, RG, SR, TP	Sl
*Marasmius torquescens* Quél.	Marasmiaceae	CT, ME	Sl
*Marasmius wynneae* Berk. & Broome	Marasmiaceae	CT, ME, PA	Sl
*Marcelleina mediterranea* Lantieri & Pfister	Pezizaceae	CT, ME, PA	Sl
*Megacollybia platyphylla* (Pers.) Kotl. & Pouzar	Incertae sedis	AG, CL, CT, EN, ME, PA, RG, SR, TP	Sl
*Melanogaster ambiguus* (Vittad.) Tul. & C. Tul.	Paxillaceae	EN	Em
*Melanogaster broomeanus* Berk.	Paxillaceae	PA	Em
*Melanogaster macrosporus* Velen.	Paxillaceae	AG, PA	Em
*Melanogaster tuberiformis* Corda	Paxillaceae	PA	Em
*Melanogaster variegatus* (Vittad.) Tul. & C. Tul.	Paxillaceae	PA, TP	Em
*Melanoleuca bresadolana* Bon	Incertae sedis	CT, ME	Sl
*Melanoleuca brevipes* (Bull.) Pat.	Incertae sedis	CT, ME, PA	Sl
*Melanoleuca cinereifolia* (Bon) Bon	Incertae sedis	CT, ME, PA	Sl
*Melanoleuca cognata* (Fr.) Konrad & Maubl.	Incertae Sedis	CT, ME, PA	Sl
*Melanoleuca curtipes* (Fr.) Bon	Incertae sedis	CT, ME, PA, TP	Sl
*Melanoleuca diverticulata* G. Moreno & Bon	Incertae sedis	CL, CT, ME, PA	Sl
*Melanoleuca exscissa* (Fr.) Singer	Incertae Sedis	CT, ME, PA	Sl
*Melanoleuca graminicola* Kühner & Maire	Incertae Sedis	AG, PA, TP	Sl
*Melanoleuca grammopodia* (Bull.) Murrill	Incertae Sedis	CT, ME, PA	Sl
*Melanoleuca humilis* (Pers.) Pat.	Incertae sedis	CT, ME, PA	Sl
*Melanoleuca leucophylloides* (Bon) Bon	Incertae sedis	CT, ME	Sl
*Melanoleuca melaleuca* (Pers.) Murrill	Incertae Sedis	AG, CL, CT, EN, ME, PA, RG, SR, TP	Sl
*Melanoleuca meridionalis* G. Moreno & Barrasa	Incertae sedis	CT, ME, PA	Sl
*Melanoleuca microcephala* (P. Karst.) Singer	Incertae sedis	CT, ME, PA	Sl
*Melanoleuca polioleuca* (Fr.) Kühner & Maire	Incertae Sedis	CT, ME, PA	Sl
*Melanoleuca pseudobrevipes* Bon	Incertae sedis	CT, ME	Sl
*Melanoleuca pseudoevenosa* Bon ex Bon & G. Moreno	Incertae Sedis	CT, ME, PA	Sl
*Melanoleuca pseudoluscina* Bon	Incertae sedis	CT, ME	Sl
*Melanoleuca rasilis*(Fr.) Singer	Incertae sedis	CT, ME	Sl
*Melanoleuca schumacheri* (Fr.) Singer	Incertae sedis	CT, ME, PA	Sl
*Melanoleuca strictipes* (P. Karst.) Jul. Schäff.	Incertae sedis	CT, ME, PA	Sl
*Melanoleuca stridula* (Fr.) Singer	Incertae sedis	CT, ME, PA	Sl
*Melanoleuca subpulverulenta* (Pers.) Singer	Incertae Sedis	CT, ME, PA	Sl
*Melanoleuca tristis* M.M. Moser	Incertae sedis	CT, ME, PA	Sl
*Melanoleuca turrita* (Fr.) Singer	Incertae sedis	CT, ME, PA	Sl
*Melanophyllum haematospermum* (Bull.) Kreisel	Agaricaceae	CT, ME	Sl
*Melastiza chateri* (W.G. Sm.) Boud.	Pyronemataceae	AG, CL, CT, EN, ME, PA, RG, SR, TP	Sbg
*Melastiza cornubiensis* (Berk. & Broome) J. Moravec	Pyronemataceae	CT, ME, PA	Sbg
*Mensularia nodulosa* (Fr.) T. Wagner & M. Fisch.	Hymenochaetaceae	CT, ME, PA	Sw
*Mensularia radiata* (Sowerby) Lázaro Ibiza	Hymenochaetaceae	CT, ME, PA	Sw
*Meripilus giganteus* (Pers.) P. Karst.	Meripilaceae	CT, ME, PA	Sw
*Merismodes anomala* (Pers.) Singer	Niaceae	CT, ME, PA	Sw
*Metacapnodium dingleyae* S. Hughes	Metacapnodiaceae	ME	Sw
*Mollisia cinerea* (Batsch) P. Karst.	Mollisiaceae	AG, CL, CT, EN, ME, PA, RG, SR, TP	Sw
*Mollisia melaleuca* (Fr.) Sacc.	Mollisiaceae	CT, ME, PA	Sw
*Montagnea arenaria* (DC.) Zeller	Agaricaceae	AG, PA, SR, TP	Sl
*Montagnea candollei* (Fr.) Fr.	Agaricaceae	CT, ME, PA	Sl
*Morchella costata* Pers.	Morchellaceae	CT, ME, PA	St
*Morchella deliciosa* Fr.	Morchellaceae	AG, CL, CT, EN, ME, PA, RG, SR, TP	St
*Morchella elata* Fr.	Morchellaceae	CT, ME, PA	St
*Morchella esculenta* (L.) Pers.	Morchellaceae	AG, CL, CT, EN, ME, PA, RG, SR, TP	St
*Morchella hortensis* Boud.	Morchellaceae	AG, CL, CT, EN, ME, PA, RG, SR, TP	St
*Morchella semilibera* DC.	Morchellaceae	CT, ME, PA	St
*Morchella tridentina* Bres.	Morchellaceae	CT, ME	St
*Morchella vulgaris* (Pers.) Gray	Morchellaceae	CT, ME	St
*Mucidula mucida* (Schrad.) Pat.	Physalacriaceae	CT, ME, PA	Sw
*Mucronella calva* (Alb. & Schwein.) Fr.	Incertae sedis	CT, ME, PA	Sw
*Mutinus caninus* (Huds.) Fr.	Phallaceae	CT, ME, PA, TP	Sl
*Mycena acicula* (Schaeff.) P. Kumm.	Mycenaceae	CT, ME, PA	St
*Mycena aetites* (Fr.) Quél.	Mycenaceae	PA, TP	St
*Mycena alcalina* (Fr.) P. Kumm.	Mycenaceae	CT, ME, PA	St
*Mycena algeriensis Maire*	Mycenaceae	CT, ME, PA	St
*Mycena amicta* (Fr.) Quél.	Mycenaceae	CT, ME, PA	St
*Mycena arcangeliana* Bres.	Mycenaceae	CT, ME, PA	St
*Mycena aurantiomarginata* (Fr.) Quél.	Mycenaceae	CT, ME, PA	St
*Mycena capillaripes* Peck	Mycenaceae	CT, ME, PA	St
*Mycena cinerella* (P. Karst.) P. Karst.	Mycenaceae	CT, ME, PA	St
*Mycena crocata* (Schrad.) P. Kumm.	Mycenaceae	CT, ME, PA	St
*Mycena epipterygia* (Scop.) Gray	Mycenaceae	CT, ME, PA	St
*Mycena filopes* (Bull.) P. Kumm.	Mycenaceae	CT, ME	St
*Mycena flavescens* Velen.	Mycenaceae	CT, ME, PA	St
*Mycena galericulata* (Scop.) Gray	Mycenaceae	AG, CT, ME, PA, TP	St
*Mycena galopus* (Pers.) P. Kumm.	Mycenaceae	CT, ME, PA	St
*Mycena haematopus* (Pers.) P. Kumm.	Mycenaceae	PA, TP	St
*Mycena inclinata* (Fr.) Quél.	Mycenaceae	CT, ME, PA	Sw
*Mycena leptocephala* (Pers.) Gillet	Mycenaceae	PA, TP	St
*Mycena maculata* P. Karst.	Mycenaceae	CT, ME, PA	St
*Mycena megaspora* Kauffman	Mycenaceae	CT, ME	St
*Mycena meliigena* (Berk. & Cooke) Sacc.	Mycenaceae	CT, ME	St
*Mycena olivaceomarginata* (Massee) Massee	Mycenaceae	CT, ME	St
*Mycena pelianthina* (Fr.) Quél.	Mycenaceae	AG, CL, CT, EN, ME, PA, RG, SR, TP	St
*Mycena polygramma* (Bull.) Gray	Mycenaceae	AG, PA	St
*Mycena pseudocorticola* Kühner	Mycenaceae	CT, ME, PA	St
*Mycena pseudoinclinata* A.H. Sm.	Mycenaceae	CT, ME, PA	Sw
*Mycena pura* (Pers.) P. Kumm.	Mycenaceae	AG, CL, CT, EN, ME, PA, RG, SR, TP	St
*Mycena renati* Quél.	Mycenaceae	CT, ME, PA	St
*Mycena rosea* Gramberg	Mycenaceae	AG, CL, CT, EN, ME, PA, RG, SR, TP	St
*Mycena rosella* (Fr.) P. Kumm.	Mycenaceae	CT, ME	St
*Mycena seynii* Quél.	Mycenaceae	AG, CL, CT, EN, ME, PA, RG, SR, TP	Sc
*Mycena stipata* Maas Geest. & Schwöbel	Mycenaceae	CT, ME	St
*Mycena strobilicola* J. Favre & Kühner	Mycenaceae	AG, CL, CT, EN, ME, PA, RG, SR, TP	Sc
*Mycena supina* (Fr.) P. Kumm.	Mycenaceae	CT, ME, PA	St
*Mycena tintinnabulum* (Paulet) Quél.	Mycenaceae	CT, ME, PA	St
*Mycena villicaulis* Maas Geest.	Mycenaceae	CT, ME	St
*Mycena viridimarginata* P. Karst.	Mycenaceae	CT, ME, PA	St
*Mycena vitilis* (Fr.) Quél.	Mycenaceae	AG, CL, CT, EN, ME, PA, RG, SR, TP	Sw
*Mycena vulgaris* (Pers.) P. Kumm.	Mycenaceae	AG, CL, CT, EN, ME, PA, RG, SR, TP	St
*Mycena xantholeuca* Kühner	Mycenaceae	CT, ME, PA	St
*Mycenastrum corium* (Guers.) Desv.	Agaricaceae	CT, ME	St
*Mycetinis alliaceus* (Jacq.) Earle ex A.W. Wilson & Desjardin	Omphalotaceae	CT, ME, PA	Sl
*Mycetinis scorodonius* (Fr.) A.W. Wilson & Desjardin	Omphalotaceae	AG, CL, CT, EN, ME, PA, RG, SR, TP	Sl
*Mycoacia aurea* (Fr.) J. Erikss. & Ryvarden	Meruliaceae	CT, ME, PA	Sw
*Mycoacia fuscoatra* (Fr.) Donk	Meruliaceae	CT, ME, PA	Sw
*Mycoacia nothofagi* (G. Cunn.) Ryvarden	Meruliaceae	CT, ME, PA	Sw
*Mycoacia uda* (Fr.) Donk	Meruliaceae	CT, ME, PA	Sw
*Mycoaciella bispora* (Stalpers) J. Erikss. & Ryvarden	Meruliaceae	CT, ME, PA	Sw
*Myriostoma coliforme* (Dicks.) Corda	Geastraceae	PA, TP	St
*Myxomphalia maura* (Fr.) Hora	Incertae sedis	CT, ME, PA	Sl
*Nectria aurantiaca* (Bab.) Jacz.	Nectriaceae	CT, ME	Sw
*Nectria cinnabarina* (Tode) Fr.	Nectriaceae	AG, PA	Sw
*Nemania serpens* (Pers.) Gray	Xylariaceae	CT, ME, PA	Sw
*Neoantrodia serialis* (Fr.) Audet	Fomitopsidaceae	CT, ME, PA, TP	Sw
*Neoboletus erythropus* (Pers.) C. Hahn	Boletaceae	AG, CT, ME, PA	Em
*Neoboletus luridiformis* (Rostk.) Gelardi, Simonini & Vizzini	Boletaceae	AG, CT, ME, PA	Em
*Neoboletus xanthopus* (Klofac & A. Urb.) Klofac & A. Urb.	Boletaceae	CT, ME	Em
*Neobulgaria pura* (Pers.) Petr.	Gelatinodiscaceae	CT, ME, PA, TP	Sw
*Neoclitocybe nivea* Singer	Incertae sedis	CT, ME	St
*Neofavolus alveolaris* (DC.) Sotome & T. Hatt	Polyporaceae	CT, ME, PA	Sw
*Neolentinus cyathiformis* (Schaeff.) Della Magg. & Trassin.	Gloeophyllaceae	CT, ME	Em
*Neolentinus lepideus* (Fr.) Redhead & Ginns	Gloeophyllaceae	CT, ME, PA	Em
*Neonectria coccinea* (Pers.) Rossman & Samuels	Nectriaceae	CT, ME, PA	Sw
*Neottiella hetieri* Boud.	Pyronemataceae	RG	St
*Neournula pouchetii* (Berthet & Riousset) Paden	Chorioactidaceae	AG	St
*Nothopanus candidissimus* (Sacc.) Kühner	Pleurotaceae	PA	Sw
*Octaviania asterosperma* Vittad.	Boletaceae	PA	Em
*Octospora convexula* (Pers.) L.R. Batra	Pyronemataceae	RG, SR	Sm
*Octospora grimmiae* Dennis & Itzerott	Pyronemataceae	PA	Sm
*Octospora humosa* (Fr.) Dennis	Pyronemataceae	RG	Sm
*Octospora leucoloma* Hedw.	Pyronemataceae	RG	Sm
*Octospora lutziana* (Boud.) Caillet & Moyne	Pyronemataceae	CT, ME	Sm
*Octospora maireana* (Seaver) Yei Z. Wang	Pyronemataceae	PA	Sm
*Octospora rustica* (Velen.) J. Moravec	Pyronemataceae	PA	Sm
*Odontia fibrosa* (Berk. & M.A. Curtis) Kõljalg	Thelephoraceae	PA	Sw
*Odonticium flavicans* (Bres.) Nakasone	Incertae sedis	PA	Sw
*Omphalina pyxidata* (Bull.) Quél.	Incertae sedis	AG, CL, CT, EN, ME, PA, RG, SR, TP	Sm
*Omphalotus illudens* (Schwein.) Bresinsky & Besl	Omphalotaceae	AG, CL, CT, EN, ME, PA, RG, SR, TP	Sw
*Omphalotus olearius* (DC.) Singer	Omphalotaceae	AG, CL, CT, EN, ME, PA, RG, SR, TP	Sw
*Onnia tomentosa* (Fr.) P. Karst.	Hymenochaetaceae	PA	Sw
*Orbilia coccinella* Fr.	Orbiliaceae	PA	Sw
*Ossicaulis lignatilis* (Pers.) Redhead & Ginns	Lyophyllaceae	AG, CL, CT, EN, ME, PA, RG, SR, TP	Sw
*Otidea alutacea* (Pers.) Massee	Otideaceae	AG, CL, CT, EN, ME, PA, RG, SR, TP	St
*Otidea bufonia* (Pers.) Boud.	Otideaceae	CT, ME, PA, TP	St
*Otidea cochleata* (L.) Fuckel	Otideaceae	CT, ME, PA	St
*Otidea leporina* (Batsch) Fuckel	Otideaceae	CT, ME, PA	St
*Otidea onotica* (Pers.) Fuckel	Otideaceae	AG, CL, CT, EN, ME, PA, RG, SR, TP	St
*Oxyporus latemarginatus* (Durieu & Mont.) Donk	Oxyporaceae	CT, ME, PA	Sw
*Panaeolina foenisecii* (Pers.) Maire	Incertae sedis	CT, ME, PA	Sl
*Panaeolus acuminatus* (P. Kumm.) Quél.	Incertae sedis	CT, ME, PA	Sd
*Panaeolus ater* (J.E. Lange) Kühner & Romagn. ex Bon	Incertae sedis	PA, TP	Sd
*Panaeolus caliginosus* (Jungh.) Gillet	Incertae sedis	CT, ME	Sd
*Panaeolus cinctulus* (Bolton) Sacc.	Incertae sedis	CL, CT, ME, PA	Sd
*Panaeolus cyanescens* Sacc.	Incertae sedis	CT, ME, PA	Sd
*Panaeolus fimiputris* (Bull.) Quél.	Incertae sedis	CT, ME, PA	Sd
*Panaeolus guttulatus* Bres.	Incertae sedis	CT, ME, PA	Sd
*Panaeolus papilionaceus* (Bull.) Quél.	Incertae sedis	AG, CL, CT, EN, ME, PA, RG, SR, TP	Sd
*Panaeolus rickenii* Hora	Incertae sedis	AG, CL, CT, EN, ME, PA, RG, SR, TP	Sd
*Panaeolus semiovatus* (Sowerby) S. Lundell & Nannf.	Incertae sedis	AG, CL, CT, EN, ME, PA, RG, SR, TP	Sd
*Panellus stipticus* (Bull.) P. Karst.	Mycenaceae	AG, CL, CT, EN, ME, PA, RG, SR, TP	Sw
*Panus conchatus* (Bull.) Fr.	Panaceae	CT, ME, PA	Sw
*Panus neostrigosus* Drechsler-Santos & Wartchow	Panaceae	CT, ME, PA	Sw
*Paragalactinia berthetiana* (Donadini) Van Vooren	Pezizaceae	CT, ME, PA	Sm
*Paragalactinia infuscata* (Quél.) Van Vooren	Pezizaceae	CT, ME, PA	Sm
*Paragalactinia michelii* (Boud.) Van Vooren	Pezizaceae	CT, ME, TP	Sm
*Paragalactinia succosa* (Berk.) Van Vooren	Pezizaceae	CT, ME, PA	Sm
*Paralepista ameliae* (Arcang.) Vizzini	Incertae sedis	CT, ME, PA	Sl
*Paralepista flaccida* (Sowerby) Vizzini	Incertae sedis	CT, ME, PA, TP	Sl
*Parasola auricoma* (Pat.) Redhead, Vilgalys & Hopple	Psathyrellaceae	CT, ME, PA, TP	Sl
*Parasola conopilea* (Fr.) Örstadius & E. Larss.	Psathyrellaceae	CT, ME, PA	Sl
*Parasola hemerobia* (Fr.) Redhead, Vilgalys & Hopple	Psathyrellaceae	CT, ME, PA	Sl
*Parasola kuehneri* (Uljé & Bas) Redhead, Vilgalys & Hopple	Psathyrellaceae	CT, ME, PA	Sl
*Parasola megasperma* (P.D. Orton) Redhead, Vilgalys & Hopple	Psathyrellaceae	CT, ME, PA	Sl
*Parasola misera* (P. Karst.) Redhead, Vilgalys & Hopple	Psathyrellaceae	CT, ME	Sl
*Parasola plicatilis* (Curtis) Redhead, Vilgalys & Hopple	Psathyrellaceae	AG, CT, ME, PA, TP	Sl
*Parmastomyces mollissimus* (Maire) Pouzar	Fomitopsidaceae	CT, ME, PA	Sw
*Patellaria atrata* (Hedw.) Fr.	Patellariaceae	CT, ME, PA, SR	Sw
*Paxillus ammoniavirescens* Contu & Dessì	Paxillaceae	CT, ME	St
*Paxillus involutus* (Batsch) Fr.	Paxillaceae	AG, CL, CT, EN, ME, PA, RG, SR, TP	St
*Paxillus ionipus* Quél.	Paxillaceae	CT, ME	Sw
*Paxillus obscurisporus* C. Hahn	Paxillaceae	CT, ME	St
*Paxillus rubicundulus* P.D. Orton	Paxillaceae	CT, ME, PA	St
*Paxina clypeata* (Pers.) Linder	Helvellaceae	CT, ME, TP	St
*Peniophora boidinii* D.A. Reid	Peniophoraceae	AG, CL, CT, EN, ME, PA, RG, SR, TP	Sw
*Peniophora cinerea* (Pers.) Cooke	Peniophoraceae	AG, CL, CT, EN, ME, PA, RG, SR, TP	Sw
*Peniophora incarnata* (Pers.) P. Karst.	Peniophoraceae	AG, CL, CT, EN, ME, PA, RG, SR, TP	Sw
*Peniophora lycii* (Pers.) Höhn. & Litsch.	Peniophoraceae	AG, CL, CT, EN, ME, PA, RG, SR, TP	Sw
*Peniophora meridionalis* Boidin	Peniophoraceae	AG, CL, CT, EN, ME, PA, RG, SR, TP	Sw
*Peniophora nuda* (Fr.) Bres.	Peniophoraceae	PA	Sw
*Peniophora pini* (Schleich.) Boidin	Peniophoraceae	PA	Sw
*Peniophora quercina* (Pers.) Cooke	Peniophoraceae	AG, CL, CT, EN, ME, PA, RG, SR, TP	Sw
*Peniophora versicolor* (Bres.) Sacc. & P. Syd.	Peniophoraceae	PA	Sw
*Peniophora versiformis* (Berk. & M.A. Curtis) Bourdot & Galzin	Peniophoraceae	TP	Sw
*Peniophorella praetermissa* (P. Karst.) K.H. Larss.	Rickenellaceae	CT, ME, PA	Sw
*Peniophorella pubera* (Fr.) P. Karst.	Rickenellaceae	CT, ME	Sw
*Penttilamyces romellii* (Ginns) Zmitr., Kalinovskaya & Myasnikov	Coniophoraceae	PA	Sw
*Perrotia flammea* (Alb. & Schwein.) Boud.	Lachnaceae	PA	Sw
*Peziza ammophila* Durieu & Lév.	Pezizaceae	AG, CT, ME, PA, RG, SR; TP	St
*Peziza arvernensis* Roze & Boud.	Pezizaceae	CT, ME, PA	St
*Peziza brunneoatra* Desm.	Pezizaceae	TP	St
*Peziza cerea* Sowerby	Pezizaceae	PA	St
*Peziza domiciliana* Cooke	Pezizaceae	SR	St
*Peziza echinospora* P. Karst.	Pezizaceae	CT, ME	St
*Peziza fimeti* (Fuckel) E.C. Hansen	Pezizaceae	AG, CL, CT, EN, ME, PA, RG, SR, TP	Sd
*Peziza hortensis* P. Crouan & H. Crouan	Pezizaceae	CT, ME, PA	St
*Peziza lobulata* (Velen.) Svrček	Pezizaceae	PA	St
*Peziza maximovicii* (Velen.) Svrček	Pezizaceae	PA	St
*Peziza micropus* Pers.	Pezizaceae	PA	St
*Peziza praetervisa* Bres.	Pezizaceae	CT, ME	St
*Peziza proteana* (Boud.) Seaver	Pezizaceae	AG, CL, CT, EN, ME, PA, RG, SR, TP	St
*Peziza pseudoammophila* Bon & Donadini	Pezizaceae	CL, PA, RG, SR	St
*Peziza pseudoviolacea* Donadini	Pezizaceae	PA	St
*Peziza repanda* Pers.	Pezizaceae	PA	St
*Peziza sepiatra* Cooke	Pezizaceae	SR	St
*Peziza subviolacea* Svrček	Pezizaceae	PA	St
*Peziza varia* (Hedw.) Alb. & Schwein.	Pezizaceae	CL, PA, RG	St
*Peziza vesiculosa* Bull.	Pezizaceae	AG, CL, CT, EN, ME, PA, RG, SR, TP	St
*Peziza violacea* (Bull.) Relhan	Pezizaceae	PA	St
*Phaeolepiota aurea* (Matt.) Maire	Incertae sedis	AG, ME	Sl
*Phaeomarasmius erinaceus* (Fr.) Scherff. ex Romagn.	Tubariaceae	AG, CL, CT, EN, ME, PA, RG, SR, TP	Sw
*Phanerochaete laevis* (Fr.) J. Erikss. & Ryvarden	Phanerochaetaceae	CT, ME, PA, TP	Sw
*Phanerochaete martelliana* (Bres.) J. Erikss. & Ryvarden	Phanerochaetaceae	CT, ME, PA, TP	Sw
*Phanerochaete sordida* (P. Karst.) J. Erikss. & Ryvarden	Phanerochaetaceae	CT, ME, PA, TP	Sw
*Phaeoclavulina abietina* (Pers.) Giachini	Gomphaceae	CT, ME, PA, TP	St
*Phaeoclavulina flaccida* (Fr.) Giachini	Gomphaceae	CT, ME	St
*Phaeoclavulina myceliosa* (Peck) Franchi & M. Marchetti	Gomphaceae	CT, ME, PA, TP	St
*Phaeoclavulina quercus-ilicis* (Schild) Giachini	Gomphaceae	CT, ME, PA, TP	St
*Phaeolus schweinitzii* (Fr.) Pat.	Laetiporaceae	CT, ME, PA, TP	Sw
*Phaeophlebiopsis ravenelii* (Cooke) Zmitr.	Phanerochaetaceae	CT, ME, PA	Sw
*Phaeotremella foliacea* (Pers.) Wedin, J.C. Zamora & Millanes	Tremellaceae	AG, CL, CT, EN, ME, PA, RG, SR, TP	Sw
*Phallus hadriani* Vent.	Phallaceae	AG, CL, CT, EN, ME, PA, RG, SR, TP	St
*Phallus impudicus* L.	Phallaceae	AG, CL, CT, EN, ME, PA, RG, SR, TP	St
*Phanerochaete jose-ferreirae* (D.A. Reid) D.A. Reid	Phanerochaetaceae	TP	Sw
*Phanerochaete martelliana* (Bres.) J. Erikss. & Ryvarden	Phanerochaetaceae	CT, ME, PA, TP	Sw
*Phanerochaete sordida* (P. Karst.) J. Erikss. & Ryvarden	Phanerochaetaceae	CT, ME, PA, TP	Sw
*Phanerochaete velutina* (DC.) P. Karst.	Phanerochaetaceae	CT, ME, PA, TP	Sw
*Phellinopsis conchata* (Pers.) Y.C. Dai	Hymenochaetaceae	CT, ME, PA, TP	Sw
*Phellinus erectus* A. David, Dequatre & Fiasson	Hymenochaetaceae	AG, CL, CT, EN, ME, PA, RG, SR, TP	Sw
*Phellinus igniarius* (L.) Quél.	Hymenochaetaceae	AG, CL, CT, EN, ME, PA, RG, SR, TP	Sw
*Phellinus pomaceus* (Pers.) Maire	Hymenochaetaceae	CT, ME, PA	Sw
*Phellinus rhamni* (Bondartseva) H. Jahn	Hymenochaetaceae	PA	Sw
*Phellinus rimosus* (Berk.) Pilát	Hymenochaetaceae	PA	Sw
*Phellodon niger* (Fr.) P. Karst.	Thelephoraceae	CT, ME, PA, TP	Em
*Phlebia lilascens* (Bourdot) J. Erikss. & Hjortstam	Meruliaceae	PA	Sw
*Phlebia margaritae* Duhem & H. Michel	Meruliaceae	TP	Sw
*Phlebia rufa* (Pers.) M.P. Christ.	Meruliaceae	PA	Sw
*Phlebia segregata* (Bourdot & Galzin) Parmasto	Meruliaceae	PA	Sw
*Phlebia tremellosa* (Schrad.) Nakasone & Burds.	Meruliaceae	TP	Sw
*Phlebiopsis gigantea* (Fr.) Jülich	Phanerochaetaceae	CT, ME, PA	Sw
*Phlebiopsis ravenelii* (Cooke) Hjortstam	Phanerochaetaceae	PA	Sw
*Phlebiopsis roumeguerei* (Bres.) Jülich & Stalpers	Phanerochaetaceae	PA	Sw
*Phlegmacium amoenolens* (Rob. Henry ex P.D. Orton) Niskanen & Liimat.	Cortinariaceae	CT, ME	Em
*Phlegmacium balteatocumatile* (Rob. Henry ex P.D. Orton) Niskanen & Liimat.	Cortinariaceae	CT, ME	Em
*Phlegmacium moenne-loccozii* (Bidaud) Niskanen & Liimat.	Cortinariaceae	CT, ME	Em
*Phlegmacium triumphans* (Fr.) Niskanen & Liimat.	Cortinariaceae	CT, ME	Em
*Phlegmacium variiforme* (Malençon) Niskanen & Liimat.	Cortinariaceae	CT, ME	Em
*Phloeomana alba* (Bres.) Redhead	Porotheleaceae	PA	Sw
*Phloeomana minutula* (Sacc.) Redhead	Porotheleaceae	CT, ME	Sw
*Pholiota aurivella* (Batsch) P. Kumm.	Strophariaceae	CT, ME, PA	Sw
*Pholiota carbonaria* (Fr.) Singer	Strophariaceae	AG, CL, CT, EN, ME, PA, RG, SR, TP	Sbg
*Pholiota conissans* (Fr.) M.M. Moser	Strophariaceae	CT, ME	Sw
*Pholiota decussata* (Fr.) M.M. Moser	Strophariaceae	CT, ME, TP	St
*Pholiota gummosa* (Lasch) Singer	Strophariaceae	PA, TP	St
*Pholiota lenta* (Pers.) Singer	Strophariaceae	PA	St
*Pholiota lucifera* (Lasch) Quél.	Strophariaceae	PA	Sw
*Pholiota spumosa* (Fr.) Singer	Strophariaceae	PA	St
*Pholiota squarrosa* (Vahl) P. Kumm.	Strophariaceae	CT, ME, PA	Sw
*Pholiota tuberculosa* (Schaeff.) P. Kumm.	Strophariaceae	CT, ME	Sw
*Phylloporia ribis* (Schumach.) Ryvarden	Hymenochaetaceae	PA, TP	Sw
*Phylloporus pelletieri* (Lév.) Quél.	Boletaceae	CT, ME	Em
*Phylloscypha boltonii* (Quél.) Van Vooren & Hairaud	Pezizaceae	CL, SR	St
*Phylloscypha phyllogena* (Cooke) Van Vooren	Pezizaceae	CT, ME, PA	St
*Picipes badius* (Persoon) Zmitr. & Kovalenko	Polyporaceae	CT, ME	Sw
*Picipes melanopus* (Pers.) Zmitr. & Kovalenko	Polyporaceae	CT, ME	Sw
*Pilatotrama ljubarskyi* (Pilát) Zmitrovich	Polyporaceae	SR	Sw
*Pisolithus albus* (Cooke & Massee) Priest	Sclerodermataceae	PA	Em
*Pisolithus arhizus* (Scop.) Rauschert	Sclerodermataceae	AG, CL, CT, EN, ME, PA, RG, SR, TP	Em
*Pithya cupressina* (Batsch) Fuckel	Sarcoscyphaceae	RG, SR	Sw
*Plectania melastoma* (Sowerby) Fuckel	Sarcosomataceae	CT, ME	Sw
*Plectania rhytidia* (Berk.) Nannf. & Korf	Sarcosomataceae	PA	Sw
*Pleurotus cornucopiae* (Paulet) Rolland	Pleurotaceae	CT, ME, PA	Sw
*Pleurotus dryinus* (Pers.) P. Kumm.	Pleurotaceae	AG, CL, CT, EN, ME, PA, RG, SR, TP	Sw
*Pleurotus eryngii* (DC.) Quél. var. *eryngii*	Pleurotaceae	AG, CL, CT, EN, ME, PA, RG, SR, TP	St
*Pleurotus eryngii* (DC.) Quél. var. *elaeoselini* Venturella, Zervakis & La Rocca	Pleurotaceae	AG, CL, CT, EN, ME, PA, RG, SR, TP	St
*Pleurotus eryngii* (DC.) Quél. var. *ferulae* (Lanzi) Sacc.	Pleurotaceae	AG, CL, CT, EN, ME, PA, RG, SR, TP	St
*Pleurotus eryngii* (DC.) Quél. var. *thapsiae* Venturella, Zervakis & Saitta	Pleurotaceae	PA	St
*Pleurotus fuscosquamulosus* D.A. Reid & Eicker	Pleurotaceae	PA	Sw
*Pleurotus nebrodensis* (Inzenga) Quél.	Pleurotaceae	PA	St
*Pleurotus opuntiae* (Durieu & Lév.) Sacc.	Pleurotaceae	CT, ME, PA	Scl
*Pleurotus ostreatus* (Jacq.) P. Kumm.	Pleurotaceae	AG, CL, CT, EN, ME, PA, RG, SR, TP	Sw
*Plicaria carbonaria* Fuckel	Pyronemataceae	PA	Sbg
*Plicaria endocarpoides* (Berk.) Rifai	Pyronemataceae	AG, PA	St
*Pluteus atromarginatus* (Konrad) Kühner	Pluteaceae	CT, ME, PA	Sw
*Pluteus aurantiorugosus* (Trog) Sacc.	Pluteaceae	CT, ME	Sw
*Pluteus cervinus* (Schaeff.) P. Kumm.	Pluteaceae	AG, CL, CT, EN, ME, PA, RG, SR, TP	Sw
*Pluteus chrysophaeus* (Schaeff.) Quél.	Pluteaceae	CT, ME, PA	Sw
*Pluteus cinereofuscus* J.E. Lange	Pluteaceae	CT, ME, PA	Sw
*Pluteus ephebeus* (Fr.) Gillet	Pluteaceae	CT, ME, PA	Sw
*Pluteus exiguus* (Pat.) Sacc.	Pluteaceae	CT, ME	Sw
*Pluteus hiatulus* Romagn.	Pluteaceae	CT, ME, PA	Sw
*Pluteus inquilinus* Romagn.	Pluteaceae	CT, ME, PA	Sw
*Pluteus minimus* (Henn.) Sacc. & P. Syd.	Pluteaceae	CT, ME, PA	Sw
*Pluteus nanus* (Pers.) P. Kumm.	Pluteaceae	CT, ME, PA	Sw
*Pluteus petasatus* (Fr.) Gillet	Pluteaceae	PA, TP	Sw
*Pluteus pusillulus* Romagn.	Pluteaceae	CT, ME, PA	Sw
*Pluteus romellii* (Britzelm.) Sacc.	Pluteaceae	CT, ME, PA	Sw
*Pluteus salicinus* (Pers.) P. Kumm.	Pluteaceae	CT, ME	Sw
*Pluteus satur* Kühner & Romagn.	Pluteaceae	CT, ME	Sw
*Pluteus semibulbosus* (Lasch) Quél.	Pluteaceae	TP	Sw
*Pluteus thomsonii* (Berk. & Broome) Dennis	Pluteaceae	PA	Sw
*Pluteus umbrosus* (Pers.) P. Kumm.	Pluteaceae	PA	Sw
*Podoscypha multizonata* (Berk. & Broome) Pat.	Podoscyphaceae	CT, ME	Sw
*Polyozellus griseopergamaceus* (M.J. Larsen) Svantesson & Kõljalg	Thelephoraceae	CT, ME	St
*Polyporus lipsiensis* (Batsch) E.H.L. Krause	Polyporaceae	CT, ME, PA	Sw
*Poronia punctata* (L.) Fr.	Xylariaceae	PA	Sd
*Porpoloma macrorhizum* (Quél.) Bon	Pseudoclitocybaceae	TP	Em
*Porostereum spadiceum* (Pers.) Hjortstam & Ryvarden	Phanerochaetaceae	CT, ME, PA	Sw
*Postia rennyi* (Berk. & Broome) Rajchenb.	Dacryobolaceae	CT, ME	Sw
*Postia tephroleuca* (Fr.) Jülich	Dacryobolaceae	CT, ME	Sw
*Praetumpfia obducens* (Schumach.) Jaklitsch & Voglmayr	Melanommataceae	PA	Sw
*Propolis farinosa* (Pers.) Fr.	Marthamycetaceae	CT, ME, PA	Sw
*Protostropharia semiglobata* (Batsch) Redhead, Moncalvo & Vilgalys	Strophariaceae	CT, ME, PA	Sd
*Psathyrella ammophila* (Durieu & Lév.) P.D. Orton	Psathyrellaceae	CL, CT, PA, RG, SR	St
*Psathyrella artemisiae* (Pass.) Konrad & Maubl.	Psathyrellaceae	PA	St
*Psathyrella bipellis* (Quél.) A.H. Sm.	Psathyrellaceae	PA	St
*Psathyrella candolleana* (Fr.) Maire	Psathyrellaceae	AG, CL, CT, EN, ME, PA, RG, SR, TP	Sw
*Psathyrella conopilea* (Fr.) A. Pearson & Dennis	Psathyrellaceae	CT, ME, PA, TP	St
*Psathyrella corrugis* (Pers.) Konrad & Maubl.	Psathyrellaceae	CT, ME, PA	St
*Psathyrella cotonea* (Quél.) Konrad & Maubl.	Psathyrellaceae	CT, ME, PA	St
*Psathyrella flexispora* T.J. Wallace & P.D. Orton	Psathyrellaceae	CT, ME	St
*Psathyrella lacrymabunda* (Bull.) M.M. Moser ex A.H. Sm.	Psathyrellaceae	CT, ME, PA	St
*Psathyrella multipedata* (Peck) A.H. Sm.	Psathyrellaceae	CT, ME, PA	St
*Psathyrella niveobadia* (Romagn.) M.M. Moser	Psathyrellaceae	CT, ME	St
*Psathyrella orbitarum* (Romagn.) M.M. Moser	Psathyrellaceae	CT, ME	St
*Psathyrella pennata* (Fr.) A. Pearson & Dennis	Psathyrellaceae	CT, ME, PA	St
*Psathyrella piluliformis* (Bull.) P.D. Orton	Psathyrellaceae	CT, ME, PA	St
*Psathyrella pseudocasca* (Romagn.) Kits van Wav.	Psathyrellaceae	CT, ME	St
*Psathyrella saponacea* F.H. Møller	Psathyrellaceae	CT, ME, PA	St
*Psathyrella spadiceogrisea* (Schaeff.) Maire	Psathyrellaceae	CT, ME, PA	St
*Psathyrella sphagnicola* (Maire) J. Favre	Psathyrellaceae	CT, ME	St
*Psathyrella tephrophylla* (Romagn.) Bon	Psathyrellaceae	CT, ME, PA	St
*Pseudoboletus parasiticus* (Bull.) Šutara	Boletaceae	CT, ME	Em
*Pseudoclitocybe cyathiformis* (Bull.) Singer	Pseudoclitocybaceae	AG, PA	Sl
*Pseudocraterellus undulatus* (Pers.) Rauschert	Hydnaceae	CT, ME, PA	St
*Pseudoinonotus dryadeus* (Pers.) T. Wagner & M. Fisch.	Hymenochaetaceae	CT, ME, PA	Sw
*Pseudopeziza medicaginis* (Lib.) Sacc.	Drepanopezizaceae	CT, ME, PA	St
*Pseudopeziza trifolii* (Biv.) Fuckel	Drepanopezizaceae	CT, ME, PA	St
*Pseudosperma arenicola* (R. Heim) Matheny & Esteve-Rav.	Inocybaceae	PA, SR	Em
*Pseudosperma flavellum* (P. Karst.) Matheny & Esteve-Rav.	Inocybaceae	CT, ME, PA	Em
*Pseudosperma rimosum* (Bull.) Matheny & Esteve-Rav.	Inocybaceae	AG, PA, TP	Em
*Pseudosperma squamatum* (J.E. Lange) Matheny & Esteve-Rav.	Inocybaceae	CT, ME, PA, TP	Em
*Pseudotomentella tristis* (P. Karst.) M.J. Larsen	Thelephoraceae	CT, ME, PA, TP	Em
*Psilocybe aurantiaca* (Cooke) Noordel.	Hymenogastraceae	CT, ME, PA, TP	Sd
*Psilocybe coronilla* (Bull.) Noordel.	Hymenogastraceae	AG, PA, TP	Sd
*Psilocybe cyanescens* Wakef.	Hymenogastraceae	CT, ME, PA, TP	Sd
*Psilocybe semilanceata* (Fr.) P. Kumm.	Hymenogastraceae	CT, ME	Sd
*Pterula multifida* (Chevall.) Fr.	Pterulaceae	CT, ME	St
*Pulcherricium coeruleum* (Lam.) Parmasto	Phanerochaetaceae	AG, CL, CT, EN, ME, PA, RG, SR, TP	Sw
*Pulchroboletus roseoalbidus* (Alessio & Littini) Gelardi, Vizzini & Simonini	Boletaceae	CT, ME, PA	Em
*Pulvinula archeri* (Berk.) Rifai	Pulvinulaceae	CT, ME, PA	Sw
*Pulvinula johannis* Lantieri	Pulvinulaceae	CT, ME, PA	Sw
*Pycnoporus cinnabarinus* (Jacq.) P. Karst.	Polyporaceae	CT, ME, PA	Sw
*Pyronema domesticum* (Sowerby) Sacc.	Pyronemataceae	AG, CL, CT, EN, ME, PA, RG, SR, TP	Sbg
*Radulomyces confluens* (Fr.) M.P. Christ.	Radulomycetaceae	CT, ME, PA	Sw
*Radulomyces molaris* (Chaillet ex Fr.) M.P. Christ.	Radulomycetaceae	AG, CL, CT, EN, ME, PA, RG, SR, TP	Sw
*Ramaria aurea* (Schaeff.) Quél.	Gomphaceae	CT, ME, PA	St
*Ramaria botrytis* (Pers.) Bourdot	Gomphaceae	AG, CL, CT, EN, ME, PA, RG, SR, TP	St
*Ramaria fennica* (P. Karst.) Ricken	Gomphaceae	CT, ME	St
*Ramaria flava* (Schaeff.) Quél.	Gomphaceae	CT, ME, PA	St
*Ramaria flavescens* (Schaeff.) R.H. Petersen	Gomphaceae	CT, ME	St
*Ramaria formosa* (Pers.) Quél.	Gomphaceae	CT, ME, PA	St
*Ramaria gracilis* (Pers.) Quél.	Gomphaceae	CT, ME, PA	St
*Ramaria pallida* (Schaeff.) Ricken	Gomphaceae	CT, ME, PA	St
*Ramaria pseudogracilis* R.H. Petersen	Gomphaceae	CT, ME, PA	St
*Ramaria stricta* (Pers.) Quél.	Gomphaceae	AG, PA, TP	St
*Ramaricium albo-ochraceum* (Bres.) Jülich	Gomphaceae	CT, ME, PA	St
*Ramariopsis kunzei* (Fr.) Corner	Clavariaceae	CT, ME, PA	St
*Ramariopsis pulchella* (Boud.) Corner	Clavariaceae	CT, ME, PA	St
*Ramariopsis subtilis* (Pers.) R.H. Petersen	Clavariaceae	CT, ME, TP	St
*Reddellomyces donkii* (Malençon) Trappe, Castellano & Malajczuk	Tuberaceae	AG, CL, CT, EN, ME, PA, RG, SR, TP	Sw
*Resupinatus applicatus* (Batsch) Gray	Pleurotaceae	AG, CL, CT, EN, ME, PA, RG, SR, TP	Sw
*Resupinatus poriaeformis* (Pers.) Thorn, Moncalvo & Redhead	Pleurotaceae	PA	Sw
*Rickenella fibula* (Bull.) Raithelh.	Rhickenellaceae	CT, ME, PA	Sm
*Rheubarbariboletus armeniacus* (Quél.) Vizzini, Simonini & Gelardi	Boletaceae	CT, ME, PA	Em
*Rheubarbariboletus persicolor* (H. Engel, Klofac, H. Grünert & R. Grünert) Vizzini, Simonini & Gelardi	Boletaceae	CT, ME	Em
*Rhizoctonia fusispora* (J. Schröt.) Oberw., R. Bauer, Garnica & R. Kirschner	Ceratobasidiaceae	PA	Sw
*Rhizomarasmius setosus* (Sowerby) Antonín & A. Urb.	Physalacriaceae	CT, ME	Sle
*Rhizopogon luteolus* Fr.	Rhizopogonaceae	CT, ME, PA	Em
*Rhizopogon obtextus* (Spreng.) R. Rauschert	Rhizopogonaceae	CT, ME	Em
*Rhizopogon roseolus* (Corda) Th. Fr.	Rhizopogonaceae	AG, CT, ME, PA	Em
*Rhodocollybia butyracea* (Bull.) Lennox	Omphalotaceae	PA, TP	St
*Rhodocollybia maculata* (Alb. & Schwein.) Singer	Omphalotaceae	CT, ME, PA	St
*Rhodocybe truncata* (Schaeff.) Singer	Entolomataceae	CT, ME, PA	St
*Rhodophana nitellina* (Fr.) Papetti	Entolomataceae	CT, ME, PA	St
*Rhodotus palmatus* (Bull.) Maire	Physalacriaceae	CT, ME, PA	Sw
*Rigidoporus ulmarius* (Sowerby) Imazeki	Meripilaceae	AG, CL, CT, EN, ME, PA, RG, SR, TP	Sw
*Ripartites tricholoma* (Alb. & Schwein.) P. Karst.	Incertae sedis	PA	St
*Roridomyces roridus* (Fr.) Rexer	Mycenaceae	CT, ME	Sw
*Rosellinia aquila* (Fr.) Ces. & De Not.	Xylariaceae	CT, ME, PA	Sw
*Rosellinia mammiformis* (Pers.) Ces. & De Not.	Xylariaceae	CT, ME, PA	Sw
*Rubroboletus legaliae* (Pilát & Dermek) Della Magg. & Trassin.	Boletaceae	CT, ME, PA	Em
*Rubroboletus lupinus* (Fr.) Costanzo, Gelardi, Simonini & Vizzini	Boletaceae	CT, ME, PA	Em
*Rubroboletus rhodoxanthus* (Krombh.) Kuan Zhao & Zhu L. Yang	Boletaceae	CT, ME, PA	Em
*Rubroboletus rubrosanguineus* (Cheype) Kuan Zhao & Zhu L. Yang	Boletaceae	CT, ME, PA	Em
*Rubroboletus satanas* (Lenz) Kuan Zhao & Zhu L. Yang	Boletaceae	CT, ME, PA	Em
*Ruhlandiella peregrina* Lantieri & Pfister	Pezizaceae	CT, ME, PA	Em
*Russula acrifolia* Romagn.	Russulaceae	CT, ME, PA, TP	Em
*Russula adusta* (Pers.) Fr.	Russulaceae	CT, ME, PA, TP	Em
*Russula albonigra* (Krombh.) Fr.	Russulaceae	CT, ME, PA	Em
*Russula alnetorum* Romagn.	Russulaceae	CT, ME, PA	Em
*Russula alutacea* (Fr.) Fr.	Russulaceae	CT, ME	Em
*Russula amarissima* Romagn. & E.-J. Gilbert	Russulaceae	CT, ME, PA	Em
*Russula amoena* Quél.	Russulaceae	CT, ME	Em
*Russula amoenicolor* Romagn.	Russulaceae	PA, TP	Em
*Russula amoenolens* Romagn.	Russulaceae	CT, ME, TP	Em
*Russula anatina* Romagn.	Russulaceae	CT, ME, PA	Em
*Russula anthracina* Romagn.	Russulaceae	CT, ME, PA	Em
*Russula atropurpurea* (Krombh.) Britzelm.	Russulaceae	CT, ME, PA	Em
*Russula atrorubens* Quél.	Russulaceae	CT, ME, PA	Em
*Russula aurea* Pers.	Russulaceae	CT, ME, PA	Em
*Russula aurora* Krombh.	Russulaceae	CT, ME, PA	Em
*Russula badia* Quél.	Russulaceae	CT, ME, PA	Em
*Russula caerulea* Fr.	Russulaceae	CT, ME, PA	Em
*Russula carminipes* J. Blum	Russulaceae	CT, ME, PA	Em
*Russula cavipes* Britzelm.	Russulaceae	CT, ME, PA	Em
*Russula cessans* A. Pearson	Russulaceae	CT, ME, PA	Em
*Russula chloroides* (Krombh.) Bres.	Russulaceae	AG, CL, CT, EN, ME, PA, RG, SR, TP	Em
*Russula cicatricata* Romagn. ex Bon	Russulaceae	CT, ME, PA	Em
*Russula citrinochlora* Singer	Russulaceae	CT, ME, PA	Em
*Russula clariana* R. Heim ex Kuyper & Vuure	Russulaceae	CT, ME	Em
*Russula claroflava* Grove	Russulaceae	CT, ME, PA	Em
*Russula cuprea* Krombh.	Russulaceae	CT, ME, PA	Em
*Russula curtipes* F.H. Møller & Jul. Schäff.	Russulaceae	CT, ME, PA	Em
*Russula cyanoxantha* (Schaeff.) Fr.	Russulaceae	AG, CL, CT, EN, ME, PA, RG, SR, TP	Em
*Russula decipiens* (Singer) Kühner & Romagn.	Russulaceae	PA, TP	Em
*Russula decolorans* (Fr.) Fr.	Russulaceae	CT, ME, PA	Em
*Russula delica* Fr.	Russulaceae	AG, CL, CT, EN, ME, PA, RG, SR, TP	Em
*Russula densifolia* Secr. ex Gillet	Russulaceae	PA, TP	Em
*Russula elegans* Bres.	Russulaceae	CT, ME, PA	Em
*Russula emetica* (Schaeff.) Pers.	Russulaceae	AG, CL, CT, EN, ME, PA, RG, SR, TP	Em
*Russula emeticicolor* Jul. Schäff.	Russulaceae	CT, ME	Em
*Russula faginea* Romagn.	Russulaceae	CT, ME, PA	Em
*Russula farinipes* Romell	Russulaceae	CT, ME, TP	Em
*Russula foetens* Pers.	Russulaceae	CT, ME, PA, TP	Em
*Russula font-queri* Singer	Russulaceae	CT, ME, PA, TP	Em
*Russula fragrans* Romagn.	Russulaceae	CT, ME, PA, TP	Em
*Russula fragilis* Fr.	Russulaceae	CT, ME, PA, TP	Em
*Russula fragrantissima* Romagn.	Russulaceae	CT, ME, PA, TP	Em
*Russula galochroides* Sarnari	Russulaceae	CT, ME, PA, TP	Em
*Russula gracilis* Burl.	Russulaceae	CT, ME, PA, TP	Em
*Russula grata* Britzelm.	Russulaceae	CT, ME, PA, TP	Em
*Russula graveolens* Romell	Russulaceae	CT, ME, PA, TP	Em
*Russula grisea* Fr.	Russulaceae	CT, ME, PA, TP	Em
*Russula helios* Malençon ex Sarnari	Russulaceae	CT, ME	Em
*Russula heterophylla* (Fr.) Fr.	Russulaceae	CT, ME, PA, TP	Em
*Russula ilicis* Romagn., Chevassut & Privat	Russulaceae	CT, ME, PA	Em
*Russula illota* Romagn.	Russulaceae	CT, ME, PA	Em
*Russula insignis* Quél.	Russulaceae	CT, ME, PA	Em
*Russula integra* (L.) Fr.	Russulaceae	CT, ME, PA	Em
*Russula intermedia* P. Karst.	Russulaceae	CT, ME, PA	Em
*Russula laeta* Jul. Schäff.	Russulaceae	CT, ME, PA	Em
*Russula langei* Bon	Russulaceae	CT, ME	Em
*Russula luteotacta* Rea	Russulaceae	CT, ME, PA	Em
*Russula maculata* Quél.	Russulaceae	CT, ME, PA, TP	Em
*Russula medullata* Romagn.	Russulaceae	CT, ME, PA	Em
*Russula melliolens* Quél.	Russulaceae	CT, ME, PA	Em
*Russula minutula* Velen.	Russulaceae	CT, ME	Em
*Russula mollis* Quél.	Russulaceae	CT, ME, PA	Em
*Russula monspeliensis* Sarnari	Russulaceae	CT, ME, TP	Em
*Russula mustelina* Fr.	Russulaceae	CT, ME, PA	Em
*Russula nitida* (Pers.) Fr.	Russulaceae	CT, ME, PA	Em
*Russula nobilis* Velen.	Russulaceae	CT, ME, PA	Em
*Russula nuragica* Sarnari	Russulaceae	CT, ME	Em
*Russula ochroleuca* Fr.	Russulaceae	CT, ME, PA	Em
*Russula odorata* Romagn.	Russulaceae	CT, ME, TP	Em
*Russula olivacea* (Schaeff.) Fr.	Russulaceae	AG, CL, CT, EN, ME, PA, RG, SR, TP	Em
*Russula olivaceoviolascens* Gillet	Russulaceae	CT, ME, PA	Em
*Russula pallidospora* J. Blum ex Romagn.	Russulaceae	CT, ME	Em
*Russula paludosa* Britzelm.	Russulaceae	CT, ME, PA	Em
*Russula parazurea* Jul. Schäff.	Russulaceae	CT, ME, PA	Em
*Russula pectinata* Fr.	Russulaceae	AG, CL, CT, EN, ME, PA, RG, SR, TP	Em
*Russula pectinatoides* Peck	Russulaceae	PA, TP	Em
*Russula persicina* Krombh.	Russulaceae	PA, TP	Em
*Russula praetervisa* Sarnari	Russulaceae	CT, ME	Em
*Russula prinophila* Sarnari	Russulaceae	CT, TP	Em
*Russula puellaris* Fr.	Russulaceae	PA, TP	Em
*Russula purpurascens* Bres.	Russulaceae	PA, TP	Em
*Russula queletii* Fr.	Russulaceae	AG, CL, CT, EN, ME, PA, RG, SR, TP	Em
*Russula rhodomelanea* Sarnari	Russulaceae	CT, ME	Em
*Russula raoultii* Quél.	Russulaceae	CT, ME, PA	Em
*Russula ravida* Fr.	Russulaceae	CT, ME, PA	Em
*Russula risigallina* (Batsch) Sacc.	Russulaceae	CT, ME, PA	Em
*Russula rosea* Pers.	Russulaceae	CT, ME, PA, TP	Em
*Russula roseipes* Secr. ex Bres.	Russulaceae	CT, ME	Em
*Russula rubroalba* (Singer) Romagn.	Russulaceae	CT, ME, PA	Em
*Russula rubrocarminea* Romagn.	Russulaceae	CT, ME, PA	Em
*Russula sanguinaria* (Schumach.) Rauschert	Russulaceae	CT, ME, PA	Em
*Russula sanguinea* Fr.	Russulaceae	AG, CL, CT, EN, ME, PA, RG, SR, TP	Em
*Russula sardonia* Fr.	Russulaceae	CT, ME, PA, TP	Em
*Russula seperina* Dupain	Russulaceae	CT, ME, PA	Em
*Russula sericatula* Romagn.	Russulaceae	CT, ME, PA	Em
*Russula silvestris* (Singer) Reumaux	Russulaceae	CT, ME, PA	Em
*Russula sororia* (Fr.) Romell	Russulaceae	CT, ME, PA	Em
*Russula sphagnophila* Kauffman	Russulaceae	CT, ME, PA	Em
*Russula straminea* Malençon	Russulaceae	CT, ME	Em
*Russula subterfurcata* Romagn.	Russulaceae	CT, ME, PA, TP	Em
*Russula tyrrhenica* Sarnari	Russulaceae	CT, ME	Em
*Russula torulosa* Bres.	Russulaceae	AG, CL, CT, EN, ME, PA, RG, SR, TP	Em
*Russula turci* Bres.	Russulaceae	CT, ME	Em
*Russula unicolor* Romagn.	Russulaceae	CT, ME, PA, TP	Em
*Russula velenovskyi* Melzer & Zvára	Russulaceae	CT, ME, PA, TP	Em
*Russula versicolor* Jul. Schäff.	Russulaceae	CT, ME	Em
*Russula vesca* Fr.	Russulaceae	AG, CL, CT, EN, ME, PA, RG, SR, TP	Em
*Russula veternosa* Fr.	Russulaceae	CT, ME, PA	Em
*Russula vinosa* Lindblad	Russulaceae	CT, ME, PA	Em
*Russula vinosopurpurea* Jul. Schäff.	Russulaceae	CT, ME, PA	Em
*Russula violacea* Quél.	Russulaceae	AG, CL, CT, EN, ME, PA, RG, SR, TP	Em
*Russula violeipes* Quél.	Russulaceae	CT, ME, PA, TP	Em
*Russula virescens* (Schaeff.) Fr.	Russulaceae	PA, TP	Em
*Russula werneri* Maire	Russulaceae	CT, ME, TP	Em
*Russula xerampelina* (Schaeff.) Fr.	Russulaceae	AG, CL, CT, EN, ME, PA, RG, SR, TP	Em
*Russula zvarae* Velen.	Russulaceae	CT, ME	Em
*Saccobolus glaber* (Pers.) Lambotte	Ascobolaceae	PA	Sd
*Saccothecium sepincola* (Fr.) Fr.	Saccotheciaceae	PA	Sw
*Saproamanita vittadinii* (Moretti) Redhead, Vizzini, Drehmel & Contu	Amanitaceae	CT, ME, PA	Em
*Sarcodon amarescens* (Quél.) Quél.	Bankeraceae	CT, ME, PA	St
*Sarcodon cyrneus* Maas Geest.	Bankeraceae	CT, ME, PA	St
*Sarcodon imbricatus* (L.) P. Karst.	Bankeraceae	AG, CL, CT, EN, ME, PA, RG, SR, TP	St
*Sarcodon laevigatus* (Sw.) P. Karst.	Bankeraceae	CT, ME	St
*Sarcodon leucopus* (Pers.) Maas Geest. & Nannf.	Bankeraceae	CT, ME, PA	St
*Sarcodontia crocea* (Schwein.) Kotl.	Meruliaceae	CT, ME, PA	Sw
*Sarcodontia spumea* (Sowerby) Spirin	Meruliaceae	CT, ME, PA	Sw
*Sarcoscypha austriaca* (Beck ex Sacc.) Boud.	Sarcoscyphaceae	CT, ME	Sw
*Sarcoscypha coccinea* (Gray) Boud.	Sarcoscyphaceae	AG, CL, CT, EN, ME, PA, RG, SR, TP	Sw
*Sarcoscypha macaronesica* Baral & Korf	Sarcoscyphaceae	CT, ME	Sw
*Sarcosphaera coronaria* (Jacq.) J. Schröt.	Pezizaceae	AG, CL, CT, EN, ME, PA, RG, SR, TP	St
*Schenella pityophila* (Malençon & Riousset) Estrada & Lado	Geastraceae	PA	St
*Schenella simplex* T. Macbr.	Geastraceae	EN	St
*Schizophyllum commune* Fr.	Schizophyllaceae	AG, CL, CT, EN, ME, PA, RG, SR, TP	Sw
*Schizopora paradoxa* (Schrad.) Donk	Schizoporaceae	AG, CL, CT, EN, ME, PA, RG, SR, TP	Sw
*Scleroderma areolatum* Ehrenb.	Sclerodermataceae	PA, TP	Em
*Scleroderma bovista* Fr.	Sclerodermataceae	CT, ME, PA	Em
*Scleroderma cepa* Pers.	Sclerodermataceae	PA, TP	Em
*Scleroderma citrinum* Pers.	Sclerodermataceae	PA, TP	Em
*Scleroderma flavidum* Ellis & Everh.	Sclerodermataceae	AG, CL, CT, EN, ME, PA, RG, SR, TP	Em
*Scleroderma meridionale* Demoulin & Malençon	Sclerodermataceae	CT, ME, PA	Em
*Scleroderma polyrhizum* (J.F. Gmel.) Pers.	Sclerodermataceae	CT, ME, PA	Em
*Scleroderma verrucosum* (Bull.) Pers.	Sclerodermataceae	AG, CL, CT, EN, ME, PA, RG, SR, TP	Em
*Sclerogaster compactus* (Tul. & C. Tul.) Sacc.	Sclerogastraceae	CT, ME, PA	UNK
*Scopuloides rimosa* (Cooke) Jülich	Meruliaceae	CT, ME, PA	Sw
*Scutellinia hyperborea* T. Schumach.	Pyronemataceae	CT, ME, PA	Sw
*Scutellinia legaliae* Lohmeyer & Häffner	Pyronemataceae	CT, ME, PA	Sw
*Scutellinia minor* (Velen.) Svrček	Pyronemataceae	CT, ME, PA	Sw
*Scutellinia scutellata* (L.) Lambotte	Pyronemataceae	AG, CL, CT, EN, ME, PA, RG, SR, TP	Sw
*Scutellinia trechispora* (Berk. & Broome) Lambotte	Pyronemataceae	CT, ME, PA	Sw
*Scutiger pes-caprae* (Pers.) Bondartsev & Singer	Albatrellaceae	CT, ME	Sw
*Scytinostroma hemidichophyticum* Pouzar	Peniophoraceae	PA	Sw
*Scytinostroma portentosum* (Berk. & M.A. Curtis) Donk	Peniophoraceae	PA	Sw
*Sebacina dimitica* Oberw.	Sebacinaceae	PA	Em
*Sebacina epigaea* (Berk. & Broome) Bourdot & Galzin	Sebacinaceae	PA	Em
*Septobasidium rameale* (Berk.) Bres.	Septobasidiaceae	CT, ME	Sw
*Sepultariella patavina* (Cooke & Sacc.) Van Vooren, U. Lindem. & Healy	Pyronemataceae	CT, ME, PA, SR	Sbg
*Serpula lacrymans* (Wulfen) J. Schröt.	Serpulaceae	CT, ME, PA	Sw
*Setchelliogaster tenuipes* (Setch.) Pouzar	Bolbitiaceae	CL, CT, ME, PA, SR, TP	Em
*Sidera lenis* (P. Karst.) Miettinen	Incertae sedis	PA	Sw
*Simocybe sumptuosa* (P.D. Orton) Singer	Crepidotaceae	PA	Sw
*Sistotrema brinkmannii* (Bres.) J. Erikss.	Hydnaceae	CT, ME, PA	Sw
*Sistotrema confluens* Pers.	Hydnaceae	AG, PA	Sw
*Sistotrema muscicola* (Pers.) S. Lundell	Hydnaceae	CL	Sw
*Sistotrema oblongisporum* M.P. Christ. & Hauerslev	Hydnaceae	CL	Sw
*Skeletocutis nivea* (Jungh.) Jean Keller	Incrustoporiaceae	AG, CL, CT, EN, ME, PA, RG, SR, TP	Sw
*Skeletocutis percandida* (Malençon & Bertault) Jean Keller	Incrustoporiaceae	PA	Sw
*Skeletocutis subincarnata* (Peck) Jean Keller	Incrustoporiaceae	PA	Sw
*Skvortzovia georgica* (Parmasto) G. Gruhn & Hallenb.	Incertae sedis	PA	Sw
*Smardaea planchonis* (Dunal ex Boud.) Korf & W.Y. Zhuang	Pyronemataceae	RG, SR	Sm
*Sordaria fimicola* (Roberge ex Desm.) Ces. & De Not.	Sordariaceae	AG, CL, CT, EN, ME, PA, RG, SR, TP	Sd
*Sordaria humana* (Fuckel) G. Winter	Sordariaceae	PA	Sd
*Sparassis crispa* (Wulfen) Fr.	Sparassidaceae	AG, CL, CT, EN, ME, PA, RG, SR, TP	St
*Spathularia flavida* Pers.	Cudoniaceae	PA	St
*Sphaerobolus stellatus* Tode	Geastraceae	PA	Sw
*Spinellus fusiger* (Link) Tiegh.	Phycomycetaceae	CT, ME	Pm
*Steccherinum fimbriatum* (Pers.) J. Erikss.	Steccherinaceae	AG, CL, CT, EN, ME, PA, RG, SR, TP	Sw
*Steccherinum ochraceum* (Pers.) Gray	Steccherinaceae	AG, CL, CT, EN, ME, PA, RG, SR, TP	Sw
*Steccherinum oreophilum* Lindsey & Gilb.	Steccherinaceae	PA, TP	Sw
*Steccherinum straminellum* (Bres.) Melo	Steccherinaceae	PA, TP	Sw
*Stereum gausapatum* (Fr.) Fr.	Stereaceae	AG, CL, CT, EN, ME, PA, RG, SR, TP	Sw
*Stereum hirsutum* (Willd.) Pers.	Stereaceae	AG, CL, CT, EN, ME, PA, RG, SR, TP	Sw
*Stereum insignitum* Quél.	Stereaceae	PA	Sw
*Stereum ochraceoflavum* (Schwein.) Sacc.	Stereaceae	CL	Sw
*Stereum rugosum* Pers.	Stereaceae	ME, PA	Sw
*Stereum sanguinolentum* (Alb. & Schwein.) Fr.	Stereaceae	CT, ME, PA	Sw
*Stereum subtomentosum* Pouzar	Stereaceae	CT, ME, PA	Sw
*Strobilomyces strobilaceus* (Scop.) Berk.	Boletaceae	AG, CL, CT, EN, ME, PA, RG, SR, TP	Em
*Strobilurus stephanocystis* (Kühner & Romagn. ex Hora) Singer	Physalacriaceae	AG, CL, CT, EN, ME, PA, RG, SR, TP	Em
*Strobilurus tenacellus* (Pers.) Singer	Physalacriaceae	AG, CL, CT, EN, ME, PA, RG, SR, TP	Em
*Stropharia aeruginosa* (Curtis) Quél.	Strophariaceae	AG, CL, CT, EN, ME, PA, RG, SR, TP	Sd
*Stropharia caerulea* Kreisel	Strophariaceae	CT, ME, PA	Sd
*Subulicystidium longisporum* (Pat.) Parmasto	Hydnodontaceae	PA	Sw
*Subulicystidium perlongisporum* Boidin & Gilles	Hydnodontaceae	PA	Sw
*Suillellus adalgisae* (Marsico & Musumeci) N. Schwab	Boletaceae	CT, ME, PA, TP	Em
*Suillellus caucasicus* (Singer ex Alessio) Blanco-Dios	Boletaceae	CT, ME, PA, TP	Em
*Suillellus comptus* (Simonini) Vizzini, Simonini & Gelardi	Boletaceae	CT, ME, PA, TP	Em
*Suillellus luridus* (Schaeff.) Murrill	Boletaceae	AG, CL, CT, EN, ME, PA, RG, SR, TP	Em
*Suillellus mendax* (Simonini & Vizzini) Vizzini, Simonini & Gelardi	Boletaceae	CT, ME	Em
*Suillellus permagnificus* (Pöder) Blanco-Dios	Boletaceae	CT, ME, PA	Em
*Suillellus queletii* (Schulzer) Vizzini, Simonini & Gelardi	Boletaceae	AG, CL, CT, EN, ME, PA, RG, SR, TP	Em
*Suillus alkaliaurantians* Pantidou & Watling	Suillaceae	AG, CL, CT, EN, ME, PA, RG, SR, TP	Em
*Suillus bellinii* (Inzenga) Kuntze	Suillaceae	AG, CL, CT, EN, ME, PA, RG, SR, TP	Em
*Suillus bovinus* (L.) Roussel	Suillaceae	AG, CL, CT, EN, ME, PA, RG, SR, TP	Em
*Suillus collinitus* (Fr.) Kuntze	Suillaceae	AG, CL, CT, EN, ME, PA, RG, SR, TP	Em
*Suillus granulatus* (L.) Roussel	Suillaceae	AG, CL, CT, EN, ME, PA, RG, SR, TP	Em
*Suillus lakei* (Murrill) A.H. Sm. & Thiers	Suillaceae	CT, ME, PA	Em
*Suillus luteus* (L.) Roussel	Suillaceae	AG, CL, CT, EN, ME, PA, RG, SR, TP	Em
*Suillus mediterraneensis* (Jacquet. & J. Blum) Redeuilh	Suillaceae	AG, CL, CT, EN, ME, PA, RG, SR, TP	Em
*Suillus placidus* (Bonord.) Singer	Suillaceae	CT, ME, PA	Em
*Szczepkamyces campestris* (Quél.) Zmitr.	Polyporaceae	AG, PA	Sw
*Syzygospora tumefaciens* (Ginns & Sunhede) Ginns	Filobasidiaceae	CT, ME, PA	Pm
*Tapinella atrotomentosa* (Batsch) Šutara	Tapinellaceae	CT, ME, PA	Sw
*Tapinella panuoides* (Fr.) E.-J. Gilbert	Tapinellaceae	AG, CT, PA, RG, TP	Sw
*Tarzetta catinus* (Holmsk.) Korf & J.K. Rogers	Tarzettaceae	AG, CL, CT, EN, ME, PA, RG, SR, TP	Sl
*Tarzetta cupularis* (L.) Lambotte	Tarzettaceae	AG, CL, CT, EN, ME, PA, RG, SR, TP	St
*Tephrocybe ambusta* (Fr.) Donk	Lyophyllaceae	CT, ME, PA	Sbg
*Tephrocybe anthracophila* (Lasch) P.D. Orton	Lyophyllaceae	CT, ME, PA	Sbg
*Tephrocybe atrata* (Fr.) Donk	Lyophyllaceae	CT, ME, PA, TP	Sbg
*Tephrocybe confusa* (P.D. Orton) P.D. Orton	Lyophyllaceae	CT, ME	Sbg
*Tephrocybe putida* (P. Karst.) M.M. Moser	Lyophyllaceae	CT, ME	Sbg
*Tephrocybe rancida* (Fr.) Donk	Lyophyllaceae	CT, ME, PA	Sbg
*Terana coerulea* (Lam.) Kuntze	Phanerochaetaceae	CT, ME, PA	Sw
*Terfezia arenaria* (Moris) Trappe	Pezizaceae	PA	Em
*Terfezia boudieri* Chatin	Pezizaceae	PA	Em
*Thaxterogaster purpurascens* (Fr.) Niskanen & Liimat.	Cortinariaceae	CT, ME	Em
*Thelephora anthocephala* (Bull.) Fr.	Thelephoraceae	CT, ME	Sl
*Thelephora caryophyllea* (Schaeff.) Pers.	Thelephoraceae	CT, ME, PA	Sl
*Thelephora palmata* (Scop.) Fr.	Thelephoraceae	CT, ME, PA	Sl
*Thelephora terrestris* Ehrh.	Thelephoraceae	AG, CL, CT, EN, ME, PA, RG, SR, TP	Sl
*Thelephora wakefieldiae* Zmitr., Shchepin, Volobuev & Myasnikov	Thelephoraceae	PA	Sl
*Tomentella badia* (Link) Stalpers	Thelephoraceae	PA, TP	Em
*Tomentella bryophila* (Pers.) M.J. Larsen	Thelephoraceae	PA, TP	Em
*Tomentella clavigera* Litsch.	Thelephoraceae	PA, TP	Em
*Tomentella coerulea* Höhn. & Litsch.	Thelephoraceae	PA, TP	Em
*Tomentella fuscocinerea* (Pers.) Donk	Thelephoraceae	PA, TP	Em
*Tomentella italica* (Sacc.) M.J. Larsen	Thelephoraceae	CT, ME	Em
*Tomentella lapida* (Pers.) Stalpers	Thelephoraceae	CT, ME	Em
*Tomentella lateritia* Pat.	Thelephoraceae	PA	Em
*Tomentella lilacinogrisea* Wakef.	Thelephoraceae	CT, ME	Em
*Tomentella oligofibula* M.J. Larsen, Beltrán-Tej. & Rodr.-Armas	Thelephoraceae	CT, ME	Em
*Tomentella pilosa* (Burt) Bourdot & Galzin	Thelephoraceae	CT, ME, PA	Em
*Tomentella punicea* (Alb. & Schwein.) J. Schröt.	Thelephoraceae	CT, ME, PA	Em
*Tomentella subtestacea* Bourdot & Galzin	Thelephoraceae	CT, ME, PA	Em
*Tomentella stuposa* (Link) Stalpers	Thelephoraceae	CT, ME, PA	Em
*Tomentella subcinerascens* Litsch.	Thelephoraceae	CT, ME	Em
*Tomentella subclavigera* Litsch.	Thelephoraceae	PA	Em
*Tomentella terrestris* (Berk. & Broome) M.J. Larsen	Thelephoraceae	PA	Em
*Trametes gibbosa* (Pers.) Fr.	Polyporaceae	AG, CL, CT, EN, ME, PA, RG, SR, TP	Sw
*Trametes hirsuta* (Wulfen) Lloyd	Polyporaceae	AG, CL, CT, EN, ME, PA, RG, SR, TP	Sw
*Trametes ochracea* (Pers.) Gilb. & Ryvarden	Polyporaceae	PA	Sw
*Trametes pubescens* (Schumach.) Pilát	Polyporaceae	CT, ME, PA	Sw
*Trametes suaveolens* (L.) Fr.	Polyporaceae	CT, ME, PA	Sw
*Trametes trogii* Berk.	Polyporaceae	CT, ME, PA, TP	Sw
*Trametes versicolor* (L.) Lloyd	Polyporaceae	AG, CL, CT, EN, ME, PA, RG, SR, TP	Sw
*Trechispora cohaerens* (Schwein.) Jülich & Stalpers	Hydnodontaceae	PA	Sw
*Trechispora farinacea* (Pers.) Liberta	Hydnodontaceae	PA	Sw
*Trechispora fastidiosa* (Pers.) Liberta	Hydnodontaceae	PA	Sw
*Trechispora microspora* (P. Karst.) Liberta	Hydnodontaceae	TP	Sw
*Trechispora nivea* (Pers.) K.H. Larss.	Hydnodontaceae	PA	Sw
*Trechispora stellulata* (Bourdot & Galzin) Liberta	Hydnodontaceae	PA, TP	Sw
*Trechispora stevensonii* (Berk. & Broome) K.H. Larss.	Hydnodontaceae	PA	Sw
*Tremella mesenterica* Retz.	Tremellaceae	AG, CL, CT, EN, ME, PA, RG, SR, TP	Sw
*Tremellodendropsis tuberosa* (Grev.) D.A. Crawford	Tremellodendropsidaceae	CT, ME	St
*Trichaptum abietinum* (Pers. ex J.F. Gmel.) Ryvarden	Incertae sedis	CT, ME, PA	Sw
*Trichaptum biforme* (Fr.) Ryvarden	Incertae sedis	AG, CL, CT, EN, ME, PA, RG, SR, TP	Sw
*Trichaptum fuscoviolaceum* (Ehrenb.) Ryvarden	Incertae sedis	PA, TP	Sw
*Tricharina gilva* (Boud. ex Cooke) Eckblad	Pyronemataceae	PA	St
*Tricharina praecox* (P. Karst.) Dennis	Pyronemataceae	CT, ME	St
*Trichoglossum hirsutum* (Pers.) Boud.	Geoglossaceae	CT, ME, PA	Sl
*Trichoglossum tetrasporum* Sinden & Fitzp.	Geoglossaceae	ME, PA	Sl
*Tricholoma acerbum* (Bull.) Quél.	Tricholomataceae	ME, PA	Em
*Tricholoma albatum* Velen.	Tricholomataceae	ME, PA	Em
*Tricholoma albobrunneum* (Pers.) P. Kumm.	Tricholomataceae	ME, PA	Em
*Tricholoma album* (Schaeff.) P. Kumm.	Tricholomataceae	ME, PA	Em
*Tricholoma apium* Jul. Schäff.	Tricholomataceae	ME, PA	Em
*Tricholoma argyraceum* (Bull.) Gillet	Tricholomataceae	ME, PA	Em
*Tricholoma atrosquamosum* Sacc.	Tricholomataceae	CT, ME, PA, TP	Em
*Tricholoma aurantium* (Schaeff.) Ricken	Tricholomataceae	CT, ME, PA	Em
*Tricholoma basirubens* (Bon) A. Riva & Bon	Tricholomataceae	CT, ME, PA	Em
*Tricholoma batschii* Gulden ex Mort. Chr. & Noordel.	Tricholomataceae	PA, TP	Em
*Tricholoma bresadolanum* Clémençon	Tricholomataceae	ME, PA	Em
*Tricholoma bufonium* (Pers.) Gillet	Tricholomataceae	CT, ME	Em
*Tricholoma caligatum* (Viv.) Ricken	Tricholomataceae	CT, ME, PA	Em
*Tricholoma cedretorum* (Bon) A. Riva	Tricholomataceae	AG, PA	Em
*Tricholoma chrysophyllum* A. Riva, C.E. Hermos. & Jul. Sánchez	Tricholomataceae	CT, ME	Em
*Tricholoma columbetta* (Fr.) P. Kumm.	Tricholomataceae	CT, ME, PA	Em
*Tricholoma equestre* (L.) P. Kumm.	Tricholomataceae	AG, CL, CT, EN, ME, PA, RG, SR, TP	Em
*Tricholoma evenosum* (Sacc.) Rea	Tricholomataceae	CT, ME, PA	Em
*Tricholoma focale* (Fr.) Ricken	Tricholomataceae	CT, ME, PA	Em
*Tricholoma fracticum* (Britzelm.) Kreisel	Tricholomataceae	AG, PA	Em
*Tricholoma fulvum* (DC.) Bigeard & H. Guill.	Tricholomataceae	CT, ME	Em
*Tricholoma gausapatum* (Fr.) Quél.	Tricholomataceae	PA, TP	Em
*Tricholoma imbricatum* (Fr.) P. Kumm.	Tricholomataceae	PA, TP	Em
*Tricholoma lascivum* (Fr.) Gillet	Tricholomataceae	PA, TP	Em
*Tricholoma leucoterreum* Mariotto & Turetta	Tricholomataceae	CT, ME	Em
*Tricholoma orirubens* Quél.	Tricholomataceae	AG, CL, CT, EN, ME, PA, RG, SR, TP	Em
*Tricholoma pardinum* (Pers.) Quél.	Tricholomataceae	CT, ME, PA	Em
*Tricholoma pessundatum* (Fr.) Quél.	Tricholomataceae	AG, PA	Em
*Tricholoma populinum* J.E. Lange	Tricholomataceae	AG, CL, CT, EN, ME, PA, RG, SR, TP	Em
*Tricholoma portentosum* (Fr.) Quél.	Tricholomataceae	ME, PA	Em
*Tricholoma psammopus* (Kalchbr.) Quél.	Tricholomataceae	AG, PA	Em
*Tricholoma pseudonictitans* Bon	Tricholomataceae	CT, ME, PA	Em
*Tricholoma quercetorum* Contu	Tricholomataceae	CT, ME	Em
*Tricholoma roseoacerbum* A. Riva	Tricholomataceae	CT, ME, PA	Em
*Tricholoma saponaceum* (Fr.) P. Kumm.	Tricholomataceae	AG, CT, ME, PA, TP	Em
*Tricholoma scalpturatum* (Fr.) Quél.	Tricholomataceae	PA, TP	Em
*Tricholoma sciodes* (Pers.) C. Martín	Tricholomataceae	CT, ME, PA	Em
*Tricholoma sejunctum* (Sowerby) Quél.	Tricholomataceae	AG, ME, PA	Em
*Tricholoma stans* (Fr.) Sacc.	Tricholomataceae	CT, ME, TP	Em
*Tricholoma stiparophyllum* (N. Lund) P. Karst.	Tricholomataceae	CT, ME, PA	Em
*Tricholoma striatum* (Schaeff.) Quél.	Tricholomataceae	CT, ME, PA, TP	Em
*Tricholoma sulphurescens Bres.*	Tricholomataceae	CT, ME	Em
*Tricholoma sulphureum* (Bull.) P. Kumm.	Tricholomataceae	AG, CL, CT, EN, ME, PA, RG, SR, TP	Em
*Tricholoma terreum* (Schaeff.) P. Kumm.	Tricholomataceae	AG, CL, CT, EN, ME, PA, RG, SR, TP	Em
*Tricholoma terreum* (Schaeff.) P. Kumm. var. *album* Antonín & A. Vágner	Tricholomataceae	AG, CL, CT, EN, ME, PA, RG, SR, TP	Em
*Tricholoma ustale* (Fr.) P. Kumm.	Tricholomataceae	AG, CL, CT, EN, ME, PA, RG, SR, TP	Em
*Tricholoma ustaloides* Romagn.	Tricholomataceae	AG, CL, CT, EN, ME, PA, RG, SR, TP	Em
*Tricholoma virgatum* (Fr.) P. Kumm.	Tricholomataceae	CT, ME, PA	Em
*Tricholomella constricta* (Fr.) Zerova ex Kalamees	Lyophyllaceae	CT, ME, PA	Sw
*Tricholomopsis rutilans* (Schaeff.) Singer	Incertae sedis	AG, CL, CT, EN, ME, PA, RG, SR, TP	Sw
*Tricholosporum goniospermum* (Bres.) Guzmán ex T.J. Baroni	Incertae sedis	CT, ME	Sl
*Trichophaea woolhopeia* (Cooke & W. Phillips) Boud.	Pyronemataceae	PA, TP	St
*Truncospora ochroleuca* (Berk.) Pilát	Polyporaceae	PA, TP	Sw
*Tubaria conspersa* (Pers.) Fayod	Tubariaceae	CT, ME, PA, TP	St
*Tubaria dispersa* (Pers.) Singer	Tubariaceae	CT, PA	St
*Tubaria furfuracea* (Pers.) Gillet	Tubariaceae	CL, CT, ME, PA, TP	St
*Tuber aestivum* Vittad.	Tuberaceae	AG, CL, CT, EN, ME, PA, RG, SR, TP	Em
*Tuber bellonae Quél.*	Tuberaceae	CT, ME	Em
*Tuber borchii* Vitt.	Tuberaceae	AG, CL, CT, EN, ME, PA, RG, SR, TP	Em
*Tuber brumale* Vittad.	Tuberaceae	ME, PA, SR	Em
*Tuber dryophilum* Tul. & C. Tul.	Tuberaceae	CT, ME	Em
*Tuber excavatum* Vittad.	Tuberaceae	ME, PA, SR, TP	Em
*Tuber foetidum* Vittad.	Tuberaceae	AG, PA, SR	Em
*Tuber gennadii* (Chatin) Pat.	Tuberaceae	CT, ME	Em
*Tuber magnatum* Picco	Tuberaceae	EN	Em
*Tuber mesentericum* Vittad.	Tuberaceae	ME, PA, SR	Em
*Tuber nitidum* Vittad.	Tuberaceae	PA, TP, SR	Em
*Tuber oligospermum* (Tul. & C. Tul.) Trappe	Tuberaceae	CT, PA	Em
*Tuber panniferum* Tul. & C. Tul.	Tuberaceae	ME, PA, TP, SR	Em
*Tuber puberulum* Berk. & Broome	Tuberaceae	AG, ME, PA, SR, TP	Em
*Tuber rufum* Pollini	Tuberaceae	AG, CL, CT, EN, ME, PA, RG, SR, TP	Em
*Tubulicium vermiferum* (Bourdot) Oberw. ex Jülich	Hydnodontaceae	AG, CL, CT, EN, ME, PA, RG, SR, TP	Sw
*Tubulicrinis calothrix* (Pat.) Donk	Hymenochaetaceae	AG, CL, CT, EN, ME, PA, RG, SR, TP	Sw
*Tubulicrinis medius* (Bourdot & Galzin) Oberw.	Hymenochaetaceae	AG, CL, CT, EN, ME, PA, RG, SR, TP	Sw
*Tubulicrinis subulatus* (Bourdot & Galzin) Donk	Hymenochaetaceae	AG, CL, CT, EN, ME, PA, RG, SR, TP	Sw
*Tulostoma brumale* Pers.	Agaricaceae	CL, CT, PA	Sm
*Tulostoma fimbriatum* Fr.	Agaricaceae	CT, ME, PA	Sm
*Tulostoma kotlabae* Pouzar	Agaricaceae	CT, ME	Sm
*Tulostoma melanocyclum* Bres.	Agaricaceae	CT, ME, PA	Sm
*Tulostoma squamosum* (J.F. Gmel.) Pers.	Agaricaceae	CT, ME, PA	Sm
*Tylopilus felleus* (Bull.) P. Karst.	Boletaceae	AG, CL, CT, EN, ME, PA, RG, SR, TP	Em
*Typhula contorta* (Holmsk.) Olariaga	Typhulaceae	CT, ME, PA	Sw
*Typhula fistulosa* (Holmsk.) Olariaga	Typhulaceae	CT, ME, PA	Sw
*Typhula juncea* (Alb. & Schwein.) P. Karst.	Typhulaceae	CT, ME, PA	Sw
*Vanderbylia fraxinea* (Bull.) D.A. Reid	Polyporaceae	CT, ME, PA	Sw
*Vararia ochroleuca* (Bourdot & Galzin) Donk	Peniophoraceae	CT, ME, PA	Sw
*Verpa conica* (O.F. Müll.) Sw.	Morchellaceae	CT, ME, PA	St
*Verpa digitaliformis* Pers.	Morchellaceae	CT, ME, PA	St
*Vitreoporus dichrous* (Fr.) Zmitr.	Irpicaceae	AG, PA	Sw
*Volvariella bombycina* (Schaeff.) Singer	Pluteaceae	CT, ME, PA	Sw
*Volvariella caesiotincta* P.D. Orton	Pluteaceae	CT, ME, PA	Sw
*Volvariella cookei* Contu	Pluteaceae	CT, ME, PA	Sw
*Volvariella hypopithys* (Fr.) Shaffer	Pluteaceae	CT, ME	Sw
*Volvariella media* (Schumach.) Singer	Pluteaceae	CT, ME, PA, TP	Sw
*Volvariella surrecta* (Knapp) Singer	Pluteaceae	CT, ME	Sw
*Volvariella volvacea* (Bull.) Singer	Pluteaceae	AG, CL, CT, EN, ME, PA, RG, SR, TP	Sw
*Volvopluteus gloiocephalus* (DC.) Vizzini, Contu & Justo	Pluteaceae	AG, CL, CT, EN, ME, PA, RG, SR, TP	Sl
*Vuilleminia comedens* (Nees) Maire	Vuilleminiaceae	AG, CL, CT, EN, ME, PA, RG, SR, TP	Sw
*Vuilleminia coryli* Boidin, Lanq. & Gilles	Vuilleminiaceae	ME, PA	Sw
*Vuilleminia cystidiata* Parmasto	Vuilleminiaceae	CT, ME, PA, TP	Sw
*Vuilleminia megalospora* Bres.	Vuilleminiaceae	CT, ME, PA, TP	Sw
*Vuilleminia pseudocystidiata* Boidin, Lanq. & Gilles	Vuilleminiaceae	CT, ME, PA, TP	Sw
*Xanthoporia radiata* (Sowerby) Ţura, Zmitr., Wasser, Raats & Nevo	Hymenochaetaceae	CT, ME, PA	Sw
*Xenasma pruinosum* (Pat.) Donk	Xenasmataceae	AG, CT, ME, PA, TP	Sw
*Xenasmatella vaga* (Fr.) Stalpers	Xenasmataceae	CT, ME, PA	Sw
*Xerocomellus chrysenteron* (Bull.) Šutara	Boletaceae	AG, CL, EN, CT, ME, PA, RG, SR, TP	Em
*Xerocomellus cisalpinus* (Simonini, H. Ladurner & Peintner) Klofac	Boletaceae	AG, CL, EN, CT, ME, PA, RG, SR, TP	Em
*Xerocomellus communis* Xue T. Zhu & Zhu L. Yang	Boletaceae	CT, ME	Em
*Xerocomellus dryophilus* (Thiers) N. Siegel, C.F. Schwarz & J.L. Frank	Boletaceae	CT, ME, PA, TP	Em
*Xerocomellus porosporus* (Imler ex Watling) Šutara	Boletaceae	CT, ME, PA, TP	Em
*Xerocomellus pruinatus* (Fr. & Hök) Šutara	Boletaceae	CT, ME, PA	Em
*Xerocomellus truncatus* (Singer, Snell & E.A. Dick) Klofac	Boletaceae	AG, CL, EN, CT, ME, PA, RG, SR, TP	Em
*Xerocomus ferrugineus* (Schaeff.) Alessio	Boletaceae	AG, CT, ME, PA	Em
*Xerocomus rubellus* Quél.	Boletaceae	CT, ME, PA	Em
*Xerocomus subtomentosus* (L.) Quél.	Boletaceae	AG, CL, CT, EN, ME, PA, RG, SR, TP	Em
*Xeromphalina campanella* (Batsch) Kühner & Maire	Mycenaceae	CT, ME, PA	Sw
*Xerula pudens* (Pers.) Singer	Physalacriaceae	AG, CL, CT, EN, ME, PA, RG, SR, TP	Sw
*Xylaria hypoxylon* (L.) Grev.	Xylariaceae	AG, CL, CT, EN, ME, PA, RG, SR, TP	Sw
*Xylaria longipes* Nitschke	Xylariaceae	AG, CL, CT, EN, ME, PA, RG, SR, TP	Sw
*Xylaria polymorpha* (Pers.) Grev.	Xylariaceae	AG, CL, CT, EN, ME, PA, RG, SR, TP	Sw
*Xylaria sicula* Pass. & Beltrani	Xylariaceae	AG, CL, CT, EN, ME, PA, RG, SR, TP	Sw
*Xylodon detriticus* (Bourdot) K.H. Larss., Viner & Spirin	Schizoporaceae	CT, ME, PA	Sw
*Xylodon nesporii* (Bres.) Hjortstam & Ryvarden	Schizoporaceae	CT, ME, PA	Sw
*Xylodon rimosissimus* (Peck) Hjortstam & Ryvarden	Schizoporaceae	CT, ME, PA	Sw
*Xylodon sambuci* (Pers.) Ţura, Zmitr., Wasser & Spirin	Schizoporaceae	CT, ME, PA	Sw
*Zhuliangomyces illinitus* (Fr.) Redhead	Amanitaceae	CT, ME, PA	Sl
*Zhuliangomyces lenticularis* (Lasch) Redhead	Amanitaceae	CT, ME, PA	Sl
*Zhuliangomyces ochraceoluteus* (P.D. Orton) Redhead	Amanitaceae	CT, ME, PA	Sl

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
