# Peer review of "The Checklist of Sicilian Macrofungi: Second Edition"

_jof, 2022, doi:10.3390/jof8060566_

Round 1

Reviewer 1 Report

This manuscript provided the updated checklist of fungi from Sicily. It is suitable for publication of Journal of Fungi. This study reported the important information on the species diversity of fungi from Sicily. I think the manuscript can be accepted after minor revision in Journal of Fungi.

The following are some small suggestions:

  1. This study is focused on the macrofungi, the micromycetes are not mentioned. It is better to change fungi to macrofungi in the title and other relevant parts.
  2. In the Introduction section, it is better to provide more information on the background of research advances of macrofungi in Sicily, Italy, Europe and even in the world.
  3. Are all the specimens identified only based on morphological features, the sequences analyses were not used?
  4. The footnotes for the Provinces and Ecological categories are not provided.
  5. The family status of some taxa need to be checked. Such as, Amaropostia stiptica is Incertae Sedis, Schizopora paradoxa is Schizoporaceae, ……
  6. The conclusion part needs to be revised, the current form is not like a conclusion.

Author Response

We resubmit the corrected version of the manuscript on which all the changes proposed by referee 1 and referee 2 have been incorporated. 

Best regards,

Dr. Fortunato Cirlincione

Reviewer 2 Report

This manuscript list 1919 infraspecific taxa of 508 genera belonging to 150 families in the Sicilian territory based on the data during 30 years of long-term observation, which is valuable for the research of fungal diversity in Italy, but it should be improved in the English writing and the structure, especially in the discussion. The comments are added in the pdf. Please check them. 

Author Response

(The authors gave the same response as above.)

Round 2

Reviewer 2 Report

Thank you so much for improving your manuscript. This version is appropriate for publication in this JoF.